# Using maximal information auxiliary variables to improve synthetic data generation based on TabPFN foundation models

**Elias Chaibub Neto**
Sage Bionetworks, Seattle, WA 98121, USA
`elias.chaibub.neto@sagebase.org`

## Abstract

Synthetic data generation for tabular datasets is shifting toward the use of large, general-purpose foundation models. TabPFN, a state-of-the-art example, uses in-context learning to generate probabilistic predictions conditioned on observed examples in a single forward pass. However, when variables are only weakly associated with others, the model's ability to generate realistic synthetic data deteriorates, as the context examples provide little predictive signal. To address this, we introduce the maximal information auxiliary variable (MIAV) strategy, which increases context information with auxiliary variables constructed by rank-matching random noise variables to real data. We establish theoretical properties of the approach which explain its good performance for weakly associated variables. Additional practical advantages of the MIAV approach include improved computational efficiency and invariance to variable order during the synthetic data generation process. Empirical evaluations, on simulated and real datasets, illustrate how the MIAV strategy improves data generation when compared to direct application of TabPFN, and is competitive against other baselines. To illustrate the generality of the MIAV approach we also present an implementation based on the TabICL model (a more scalable tabular foundation model restricted to classification tasks) for performing synthetic data generation on categorical datasets. Overall, MIAV offers an effective foundation model–based alternative to bespoke synthetic data generators.

## 1 Introduction

Accessible data is crucial for advancing machine learning research. In practice, however, real-world datasets often contain sensitive information, restricting their open distribution within the research community. A promising solution is the generation of synthetic datasets that closely replicate the properties of real data while avoiding direct disclosure (Lu et al., 2023).

While synthetic data has long been explored through bespoke statistical models and machine learning algorithms, the field is now undergoing a paradigm shift driven by advances in large-scale, general-purpose models. Traditional approaches, such as those by Borisov et al. (2023), Cresswell and Kim (2024), Jolicoeur-Martineau et al. (2024), Kotelnikov et al. (2023), Nowok et al. (2016), Reiter (2005), Shi et al. (2025), Watson et al. (2023), Xu et al. (2019), Young et al. (2009), Zhang et al. (2024), Xu et al. (2025), among many others, typically rely on dataset-specific training, demand substantial domain expertise, and often struggle with knowledge transfer across datasets. Tabular foundation models (Hollmann et al., 2023; den Breejen et al., 2024; Koshil et al., 2024; Feuer et al., 2024; Ma et al., 2024a; Ma et al., 2024b; Xu et al., 2024; Zeng et al., 2024; Muller et al., 2025; Hollmann et al., 2025; Qu et al., 2025; Garg et al., 2025) offer a promising alternative. By learning broad, transferable representations of tabular data, they enable strong performance in supervised learning tasks with minimal additional training.

In particular, TabPFN (Hollmann et al., 2025) represents a state-of-the-art tabular foundation model, trained on millions of diverse synthetic datasets covering a wide range of feature types, noise structures, and functional relationships. This diversity allows it to leverage a broad, transferable prior over tabular data distributions. TabPFN enjoys a solid theoretical foundation as it corresponds

to a prior-data fitted network (PFN) (Müller et al., 2022) and can be interpreted as approximating Bayesian prediction under the prior induced by its synthetic training data. TabPFN relies on in-context learning (Brown et al., 2020) (ICL) for generating probabilistic predictions. At inference time, the pre-trained foundation model employs training features, $\mathbf{X}^{tr}$, and training targets, $\mathbf{y}^{tr}$, as the "context" data, whereas the test set features, $\mathbf{X}^{ts}$, play the role of the "query". The output of the query is a sample/prediction $\hat{\mathbf{y}}^{ts}$ from the posterior predictive distribution of $\mathbf{y}^{ts}$, $P(\boldsymbol{y}^{ts} \mid \boldsymbol{X}^{ts}, \boldsymbol{X}^{tr}, \boldsymbol{y}^{tr})$, generated by a single forward pass through the model's neural network. TabPFN has been shown to achieve state-of-the-art performance in classification and regression tasks in small datasets and, due to its generative nature, can also be directly used to perform synthetic data generation.

Despite its promise, applying TabPFN directly to synthetic data generation reveals an important limitation: the method performs poorly for variables that are only weakly associated with the rest of the dataset. This is expected: when the target data $\mathbf{y}$ is uncorrelated with the features $\mathbf{X}$, the context examples $\mathbf{X}^{tr}, \mathbf{y}^{tr}$ provide no useful signal for learning how to map $\mathbf{X}$ to $\mathbf{y}$. Consequently, when queried with $\mathbf{X}^{ts}$, the model is unable to approximate the distribution of $\mathbf{y}^{ts}$. While this limitation is less consequential for supervised learning tasks, it poses a significant caveat for synthetic data generation (see Appendix A for details). In principle, exemplar-based declarative programming strategies could mitigate this issue, but doing so would likely require fine-tuning, or even retraining the foundation models.

In this paper, we address this problem by showing how to generate high-quality synthetic datasets with the current TabPFN model. Our approach leverages maximal information auxiliary variables (MIAV) for in-context learning. We construct these variables through simple rank-matching of random noise to the real data and establish two key theoretical properties: (i) conditional on its MIAV, a variable $X_j$ is independent of all other variables, and (ii) the MIAV of $X_j$ retains maximal information about $X_j$ in a non-parametric, information-theoretic sense (see Theorem 1). We further demonstrate that MIAV-based synthetic data generation corresponds to the correct factorization of the posterior predictive distribution conditioned on the original data and MIAVs. Together, these results provide the foundation for more effective synthetic data generation strategies using TabPFN models.

In addition to its theoretical strengths and its ability to generate high-quality synthetic data under weak association settings, our proposed strategy offers several practical advantages. First, unlike the direct synthetic data generation approach of Hollmann et al. (2025), which is sensitive to variable order and therefore requires aggregating results across multiple variable order permutations, our method is invariant to variable order. Second, regarding computational efficiency, TabPFN's runtime for a fixed sample size is primarily determined by the number of context features, as its complexity scales quadratically with the number of features. Since our approach uses only one feature per variable when generating synthetic data, it attains maximal efficiency and eliminates the need for aggregation across multiple runs.

To illustrate the issues around the direct use of TabPFN for synthetic data generation under weak association settings, we describe a couple of direct implementations and compare their performance against the MIAV approach in simulated data experiments (where we are able to control the strength of the statistical associations among the data variables), as well as, on extensive real-world data experiments based on 43 distinct datasets. (For completeness, we also include comparisons against other baseline generators.)

We conduct our evaluations in the setting of privacy-preserving data sharing (Rajotte et al., 2022), where the objective is to produce synthetic copies of real datasets that retain their statistical properties while simultaneously mitigating privacy risks. Accordingly, we assess the performance of the TabPFN-based synthetic data generation strategies using both data fidelity and privacy metrics.

Importantly, our synthetic data generation strategy can be directly applied to other tabular foundation models that approximate Bayesian inference. To demonstrate this, we also implemented our approach using the more scalable TabICL foundation model (Qu et al., 2025), a recently proposed alternative to TabPFN that alleviates some of its data size limitations. (Since the current version of TabICL only supports classification, our implementation and evaluation were restricted to 8 additional real-world categorical datasets.) These additional results highlight the generality of our strategy and suggest that, as PFN-based tabular foundation models continue to evolve, they can be seamlessly integrated into our framework.

In summary, this paper proposes an effective, computationally efficient, and generalizable approach that leverages the in-context learning capabilities of modern tabular foundation models to generate synthetic data aiming to facilitate data sharing. It provides a foundation-model alternative to traditional synthetic data generators built on the earlier paradigm of bespoke ML models.

## 2 NOTATION

Throughout the text, random variables are represented in italics, vectors of random variables are shown in boldface, e.g., $\boldsymbol{X} = (X_1, \ldots, X_p)^t$, and $P(\boldsymbol{X})$ is used for probability statements involving random variables. Data matrices and data vectors are represented in uppercase and lower case boldface, respectively. (E.g., if $\mathbf{X}$ is an $n \times p$ matrix, than the $j$th column of $\mathbf{X}$ is represented by $\mathbf{x}_j$.) We use the notation $\mathbf{X}_{-j}$ to represent the matrix obtained by removing the $j$th column from $\mathbf{X}$, and the notation $\mathbf{X}_{<j}$ to represent the matrix comprised by the first $j-1$ columns of $\mathbf{X}$. (Similarly, in the case of a set of random variables, we use the notation $\boldsymbol{X}_{<j}$ to represent the subset of $\boldsymbol{X}$ containing elements 1 to $j-1$.) We adopt the superscripts $tr$ and $ts$ to represent the training and test sets, and we let $q_\theta(\mathbf{x}_j^{ts} \mid \mathbf{X}_{-j}^{ts}, \mathbf{X}^{tr})$ represent a TabPFN model (either a regression or a classification model depending on whether the variable $j$ is numeric or categorical), where $\hat{\mathbf{x}}_j^{ts} \sim q_\theta(\mathbf{x}_j^{ts} \mid \mathbf{X}_{-j}^{ts}, \mathbf{X}^{tr})$ represents the prediction generated by the model. TabPFN uses both the training features, $\mathbf{X}_{-j}^{tr}$, and training targets, $\mathbf{x}_j^{tr}$, as examples during the in-context training step, but only the test set features, $\mathbf{X}_{-j}^{ts}$, during the in-context query step, where the model is asked to generate a prediction of the test set targets based on the examples from the training set and the values of the test set features.

Because our goal is to generate synthetic data copies of given datasets, rather than performing supervised learning tasks, our notation does not explicitly differentiate between feature and targets variables (as the same variable can sometimes play the role of a feature and sometimes of a target during the synthetic data generation process). Hence, we use the notation $q_\theta(\mathbf{x}_j^{ts} \mid \mathbf{X}_{-j}^{ts}, \mathbf{X}^{tr}) = q_\theta(\mathbf{x}_j^{ts} \mid \mathbf{X}_{-j}^{ts}, \mathbf{X}_{-j}^{tr}, \mathbf{x}_j^{tr})$ instead of the notation $q_\theta(\mathbf{y}^{ts} \mid \mathbf{X}^{ts}, \mathbf{X}^{tr}, \mathbf{y}^{tr})$, more commonly used in the TabPFN literature.

## 3 RELATED WORK

Although the literature on synthetic data generation (SDG) using bespoke machine learning models is extensive (see Bond-Taylor et al., 2021; Lu et al., 2023, and references therein), SDG based on tabular foundation models remains underexplored. To the best of our knowledge, only two prior studies have addressed this problem. Ma et al. (2023) introduced the TabPFGen algorithm, which relied on an earlier version of TabPFN (Hollmann et al., 2023) that could not handle regression tasks. To produce continuous data, TabPFGen employed an energy-based procedure for generating features conditional on classification labels. Such procedures are no longer necessary with the current version of TabPFN (Hollmann et al., 2025), which supports both classification and regression and can directly generate categorical and numerical data. SDG was also discussed in Hollmann et al. (2025), but only superficially: it was demonstrated on a single dataset without any formal evaluation of synthetic data quality. That work primarily focused on establishing TabPFN's state-of-the-art performance in supervised learning, with SDG presented merely as a secondary capability. In this paper, we (i) highlight the limitations of the direct SDG approach in Hollmann et al. (2025), (ii) propose a more effective alternative strategy, and (iii) conduct extensive evaluations of TabPFN-based SDG methods.

## 4 DIRECT STRATEGIES FOR SDG BASED ON TABPFN

Here we describe the simple strategy, suggested in Hollmann et al. (2025), for performing synthetic data generation with the TabPFN model, alongside an alternative variation of this direct approach. Their main limitations are discussed and illustrated using simulated datasets (where we can control the strength of the statistical associations between the variables).

Let $P(\boldsymbol{X}^{ts} \mid \boldsymbol{X}^{tr})$ represent the posterior predictive distribution (PPD) of the test data conditional on the training data. This conditional joint probability distribution can be fully factorize as,

$$P(\boldsymbol{X}^{ts} \mid \boldsymbol{X}^{tr}) = \prod_{j=1}^{p} P(X_j^{ts} \mid \boldsymbol{X}_{<j}^{ts}, \boldsymbol{X}^{tr}), \tag{1}$$

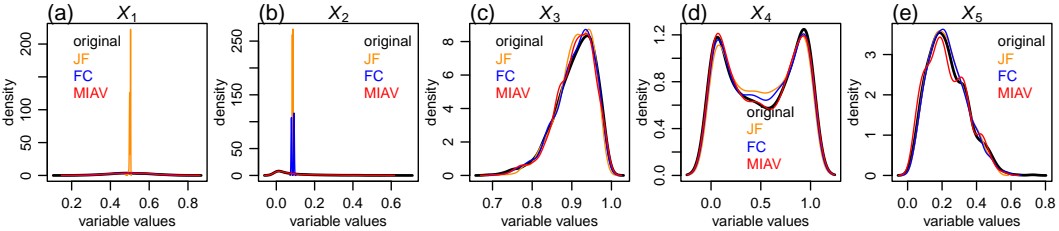

Figure 1: Comparison of marginal distributions generated with the JF, FC, and MIAV strategies.

where $\boldsymbol{X}_{<j}$ represents a subset of $\boldsymbol{X}$ containing elements 1 to $j-1$, and $p$ represents the number of features.

Hollmann et al. (2025) suggested generating synthetic data by following the factorization of the joint PPD,

$$P(\boldsymbol{X}^{ts} \mid \boldsymbol{X}^{tr}) \approx \prod_{j=1}^{p} q_\theta(\mathbf{x}_j^{ts} \mid \mathbf{X}_{<j}^{ts}, \mathbf{X}_{<j}^{tr}, \mathbf{x}_j^{tr}), \tag{2}$$

where, due to the in-context learning (ICL) nature of TabPFN, the conditioning on the training set is done on $\mathbf{X}_{<j}^{tr}$ and $\mathbf{x}_j^{tr}$ rather than $\mathbf{X}^{tr}$. Note that the approximation in equation 2 represents, in actuality, two different levels of approximation. The first is w.r.t. the approximation of the distribution $P(X_j^{ts} \mid \boldsymbol{X}_{<j}^{ts}, \boldsymbol{X}^{tr})$ by the distinct distribution $P(X_j^{ts} \mid \boldsymbol{X}_{<j}^{ts}, \boldsymbol{X}_{<j}^{tr}, X_j^{tr})$, which no longer conditions on training variables $X_{j'}$ for $j' > j$. The second is w.r.t. the approximation of $P(X_j^{ts} \mid \boldsymbol{X}_{<j}^{ts}, \boldsymbol{X}_{<j}^{tr}, X_j^{tr})$ by the transformer model $q_\theta(\mathbf{x}_j^{ts} \mid \mathbf{X}_{<j}^{ts}, \mathbf{X}_{<j}^{tr}, \mathbf{x}_j^{tr})$.

Furthermore, because the TabPFN model cannot condition on an empty set (as you need to provide the model with some input for it to perform ICL) the first product term in equation 2 requires conditioning in a variable $X_0$, which is not part of the data. Following the suggestion by Hollmann et al. (2025), we adopt a random noise feature as our $X_0$. A detailed description of the implementation of this strategy, denoted "factorization of the joint PPD" (or JF for short), is provided in Algorithms 2 and 3 in Appendix B. (As pointed by Hollmann et al. (2025), the order of the variables in the joint factorization of the PPD can also affect the results, and Hollmann et al. suggest using a permutation sampling approximation of Janossy pooling[1] to remedy this issue. This requires, however, the generation and aggregation of multiple synthetic datasets generated from random permutations of the order of the columns of the real data and is not implemented in our experiments.)

Figure 1 illustrates the application of the JF generation strategy (and other approaches that will be described later) to a simulated dataset containing data draw from highly correlated beta distributions. (See Appendix C for details.) To simulate an uninformative feature, we randomly shuffled the data of variable $X_2$, so that it is completely uncorrelated with the other variables. (Figure 5a in Appendix D shows the correlation matrix for these variables.) The black densities represent the original ("real") data while the orange ones show the synthetic data generated with the JF strategy. Not surprisingly, this example shows that TabPFN provided very poor approximations for the distributions of $X_1$ and $X_2$. In the case of $X_1$, the ICL based on $X_0$ is poor because $X_0$ is a random noise variable which contains no information about $X_1$. In the case of $X_2$, the ICL again fails because $X_1$ does not contain information about $X_2$ (which is uncorrelated from all other variables).

An alternative approach is to use the full conditional distributions of each variable in the synthetic data generation (denoted as FC, for short). This strategy is implemented using the factorization,

$$\prod_{j=1}^{p} q_\theta(\mathbf{x}_j^{ts} \mid \mathbf{X}_{-j}^{ts}, \mathbf{X}^{tr}) \tag{3}$$

where $\mathbf{X}_{-j}$ represents a subset matrix obtained by dropping column $j$ from $\mathbf{X}$. A detailed description of our implementation of this strategy is provided in Algorithms 4 and 5 in Appendix B.

---

[1]Namely, Hollmann et al. (2025) generate $N$ distinct synthetic datasets, using different random permutations of the order of the variables during the synthetic data generation process, and average the results across the $N$ synthetic datasets to reduce variability and decrease the dependence of the result on the variable order.

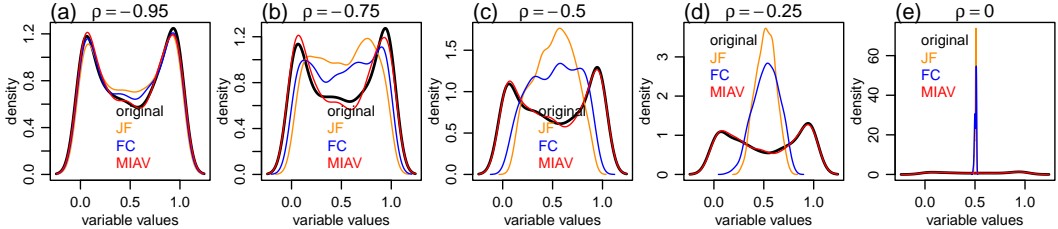

Figure 2: Degrading performance of JF and FC as association strength decreases. MIAV is unaffected.

Despite the fact that this fully conditional factorization does not correspond to a proper factorization of the PPD, this strategy has a few practical advantages. First, it eliminates the need for coming up with a $X_0$ variable, as the data for $X_1$ is generated from it's full conditional distribution. Second, the generation of each variable leverages information from all other variables. Third, this approach is unaffected by the order of the variables. It's main practical disadvantage is that it is more expensive to compute since increasing numbers of variables lead to increases in compute time. (As pointed by Hollmann et al. (2025), the time complexity of TabPFN is $O(n^2 + p^2)$, where $n$ and $p$ represent, respectively, the number of rows and columns of the data.)

The blue densities in Figure 1 illustrate the application of the FC generation strategy. Now, the distribution of $X_1$ is nicely recovered by the FC strategy. (This is easier to visualize in Figure 5m in Appendix D, which reports the same results as Figure 1 using a different display.) The approach, however, still fails for $X_2$ because this variable is uncorrelated to all other variables.

In Appendix E we describe additional variations of the JF direct data generation strategy.

### 4.1 PERFORMANCE DEGRADATION IN DATASETS CONTAINING WEAK ASSOCIATIONS

The illustrative examples in Figure 1 were based in data simulated with very strong correlations. The performance of direct strategies such as JF and FC, however, is strongly influenced by the strength of the statistical associations among the variables. Because TabPFN relies on ICL for generating predictions, and weakly associated features provide little information about the target variable, its performance suffers in datasets with weakly associated variables. To illustrate this point, Figure 2 presents the application of the JF and FC strategies (among others) to datasets with decreasing correlation strengths. (The association strength is controlled by the $\rho$ parameter, as described in Appendix C). Due to space limitations, the figure only reports results for a single variable ($X_4$). The full results are presented in Figures 5 to 9 in the Appendix D. Figure 2 clearly shows that the quality of the synthetic data generated by the JF and FC approaches (orange and blue densities) decreases with decreasing correlation strengths.

## 5 CONSTRUCTING MAXIMAL INFORMATION AUXILIARY VARIABLES

The previous section illustrates how direct application of TabPFN for synthetic data generation is problematic when variables lack strong statistical associations with other variables. A simple strategy for improving the synthetic data quality is to augment the dataset with auxiliary variables that are highly associated with the real data variables.

To this end, inspired by recent work in non-parametric and model free data synthesis (Chaibub Neto, 2025), we a propose a simple approach in which we rank-match arbitrary noise variables to the real data variables. The procedure is described in detail in Algorithm 1, and its output is what we denote as a maximal information auxiliary variable (MIAV).

The basic idea is to induce a monotonic mapping between a random noise variable and the real data vector $\mathbf{x}_j$. Starting in line 2, the algorithm first draws a sample of size $n$ (equal to the length of $\mathbf{x}_j$) from an arbitrary random noise variable and then sorts it from lowest to highest values. (In our implementation we sample random noise from a uniform distribution in the [0, 1] interval. This choice is, nonetheless, unimportant as the approach is not sensitive to the noise distribution used to construct the MIAV. See Appendix L for further details.) In lines 3 to 8 the algorithm ranks the values of $\mathbf{x}_j$. If

---

**Algorithm 1** GenerateMaximalInformationAuxiliaryVariable($\mathbf{x}_j$)

1: **Input:** data vector, $\mathbf{x}_j$
2: $\mathbf{m}_j \leftarrow \text{Sort}(\text{GenerateRandomNoiseVector}(n = \text{length}(\mathbf{x}_j)))$ {Generate a sorted random noise vector.}
3: **if** $\mathbf{x}_j$ is numeric **then**
4:  $\mathbf{r} \leftarrow \text{Rank}(\mathbf{x}_j)$ {Compute the ranks of $\mathbf{x}_j$. Ties are broken at random.}
5: **end if**
6: **if** $\mathbf{x}_j$ is categorical **then**
7:  $\mathbf{r} \leftarrow \text{NumericRankEncondingOfCategoticalVariables}(\mathbf{x}_j)$ {Described in Algorithm 10 in the Appendix.}
8: **end if**
9: $\mathbf{m}_j \leftarrow \mathbf{m}_j[\mathbf{r}]$ {Re-order the entries of $\mathbf{m}_j$ according to the ranks of $\mathbf{x}_j$. The result is a vector $\mathbf{m}_j$ with identical ranks as $\mathbf{x}_j$.}
10: **Output:** the auxiliary variable $\mathbf{m}_j$

---

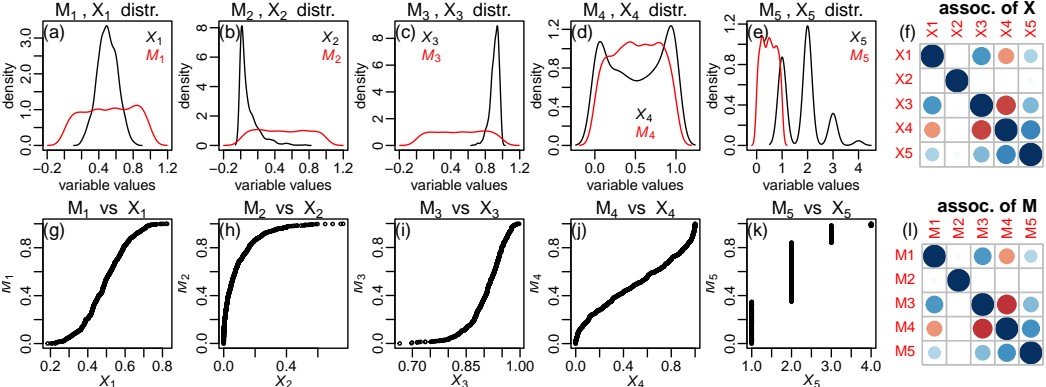

Figure 3: MIAVs illustrative example. In panels f and l, positive correlations are represented in blue and negative correlations are represented in red.

$\mathbf{x}_j$ is numeric, the algorithm computes its ranks in the standard way (line 4), breaking ties among identical values using the random assignment approach. If $\mathbf{x}_j$ is categorical, the algorithm applies a numeric rank encoding to the categorical variables, as described in Algorithm 10 in Appendix F, and originally proposed by Chaibub Neto (2025). (In a nutshell, Algorithm 10 counts the number elements of $\mathbf{x}_j$ in each factor level and distributes numerical ranks ranging from 1 to $n$ randomly within each level of the categorical variable $\mathbf{x}_j$. Appendix F also includes an illustrative example.) Finally, in line 9 the algorithm re-orders the entries of $\mathbf{m}_j$ according to the ranks of $\mathbf{x}_j$. The result is a vector $\mathbf{m}_j$ with identical ranks to $\mathbf{x}_j$, that is, the position of the lowest value of $\mathbf{m}_j$ is the same as the position of the lowest value of $\mathbf{x}_j$, the position of the 2nd lowest value of $\mathbf{m}_j$ is the same as the second lowest value of $\mathbf{x}_j$, and so on. Figure 3 provides some examples. Panels a to e show the distributions of the $X_j$ variables in black and their respective MIAVs, $M_j$, in red. (The MIAVs follow uniform distributions since in our implementation we draw random noise from uniform distributions.) Variable $X_5$ is discrete assuming values 1, 2, 3, and 4. Panels g to k show scatterplots of the $M_j$ vs $X_j$ data, illustrating the monotonic relations. Panels f and l show the pairwise associations for the $X_j$ and $M_j$ variables, respectively, and illustrate that, as expected, the $M_j$ variables recapitulate the associations of the $X_j$ data.

This procedure has some nice theoretical properties, described in the following result.

**Theorem 1.** *Let $M_j$ represent the auxiliary variable generated from $X_j$ by Algorithm 1, and $Y$ represent an arbitrary variable other than $X_j$ or $M_j$. Then, non-parametrically,*

1. *$I(X_j; Y \mid M_j) = 0$, i.e., the conditional mutual information of $X_j$ and $Y$ given $M_j$ is 0.*

2. *$H(X_j \mid M_j) = 0$, i.e., the conditional entropy of $X_j$ given $M_j$ is 0.*

The proof is presented in Appendix G. Note that this result holds non-parametrically, in the sense that any continuous variable is first discretized into $n$ bins (where $n$ represents sample size), so that we can use the discrete probability (and non-parametric) definitions of these information-theoretic

quantities. This is justifiable because a sample of size $n$ from a continuous variable can always be viewed as the sample of a discrete variable with $n$ distinct levels, each observed with frequency $1/n$.

This result shows that the auxiliary variable $M_j$, generated by Algorithm 1 has the following two key properties. First, it contains maximal information about $X_j$. (This holds in the conditional entropy sense, since $H(X_j \mid M_j) = 0$ implies that $X_j$ is completely determined by $M_j$ in a non-parametric rank-based sense.) Second, conditional of $M_j$, $X_j$ is statistically independent of any other variables. These two properties are key for the synthetic data generation approach that we propose next.

## 6  SDG USING MAXIMAL INFORMATION AUXILIARY VARIABLES

Let $\boldsymbol{M} = (M_1, \ldots, M_p)^t$ represent the set of random variables $M_j$ generated by Algorithm 1. Now, consider the augmented posterior predictive distribution of $\boldsymbol{X}^{ts}$ given $\boldsymbol{X}^{tr}$, $\boldsymbol{M}^{ts}$, and $\boldsymbol{M}^{tr}$,

$$P(\boldsymbol{X}^{ts} \mid \boldsymbol{X}^{tr}, \boldsymbol{M}^{ts}, \boldsymbol{M}^{tr}) = \prod_{j=1}^{p} P(X_j^{ts} \mid \boldsymbol{X}_{<j}^{ts}, \boldsymbol{X}^{tr}, \boldsymbol{M}^{ts}, \boldsymbol{M}^{tr}). \tag{4}$$

Now re-writing,

$$P(X_j^{ts} \mid \boldsymbol{X}_{<j}^{ts}, \boldsymbol{X}^{tr}, \boldsymbol{M}^{ts}, \boldsymbol{M}^{tr}) = P(X_j^{ts} \mid \boldsymbol{X}_{<j}^{ts}, \boldsymbol{X}^{tr}, \boldsymbol{M}_{-j}^{ts}, M_j^{ts}, \boldsymbol{M}_{-j}^{tr}, M_j^{tr}), \tag{5}$$

and recalling that $X_j^{ts}$ is independent from all other variables conditional on $M_j^{ts}$, its follows that,

$$P(X_j^{ts} \mid \boldsymbol{X}_{<j}^{ts}, \boldsymbol{X}^{tr}, \boldsymbol{M}_{-j}^{ts}, M_j^{ts}, \boldsymbol{M}_{-j}^{tr}, M_j^{tr}) = P(X_j^{ts} \mid M_j^{ts}), \tag{6}$$

which, for the same reason, can also be re-expressed as,

$$P(X_j^{ts} \mid M_j^{ts}) = P(X_j^{ts} \mid M_j^{ts}, M_j^{tr}, X_j^{tr}), \tag{7}$$

a format that is better suited for performing in-context learning with a PFN model. Hence, the PPD augmented with the set of maximal information auxiliary variables can formally be expressed as,

$$P(\boldsymbol{X}^{ts} \mid \boldsymbol{X}^{tr}, \boldsymbol{M}^{ts}, \boldsymbol{M}^{tr}) = \prod_{j=1}^{p} P(X_j^{ts} \mid M_j^{ts}, M_j^{tr}, X_j^{tr}), \tag{8}$$

and readily approximated by a trained TabPFN model as,

$$P(\boldsymbol{X}^{ts} \mid \boldsymbol{X}^{tr}, \boldsymbol{M}^{ts}, \boldsymbol{M}^{tr}) \approx \prod_{j=1}^{p} q_\theta(\mathbf{x}_j^{ts} \mid \mathbf{m}_j^{ts}, \mathbf{m}_j^{tr}, \mathbf{x}_j^{tr}), \tag{9}$$

where ICL for each variable $X_j$ is performed by training on $\mathbf{m}_j^{tr}$ and $\mathbf{x}_j^{tr}$ and querying on $\mathbf{m}_j^{ts}$.[2]

This formulation, denoted the "Maximal Information Auxiliary Variables" strategy (or MIAV strategy for short) has several practical advantages. First, it is the most efficient strategy in terms of computation, since ICL of each variable $X_j$ is performed using a single variable $M_j$ (recall that the complexity of the TabPFN model scales quadratically on the number of columns of the table). Appendix H reports complexity analyses and compute time benchmark experiments comparing the MIAV, JF, and FC strategies. Second, contrary to all the other direct generation strategies described in Section 4, the MIAV approach is based on a proper factorization of the (augmented) PPD (the approximation in equation 9 is only w.r.t. the transformer model approximation to the predictive distribution). Third, contrary to the JF strategy, the MIAV approach is invariant with respect to the order of the dataset columns. Fourth, and most importantly, the MIAV approach handles uninformative features and performs well in datasets containing weakly associated features. This last point is illustrated by the red densities in Figures 1 and 2 (see also panels s to w in Figures 5 to 9 in Appendix D), where the MIAV approach closely recapitulates the original marginal distributions. Quite importantly, note that while the augmented joint PPD in equation 8 factorizes into separate $P(X_j^{ts} \mid X_j^{tr}, M_j^{ts}, M_j^{tr})$ components, the synthetic data generated by the $q_\theta(\mathbf{x}_j^{ts} \mid \mathbf{x}_j^{tr}, \mathbf{m}_j^{tr}, \mathbf{m}_j^{ts})$ still recapitulates the association structure of $\mathbf{X}$ because this association is indirectly induced by the MIAVs (recall that $\mathbf{M}$ mimics the association structure in $\mathbf{X}$, as illustrated in panels f and l of Figure 3. Algorithms 6 and 7 in Appendix B provide implementation details about the MIAV synthetic data generation approach.

---

[2]Here, it is important to clarify that we are able to condition on the test set auxiliary variables, $\boldsymbol{M}^{ts}$, because we always have unrestricted access to the full $\mathbf{X}$ data (which is only split into $\mathbf{X}^{tr}$ and $\mathbf{X}^{ts}$ for the sake of ICL). Hence, the test set is always available and we can generate the corresponding MIAV matrix $\mathbf{M}^{ts}$. (Our goal is to generate a synthetic copy of the original data, rather than making predictions about unseen test data.)

## 7 EXPERIMENTS BASED ON TABPFN MODELS

**Experiments.** We performed three sets of experiments. The first, used simulated data draw from correlated beta distributions (generated as described in Appendix C), for which we can control the strength of the associations among the variables. The second experiment, used a subset of 36 real-world datasets (Table 4) from the OpenML-CC18 benchmark suite (Bischl et al., 2021). The third one, used 7 additional datasets (Table 5) evaluated in Hansen et al. (2023) and Chaibub Neto (2025). (Appendix I.3 contains further information and describes our rationale for selecting these datasets.)

**Data splits.** In each of the three experiment sets, every dataset was divided into two equal subsets. The first, referred to as the original data, was provided to the synthesizers, while the second, the holdout data, was never accessed by them. (Appendix I.1 provides further details about the data splits, including the description of additional data splits performed in the original data for generating the training and test sets used by the TabPFN models when performing in-context learning.)

**Evaluation metrics.** Synthetic datasets generated from the original data were evaluated using fidelity, utility, and privacy metrics. Fidelity was assessed with the average KS-test statistic (KS), the L2 distance between association matrices (L2D), the detection test (DT), the Wasserstein distance (WD), and the energy distance (ED), which measure agreement with marginal distributions, preservation of pairwise statistical associations, distinguishability of real versus synthetic samples, and agreement with respect to joint distributions, respectively. Utility was assessed with machine learning efficiency (MLE) metric, which measures utility with respect to performance in downstream prediction tasks by training learners on synthetic data and evaluating their predictive performances on real data (i.e., the holdout set). Privacy was evaluated with the distance to closest record (DCR) and the sorted standard deviation interval distance (SSDID), which capture attribute disclosure risks, as well as the sorted distance-based record linkage (SDBRL), which measures re-identification risks. Further details on all evaluation metrics are provided in Appendix I.5.

**Baselines.** In addition to comparing the MIAV-based synthetic data generation strategy with the joint factorization (JF) and full conditional (FC) approaches, experiments 1 and 2 also included the SMOTE generator (Chawla et al., 2002). SMOTE is well known for producing high-fidelity synthetic data, although this often comes at the expense of privacy when compared to other baselines (Kotelnikov et al., 2023; Kindji et al., 2024). We selected SMOTE as a baseline because, unlike deep learning–based generators, it is applicable to the small datasets used in our evaluations. (As shown in Table 4, 26 out of the 36 datasets contain fewer than 2,000 samples.) In the third experiment, which involves larger datasets, we extended our comparisons to include additional baseline generators: DDPM (Kotelnikov et al., 2023), ARF (Watson et al., 2023), TVAE (Xu et al., 2019), CTGAN (Xu et al., 2019), and Bayesian networks (Young et al., 2009), all implemented in Synthcity (Qian et al., 2024). For DDPM, TVAE, and CTGAN, we adopted the hyperparameter values reported by Hansen et al. (2023) and Chaibub Neto (2025), which had been optimized with Optuna (Akiba et al., 2019) using AUROC minimization of an XGBoost classifier. For ARF, we used the values from Chaibub Neto (2025). The corresponding hyperparameters are listed in Tables 6 and 7. Relying on these published configurations provided substantial computational savings, since hyperparameter optimization is particularly costly for deep learning–based models, and motivated our restriction of these baseline comparisons to only these 7 datasets.

**Experimental details.** In all three experiment sets, we also compared against the holdout sets to establish reference values for each evaluation metric under the ideal scenario of a generator that samples directly from the same distribution as the original data. To enhance statistical validity, we conducted 10 replications for each real-world dataset, each based on distinct original/holdout splits. For the correlated beta distribution experiments, results were similarly averaged over 10 replications per simulation setting, with variation introduced through different simulation parameters. We considered five simulation settings corresponding to absolute correlations $|\rho| = 0, 0.25, 0.5, 0.75, 0.95$. Additional experimental details are provided in Appendix I.

### 7.1 RESULTS

Figure 4 presents results (pooled across datasets) for the KS, L2D, DT, DCR, SDBRL, and SSDID metrics.

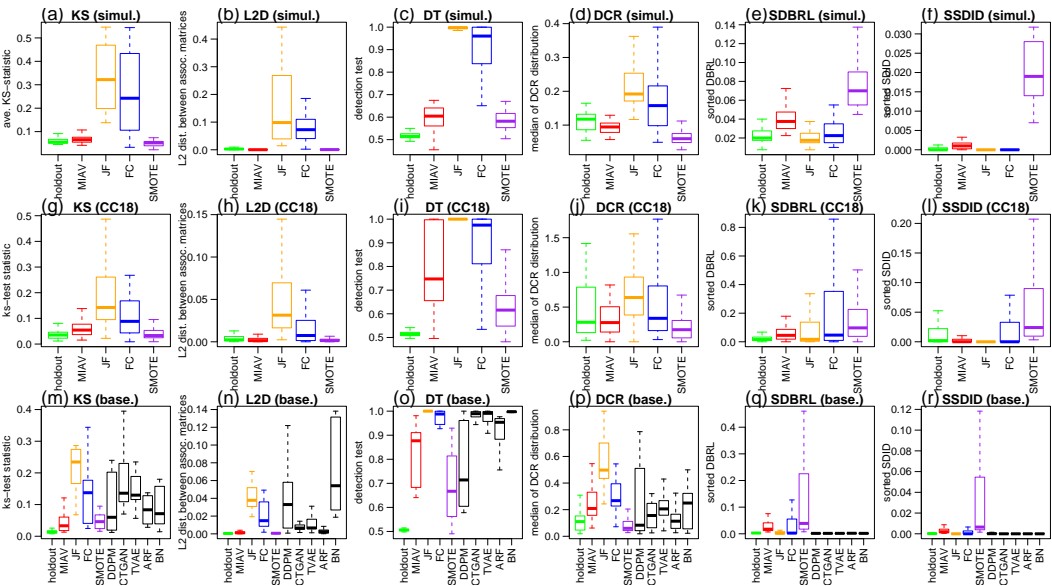

Figure 4: Pooled experimental results. Top panels show results pooled across the 5 simulated dataset settings. The middle panels show results pooled across the 36 real-world datasets selected from the OpenML-CC18 suite. The bottom panels show results pooled across the 7 real-world datasets used for the baseline generator comparisons. For the DCR metric, higher values indicate better privacy. For all other metrics, lower values indicate either better fidelity or better privacy.

In terms of fidelity, SMOTE generally performed best, with a slight advantage over MIAV (recall that lower KS, L2D, and DT values indicate higher fidelity). MIAV, however, consistently outperformed JF and FC across all experiments and surpassed all other baseline generators in the third experiment, with the single exception of DDPM, which achieved better scores w.r.t. the DT metric (see panel o) (but still did worse than MIAV w.r.t. the other fidelity scores in panels m and n).

In terms of privacy, MIAV generally outperformed SMOTE on the DCR metric (where higher values indicate stronger privacy protection) and showed even larger gains on the SDBRL and SSDID metrics (where lower values indicate better privacy). MIAV also tended to surpass the other baseline generators with respect to DCR (panel o), but performed less favorably on SDBRL and SSDID (panels q and r). Across most experiments, JF and FC produced more private data than MIAV, although this came at the cost of substantially lower fidelity.

In general terms, the fidelity–privacy tradeoff patterns observed in the simulated datasets (top panels) closely mirrored those in the real datasets (middle and bottom panels). Overall, these experiments suggest a competitive performance of the MIAV-based synthetic data generator. Additional results broken down by dataset are provided in Figures 14, 15, 16, 17, 18, and 19 in Appendix I.7.

Figure 13 in Appendix I.6 report the pooled results for the MLE, WD, and ED metrics. Overall, these evaluations show that the MIAV approach again achieves competitive performance with respect to these additional metrics (ranking among the top generators in most experiments). See Appendix I.6 for further details.

Appendix J introduces a noisy variant of the MIAV approach that applies controlled amounts of noise before generating synthetic data. This version, referred to as the noisy-MIAV strategy, can potentially enhance privacy protection in sensitive applications, albeit at the cost of reduced data fidelity.

# 8 EXPERIMENTS BASED ON TABICL MODELS

We also demonstrate the MIAV data generation strategy using the TabICL foundation model (Qu et al., 2025) and compare its performance to TabPFN-based generators across 8 categorical datasets from the OpenML-CC18 suite. Results, presented in Appendix K, show that (i) MIAV-TabICL and

MIAV-TabPFN achieve very similar performance, and (ii) MIAV-based strategies, whether built on TabICL or TabPFN, tend to outperform the JF and FC baselines. This alternative implementation highlights that our synthetic data generation approach is not limited to TabPFN and can be directly applied to other PFN-based foundation models.

# 9 FINAL REMARKS

In this paper, we introduce the MIAV strategy, a more effective approach for leveraging TabPFN models in synthetic data generation. MIAV addresses key limitations of direct TabPFN application, offers improved computational efficiency, and can be readily applied with other PFN-based foundation models. We expect it to be especially useful in small-data scenarios, settings that are typically challenging for traditional synthetic data generators but where TabPFN excels. Our experiments on real datasets indicate that MIAV is competitive with established baselines built on bespoke machine learning models. It is worth noting, however, that PFN-based tabular foundation models are still in the early stages of development. As these models continue to evolve, it is reasonable to expect that MIAV-based synthetic data generators built on future, more advanced PFN-based tabular foundation models might achieve even better performance.

Our approach inevitably inherits the limitations of the underlying tabular foundation model used for in-context learning. For the TabPFN model used in our experiments (namely, TabPFNv2), these limitations include: (i) a maximum data size of 10,000 rows; (ii) memory usage that grows linearly with dataset size, which can become prohibitive for very large data; and (iii) inference speeds that may lag behind alternative baselines. But, as mentioned above, these are early days in the development of PFN-based tabular foundation models and we expect that future releases will likely continue to relax limitations from the previous versions. For instance, a new version of the TabPFN model, denoted TabPFN-2.5, has been recently released which is able to handle datasets with up to 50,000 rows (Grinsztajn et al., 2025). Furthermore, the more scalable TabICL model is already able to handle 500,000 rows but currently supports only classification tasks. Future versions of TabICL that extend to regression could be directly integrated with the MIAV strategy, thereby helping to overcome current model constraints.

In addition to limitations on the maximum number of rows they can process, PFN-based tabular foundation models are also constrained by the number of columns they can handle. For example, TabPFNv2 and TabICL support datasets with up to 500 features, while TabPFN-2.5 increases this limit to 2,000. Importantly, however, these constraints do not affect the MIAV approach: because MIAV requires training PFN-based models using only a single feature per variable, it can be applied to datasets containing more columns than the column number limit of the underlying PFN model.

For tabular foundation models that do not approximate Bayesian inference, our approach may still provide a natural strategy for synthetic data generation through in-context learning. Assessing the feasibility of such extensions, however, is left for future work.

Finally, we point out that the MIAV strategy described here is really only intended for synthetic data generation and should not be used for improving predictive performance of supervised learners. As described in Section 6, generating MIAV variables requires unrestricted access to the full dataset $\mathbf{X}$, which is partitioned into $\mathbf{X}^{tr}$ and $\mathbf{X}^{ts}$ and used to construct the corresponding MIAV matrices $\mathbf{M}^{tr}$ and $\mathbf{M}^{ts}$. Because MIAVs must be computed on the test set, the approach is inherently incompatible with supervised learning scenarios where test-set targets are unavailable. But, more importantly, it should never be used to enhance supervised learning performance in settings where the full dataset is merely split into training and test subsets for evaluation purposes. In such cases, the generation of the MIAV variable associated with the test set target would leak information about the target into the associated MIAV, and inclusion of this MIAV as an input in a supervised model would lead to an artificial boost in predictive performance due to data leakage.

R and Python implementations of the MIAV strategy and the code for running the experiments in this paper are available on GitHub: `https://github.com/echaibub/MIAV`.

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

APPENDIX

CONTENTS

## A  Supervised learning versus synthetic data generation in the presence of uninformative features

TabPFN achieves state-of-the-art performance in supervised learning tasks (19). However, its direct application to synthetic data generation does not yield comparable results, as weakly associated variables make it challenging for in-context learning methods to capture the true data distribution. This issue is less pronounced in supervised learning, where noninformative features simply fail to contribute to the prediction. Even when the prediction is based solely on such features, TabPFN behaves appropriately. For instance, in binary classification with predictors completely uncorrelated with the target, it produces an AUROC of about 0.5, consistent with random guessing. Although uninteresting, this outcome is exactly what one would expect from any well-behaved classifier in this scenario. By contrast, in the data generation setting, TabPFN's reasonable performance in classification does not carry over, as it fails to approximate the marginal distributions of the data and to capture its statistical association structure. An illustrative example is presented in Figure 9 in Section D, where direct application of TabPFN to datasets containing completely uncorrelated data fails to recover the marginal distributions (see panels g, h, i, j, and k and m, n, o, p, and q), and fails to recover the correlation structure of the original data (compare panels b and c against panel a).

## B  Algorithms for TabPFN-based synthetic data generation

In Algorithms 2, 4, and 6, the function `GeneratePredictionUsingTabPFN(.)` represents a call to a TabPFN model $q_\theta(.)$.

---

**Algorithm 2** ICLwithJointFactorizationTabPFN($\mathbf{X}^{tr}, \mathbf{X}^{ts}$)

1: **Input:** training data for ICL, $\mathbf{X}^{tr}$; query data for ICL, $\mathbf{X}^{ts}$
2: $n_{tr} \leftarrow$ NumberOfRows($\mathbf{X}^{tr}$) {Obtain number of samples of $\mathbf{X}^{tr}$.}
3: $n_{ts} \leftarrow$ NumberOfRows($\mathbf{X}^{ts}$) {Obtain number of samples of $\mathbf{X}^{ts}$.}
4: $p \leftarrow$ NumberOfColumns($\mathbf{X}^{ts}$) {Obtain number of columns of $\mathbf{X}^{ts}$.}
5: $\mathbf{Z}^{ts} \leftarrow [,]$ {Create empty matrix to store the synthetic data.}
6: $\mathbf{x}_0^{tr} \leftarrow$ GenerateUniformlyDistributedNoise($n_{tr}$) {Draw a sample of size $n_{tr}$ from a uniform distribution.}
7: $\mathbf{x}_0^{ts} \leftarrow$ GenerateUniformlyDistributedNoise($n_{ts}$) {Draw a sample of size $n_{ts}$ from a uniform distribution.}
8: $\mathbf{Z}^{ts}[,1] \leftarrow$ GeneratePredictionUsingTabPFN($\mathbf{x}_0^{ts}, \mathbf{x}_0^{tr}, \mathbf{x}_1^{tr}$) {Predict $\mathbf{x}_1^{ts}$ using $\mathbf{x}_0^{tr}$ and $\mathbf{x}_1^{tr}$ as context, and $\mathbf{x}_0^{ts}$ as query. The prediction can be from a regression or classification TabPFN model, depending on whether $\mathbf{x}_1^{tr}$ is continuous or categorical.}
9: **for** $j = 2$ **to** $p$ **do**
10:     $\mathbf{X}_{<j}^{tr} \leftarrow \mathbf{X}^{tr}[, 1:(j-1)]$ {Select the first $j-1$ columns of $\mathbf{X}^{tr}$.}
11:     $\mathbf{X}_{<j}^{ts} \leftarrow \mathbf{X}^{ts}[, 1:(j-1)]$ {Select the first $j-1$ columns of $\mathbf{X}^{ts}$.}
12:     $\mathbf{Z}^{ts}[,j] \leftarrow$ GeneratePredictionUsingTabPFN($\mathbf{X}_{<j}^{ts}, \mathbf{X}_{<j}^{tr}, \mathbf{x}_j^{tr}$) {Predict $\mathbf{x}_j^{ts}$ using $\mathbf{X}_{<j}^{tr}$ and $\mathbf{x}_j^{tr}$ as context, and $\mathbf{X}_{<j}^{ts}$ as query. The prediction can be from a regression or classification TabPFN model, depending on whether $\mathbf{x}_j^{tr}$ is continuous or categorical.}
13: **end for**
14: **Output:** synthetic data $\mathbf{Z}^{ts}$

---

**Algorithm 3** JointFactorizationTabPFNGenerator($\mathbf{X}$)

1: **Input:** the original data, $\mathbf{X}$
2: $\mathbf{X}_1, \mathbf{X}_2 \leftarrow$ DataSplit($\mathbf{X}$) {Split the original data $\mathbf{X}$ into two subsets, $\mathbf{X}_1$ and $\mathbf{X}_2$.}
3: $\mathbf{Z}_1 \leftarrow$ ICLwithJointFactorizationTabPFN($\mathbf{X}^{tr} = \mathbf{X}_2, \mathbf{X}^{ts} = \mathbf{X}_1$) {Generate a synthetic data copy of $\mathbf{X}_1$ using Algorithm 2.}
4: $\mathbf{Z}_2 \leftarrow$ ICLwithJointFactorizationTabPFN($\mathbf{X}^{tr} = \mathbf{X}_1, \mathbf{X}^{ts} = \mathbf{X}_2$) {Generate a synthetic data copy of $\mathbf{X}_2$ using Algorithm 2.}
5: $\mathbf{Z} \leftarrow$ Concatenate($\mathbf{Z}_1, \mathbf{Z}_2$) {Concatenate the synthetic datasets $\mathbf{Z}_1$ and $\mathbf{Z}_2$.}
6: $\mathbf{Z} \leftarrow$ RoundIntegerVariables($\mathbf{X}, \mathbf{Z}$) {This function uses $\mathbf{X}$ to determine which variables have integer type and round the values of the corresponding variables in $\mathbf{Z}$ to the nearest integer.}
7: **Output:** synthetic data $\mathbf{Z}$

---

---

**Algorithm 4** ICLwithFullConditionalsTabPFN($\mathbf{X}^{tr}$, $\mathbf{X}^{ts}$)

---

1: **Input:** training data for ICL, $\mathbf{X}^{tr}$; query data for ICL, $\mathbf{X}^{ts}$
2: $p \leftarrow$ NumberOfColumns($\mathbf{X}^{ts}$) {Obtain number of columns of $\mathbf{X}^{ts}$.}
3: $\mathbf{Z}^{ts} \leftarrow [,]$ {Create empty matrix to store the synthetic data.}
4: **for** $j = 1$ **to** $p$ **do**
5:     $\mathbf{X}^{tr}_{-j} \leftarrow \mathbf{X}^{tr}[,-j]$ {Drop the $j$-th column of $\mathbf{X}^{tr}$.}
6:     $\mathbf{X}^{ts}_{-j} \leftarrow \mathbf{X}^{ts}[,-j]$ {Drop the $j$-th column of $\mathbf{X}^{ts}$.}
7:     $\mathbf{Z}^{ts}[,j] \leftarrow$ GeneratePredictionUsingTabPFN($\mathbf{X}^{ts}_{-j}, \mathbf{X}^{tr}_{-j}, \mathbf{x}^{tr}_j$) {Predict $\mathbf{x}^{ts}_j$ using $\mathbf{X}^{tr}_{-j}$ and $\mathbf{x}^{tr}_j$ as context, and $\mathbf{X}^{ts}_{-j}$ as query. The prediction can be from a regression or classification TabPFN model, depending on whether $\mathbf{x}^{tr}_j$ is continuous or categorical.}
8: **end for**
9: **Output:** synthetic data $\mathbf{Z}^{ts}$

---

**Algorithm 5** FullConditionalsTabPFNGenerator($\mathbf{X}$)

---

1: **Input:** the original data, $\mathbf{X}$
2: $\mathbf{X}_1, \mathbf{X}_2 \leftarrow$ DataSplit($\mathbf{X}$) {Split the original data $\mathbf{X}$ into two subsets, $\mathbf{X}_1$ and $\mathbf{X}_2$.}
3: $\mathbf{Z}_1 \leftarrow$ ICLwithFullConditionalsTabPFN($\mathbf{X}^{tr} = \mathbf{X}_2, \mathbf{X}^{ts} = \mathbf{X}_1$) {Generate a synthetic data copy of $\mathbf{X}_1$ using Algorithm 4.}
4: $\mathbf{Z}_2 \leftarrow$ ICLwithFullConditionalsTabPFN($\mathbf{X}^{tr} = \mathbf{X}_1, \mathbf{X}^{ts} = \mathbf{X}_2$) {Generate a synthetic data copy of $\mathbf{X}_2$ using Algorithm 4.}
5: $\mathbf{Z} \leftarrow$ Concatenate($\mathbf{Z}_1, \mathbf{Z}_2$) {Concatenate the synthetic datasets $\mathbf{Z}_1$ and $\mathbf{Z}_2$.}
6: $\mathbf{Z} \leftarrow$ RoundIntegerVariables($\mathbf{X}, \mathbf{Z}$)
7: **Output:** synthetic data $\mathbf{Z}$

---

**Algorithm 6** ICLwithMIAVTabPFN($\mathbf{X}^{tr}$, $\mathbf{X}^{ts}$)

---

1: **Input:** training data for ICL, $\mathbf{X}^{tr}$; query data for ICL, $\mathbf{X}^{ts}$
2: $p \leftarrow$ NumberOfColumns($\mathbf{X}^{ts}$) {Obtain number of columns of $\mathbf{X}^{ts}$.}
3: $\mathbf{Z}^{ts} \leftarrow [,]$ {Create empty matrix to store the synthetic data.}
4: **for** $j = 1$ **to** $p$ **do**
5:     $\mathbf{m}^{tr}_j \leftarrow$ GenerateMaximalInformationAuxiliaryVariable($\mathbf{x}^{tr}_j$) {Generate the MIAV for $\mathbf{x}^{tr}_j$ using Algorithm 1.}
6:     $\mathbf{m}^{ts}_j \leftarrow$ GenerateMaximalInformationAuxiliaryVariable($\mathbf{x}^{ts}_j$) {Generate the MIAV for $\mathbf{x}^{ts}_j$ using Algorithm 1.}
7:     $\mathbf{Z}^{ts}[,j] \leftarrow$ GeneratePredictionUsingTabPFN($\mathbf{m}^{ts}_j, \mathbf{m}^{tr}_j, \mathbf{x}^{tr}_j$) {Predict $\mathbf{x}^{ts}_j$ using $\mathbf{m}^{tr}_j$ and $\mathbf{x}^{tr}_j$ as context, and $\mathbf{m}^{ts}_j$ as query. The prediction can be from a regression or classification TabPFN model, depending on whether $\mathbf{x}^{tr}_j$ is continuous or categorical.}
8: **end for**
9: **Output:** synthetic data $\mathbf{Z}^{ts}$

---

**Algorithm 7** MIAVTabPFNGenerator($\mathbf{X}$)

---

1: **Input:** the original data, $\mathbf{X}$
2: $\mathbf{X}_1, \mathbf{X}_2 \leftarrow$ DataSplit($\mathbf{X}$) {Split the original data $\mathbf{X}$ into two subsets, $\mathbf{X}_1$ and $\mathbf{X}_2$.}
3: $\mathbf{Z}_1 \leftarrow$ ICLwithMIAVTabPFN($\mathbf{X}^{tr} = \mathbf{X}_2, \mathbf{X}^{ts} = \mathbf{X}_1$) {Generate a synthetic data copy of $\mathbf{X}_1$ using Alg. 6.}
4: $\mathbf{Z}_2 \leftarrow$ ICLwithMIAVTabPFN($\mathbf{X}^{tr} = \mathbf{X}_1, \mathbf{X}^{ts} = \mathbf{X}_2$) {Generate a synthetic data copy of $\mathbf{X}_2$ using Alg. 6.}
5: $\mathbf{Z} \leftarrow$ Concatenate($\mathbf{Z}_1, \mathbf{Z}_2$) {Concatenate the synthetic datasets $\mathbf{Z}_1$ and $\mathbf{Z}_2$.}
6: $\mathbf{Z} \leftarrow$ RoundIntegerVariables($\mathbf{X}, \mathbf{Z}$) {This function uses $\mathbf{X}$ to determine which variables have integer type and round the values of the corresponding variables in $\mathbf{Z}$ to the nearest integer.}
7: **Output:** synthetic data $\mathbf{Z}$

---

In Algorithms 3, 5, and 7 the function `RoundIntegerVariables`($\mathbf{X}, \mathbf{Z}$) uses the original data, $\mathbf{X}$, to determine which variables have integer type and round the values of the corresponding variables in the synthetic data, $\mathbf{Z}$, to the nearest integer. This post-processing step is necessary because TabPFN (and most other data synthesizers) return real values for variables that are originally of integer type. In our experiments, we apply the same post-processing step to the SMOTE baseline.

## C   SIMULATING CORRELATED BETA DISTRIBUTIONS

Here, we describe how we simulated correlated beta distributions used in the simulated data experiments, as well as, for the illustrative examples provided in Figures 1, 2, and 3 in the main text, and Figures 5, 6, 7, 8, and 9 in Appendix D.

The data was simulated as follows. First, we simulate data from a multivariate normal random variable $Y \sim N_p(\mathbf{0}, \mathbf{\Sigma})$. Next, for $j = 1, \ldots, p$, we compute the correlated uniform variables $U_j = \Phi(Z_j)$, and the correlated beta random variables $X_j = G_{a,b}^{-1}(U_j)$, where $\Phi$ and $G_{a,b}$ represent, respectively, the cumulative distribution functions of standard normal variable and a beta variable with shape parameters $a$ and $b$.

The multivariate Gausssian variable $Y$ is generated with a Toeplitz structured covariance matrix $\mathbf{\Sigma}$ with off-diagonal entries $\sigma_{ij} = \rho^{|i-j|}$, and diagonal entries $\sigma_{jj} = 1$, for $\rho \in [-1, 1]$. (Note that under this correlation structure, neighboring variables are more highly associated than more distant variables, and the association decreases the farther apart the variables are. Also, for negative $\rho$ values the direction of the association flips depending on whether the exponent $|i - j|$ is even or odd.)

For the illustrations presented in Figures 1, 2, and 3 we further randomly shuffle the data of variable $X_2$ in order to simulate an uninformative feature uncorrelated with all other variables in the dataset. Furthermore, for the illustrations presented in Figure 3, we also discretize variable $X_5$ into four classes.

## D   SUPPLEMENTARY FIGURES

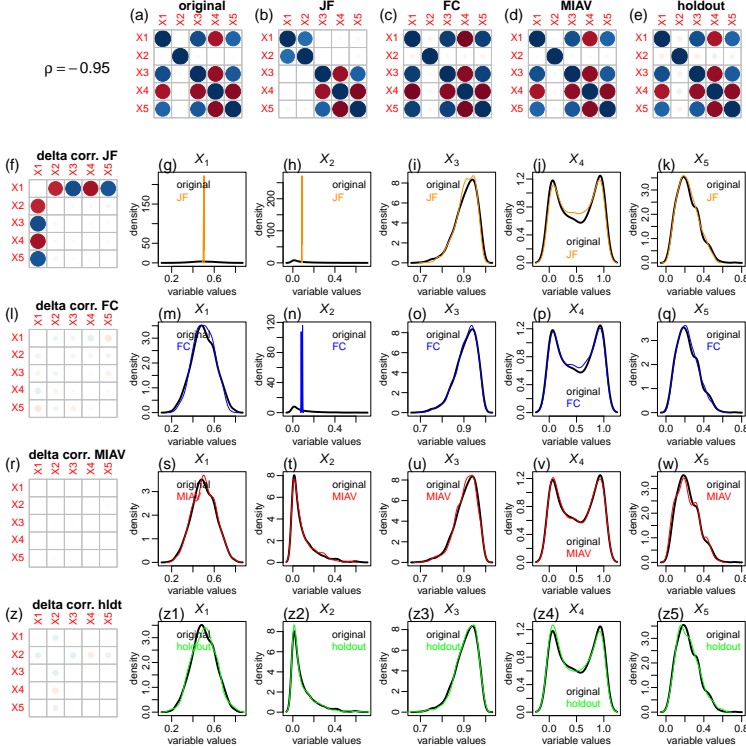

Figure 5: Comparison of the JF, FC, and MIAV synthetic data generation strategies for correlated beta variables generated with $\rho = -0.95$. Panel a shows the Pearson correlation matrices for the original data, while panels b to e show the correlation matrices for the JF, FC, MIAV synthetic datasets and the holdout set (positive correlations are represented in blue and negative correlations are represented in red). Panels f, l, r, and z show the difference between the original data correlation matrix and the respective synthetic datasets and holdout set. The remaining panels show the marginal distributions generated by the distinct synthetic data generation approaches and for the holdout set.

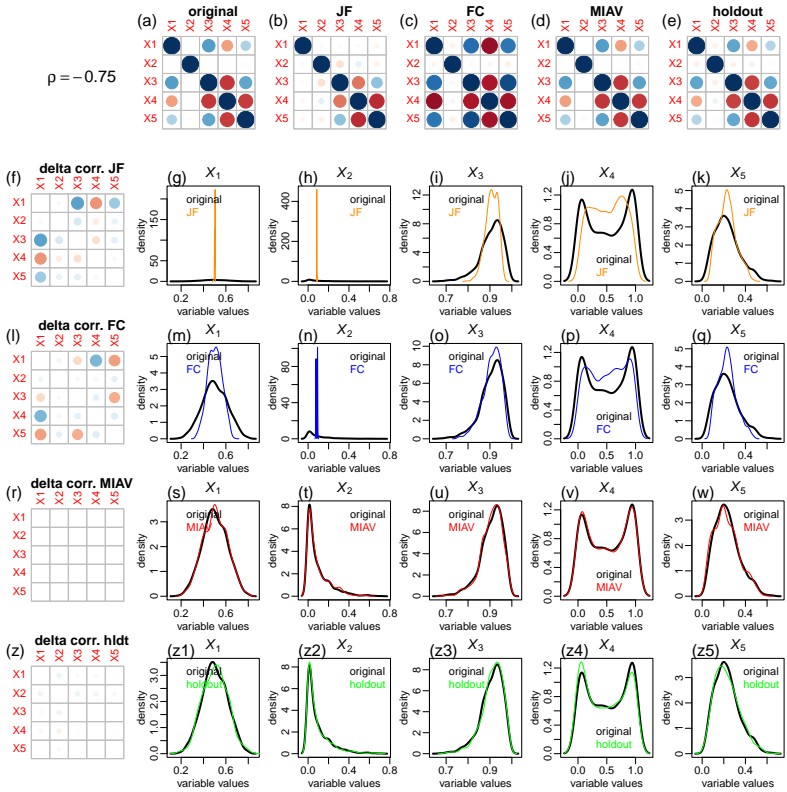

Figure 6: Analogous comparisons as in Figure 5, but for data simulated with $\rho = -0.75$.

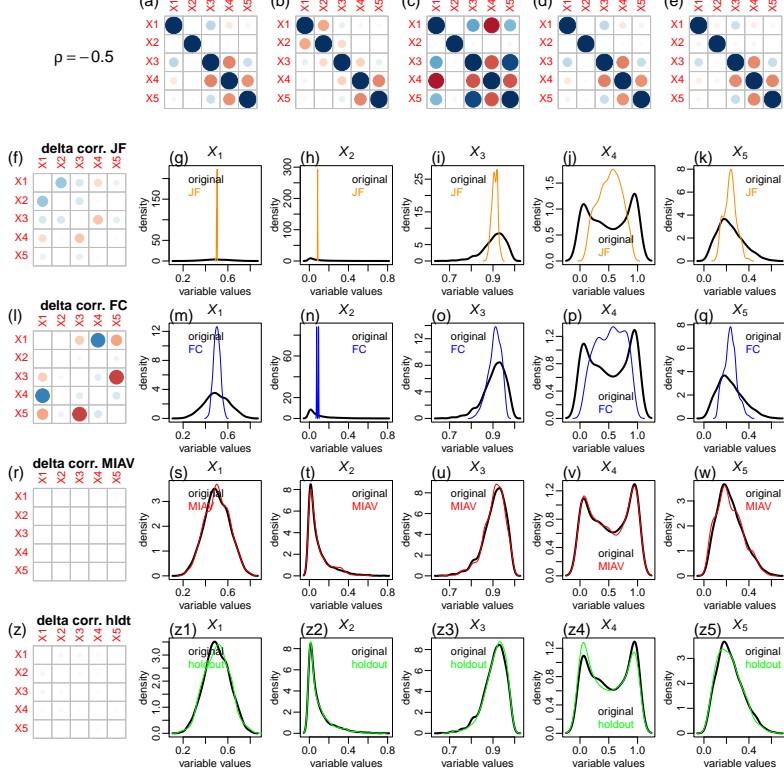

Figure 7: Analogous comparisons as in Figure 5, but for data simulated with $\rho = -0.5$.

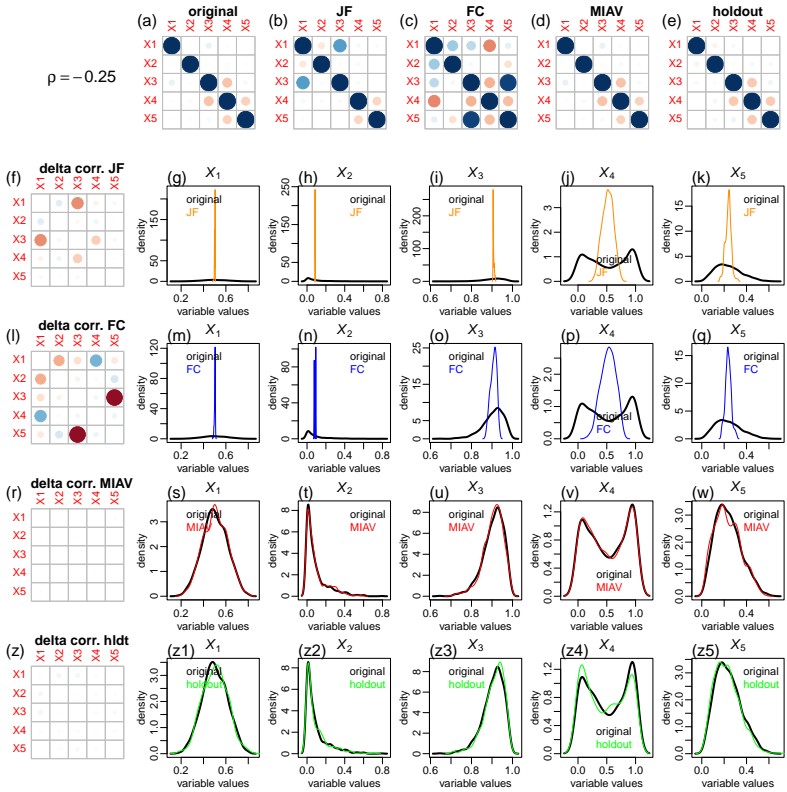

Figure 8: Analogous comparisons as in Figure 5, but for data simulated with $\rho = -0.25$.

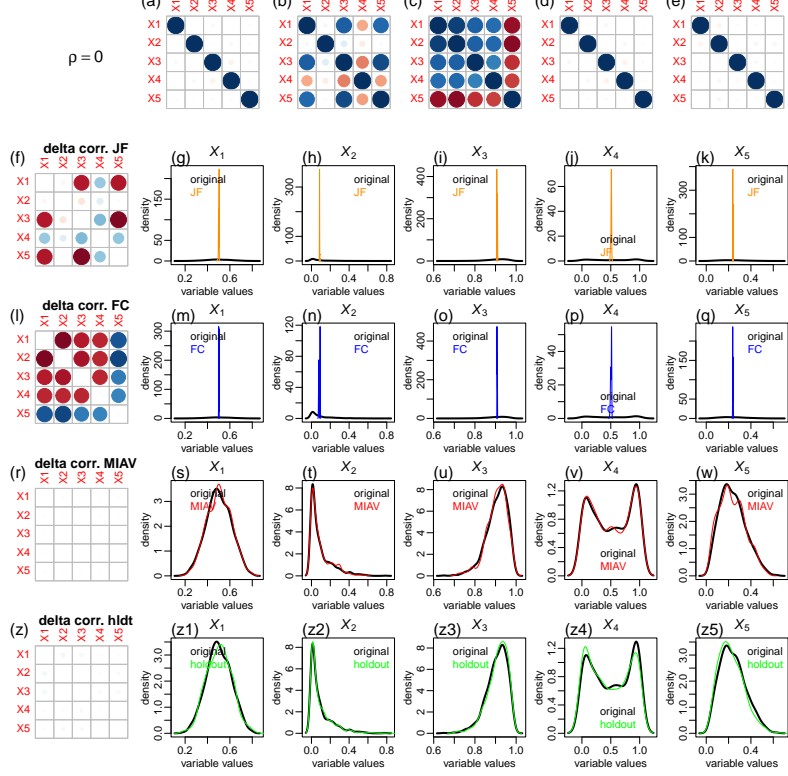

Figure 9: Analogous comparisons as in Figure 5, but for data simulated with $\rho = 0$.

## E  ADDITIONAL DIRECT SYNTHETIC DATA GENERATION STRATEGIES

As described in Algorithms 2 and 4 our implementations of the JF and FC strategies use the real test data when performing the in-context queries. In the case of the JF strategy, an alternative approach would be to use the synthetic data generated in the previous steps when we query the model in the current generation step. That is, instead of generating the data according to equation 2 we generate it according to,

$$\prod_{j=1}^{p} q_\theta(\mathbf{x}_j^{ts} \mid \hat{\mathbf{X}}_{<j}^{ts}, \mathbf{X}_{<j}^{tr}, \mathbf{x}_j^{tr}) \,, \tag{10}$$

where $\hat{\mathbf{X}}_{<j}^{ts}$ represents the predictions (i.e., the synthetic data) generated in the previous $j-1$ generation steps. We denote this strategy as the "updated joint factorization" (or UJF for short), and implement it in Algorithms 8 and 9. Note that Algorithm 8 implements two versions of the UJF approach. The first, simply takes a bootstrap sample from the original $X_1$ data as the "synthetic version" of $X_1$. The second, uses the random noise variable $X_0$ to generate $X_1$ (similarly to the JF strategy). Intuitively, we would expect UJF to under-perform in comparison with the JF strategy, since any drifts in the distribution of the synthetic data generated in the previous steps would propagate to the later ones.

---

**Algorithm 8** ICLwithUpdatedJointFactorizationTabPFN($\mathbf{X}^{tr}$, $\mathbf{X}^{ts}$, $version = 1$)

1: **Input:** training data for ICL, $\mathbf{X}^{tr}$; query data for ICL, $\mathbf{X}^{ts}$; $version$ of the UJF strategy
2: $n_{tr} \leftarrow$ NumberOfRows($\mathbf{X}^{tr}$) {Obtain number of samples of $\mathbf{X}^{tr}$.}
3: $n_{ts} \leftarrow$ NumberOfRows($\mathbf{X}^{ts}$) {Obtain number of samples of $\mathbf{X}^{ts}$.}
4: $p \leftarrow$ NumberOfColumns($\mathbf{X}^{ts}$) {Obtain number of columns of $\mathbf{X}^{ts}$.}
5: $\mathbf{Z}^{ts} \leftarrow [,]$ {Create empty matrix to store the synthetic data.}
6: **if** version $== 1$ **then**
7:     $\mathbf{Z}^{ts}[,1] \leftarrow$ BootstrapSample($\mathbf{x}_1^{ts}$)
8: **end if**
9: **if** version $== 2$ **then**
10:     $\mathbf{x}_0^{tr} \leftarrow$ GenerateUniformlyDistributedNoise($n_{tr}$) {Draw a sample of size $n_{tr}$ from a uniform distribution.}
11:     $\mathbf{x}_0^{ts} \leftarrow$ GenerateUniformlyDistributedNoise($n_{ts}$) {Draw a sample of size $n_{ts}$ from a uniform distribution.}
12:     $\mathbf{Z}^{ts}[,1] \leftarrow$ GeneratePredictionUsingTabPFN($\mathbf{x}_0^{ts}, \mathbf{x}_0^{tr}, \mathbf{x}_1^{tr}$) {Predict $\mathbf{x}_1^{ts}$ using $\mathbf{x}_0^{tr}$ and $\mathbf{x}_1^{tr}$ as context, and $\mathbf{x}_0^{ts}$ as query. The prediction can be from a regression or classification TabPFN model, depending on whether $\mathbf{x}_1^{tr}$ is continuous or categorical.}
13: **end if**
14: **for** $j = 2$ **to** $p$ **do**
15:     $\mathbf{X}_{<j}^{tr} \leftarrow \mathbf{X}^{tr}[,1:(j-1)]$ {Select the first $j-1$ columns of $\mathbf{X}^{tr}$.}
16:     $\mathbf{Z}_{<j}^{ts} \leftarrow \mathbf{Z}^{ts}[,1:(j-1)]$ {Select the first $j-1$ columns of $\mathbf{Z}^{ts}$.}
17:     $\mathbf{Z}^{ts}[,j] \leftarrow$ GeneratePredictionUsingTabPFN($\mathbf{Z}_{<j}^{ts}, \mathbf{X}_{<j}^{tr}, \mathbf{x}_j^{tr}$) {Predict $\mathbf{x}_j^{ts}$ using $\mathbf{X}_{<j}^{tr}$ and $\mathbf{x}_j^{tr}$ as context, and $\mathbf{Z}_{<j}^{ts}$ as query. The prediction can be from a regression or classification TabPFN model, depending on whether $\mathbf{x}_j^{tr}$ is continuous or categorical.}
18: **end for**
19: **Output:** synthetic data $\mathbf{Z}^{ts}$

---

**Algorithm 9** UpdatedJointFactorizationTabPFNGenerator($\mathbf{X}$)

1: **Input:** the original data, $\mathbf{X}$
2: $\mathbf{X}_1, \mathbf{X}_2 \leftarrow$ DataSplit($\mathbf{X}$) {Split the original data $\mathbf{X}$ into two subsets, $\mathbf{X}_1$ and $\mathbf{X}_2$.}
3: $\mathbf{Z}_1 \leftarrow$ ICLwithUpdatedJointFactorizationTabPFN($\mathbf{X}^{tr} = \mathbf{X}_2, \mathbf{X}^{ts} = \mathbf{X}_1$) {Generate a synthetic data copy of $\mathbf{X}_1$ using Algorithm 8.}
4: $\mathbf{Z}_2 \leftarrow$ ICLwithUpdatedJointFactorizationTabPFN($\mathbf{X}^{tr} = \mathbf{X}_1, \mathbf{X}^{ts} = \mathbf{X}_2$) {Generate a synthetic data copy of $\mathbf{X}_2$ using Algorithm 8.}
5: $\mathbf{Z} \leftarrow$ Concatenate($\mathbf{Z}_1, \mathbf{Z}_2$) {Concatenate the synthetic datasets $\mathbf{Z}_1$ and $\mathbf{Z}_2$.}
6: **Output:** synthetic data $\mathbf{Z}$

---

This intuition is confirmed in Figure 10, which shows the application of version 1 of the UJF strategy in purple (panels f to j) and of version 2 (panels l to p) in cyan. For comparison, the plot also include

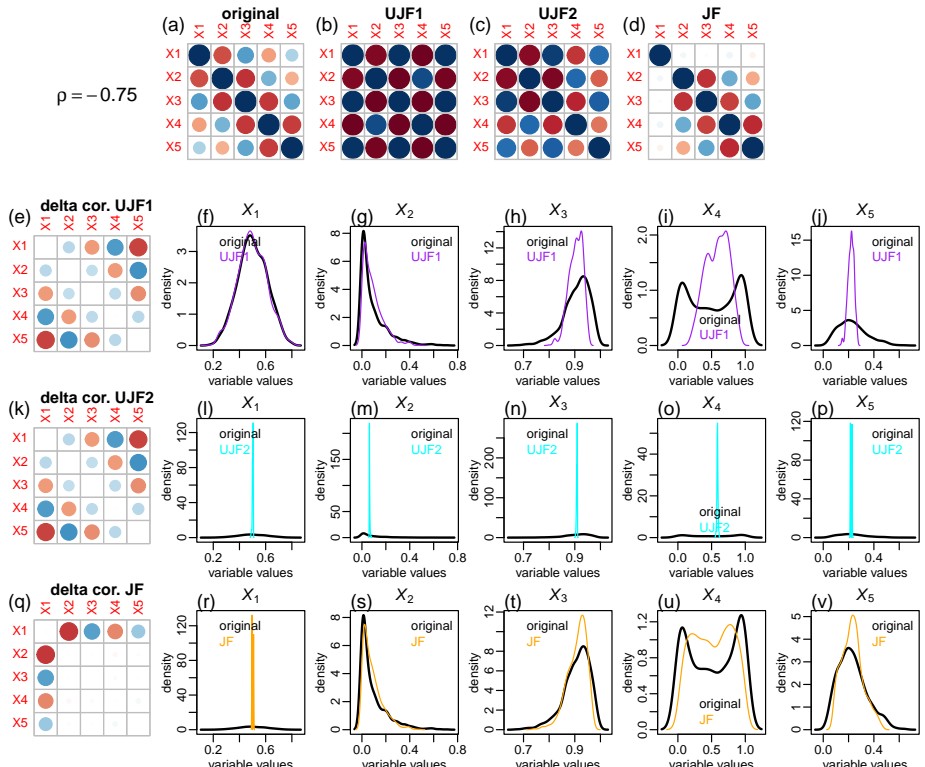

Figure 10: Comparison of JF and the UJF approaches.

the results from the JF approach (panels r to v). Even in version 1, where the algorithm samples the first variable from the correct distribution, the distributions of the synthetic data still ends drifting away from the original data distributions at later steps of the data generation approach. (Note that UJF1 performs considerably worse than the JF approach for variables $X_3$, $X_4$, and $X_5$.) In version 2 (panels l to p) the performance is even worse because the quality of the $X_1$ data generated in the first step is already very poor.

## F  ADDITIONAL ALGORITHMS

---
**Algorithm 10** NumericRankEncodingOfCategoticalVariables($\mathbf{x}_j$)

---
1: **Input:** categorical variable data $\mathbf{x}_j$
2: $levels \leftarrow$ ExtractLevels($\mathbf{x}_j$) {Obtain levels of variable $\mathbf{x}_j$.}
3: $tb \leftarrow$ Table($\mathbf{x}_j[,i]$) {Obtain level counts of variable $\mathbf{x}_j$.}
4: $cumcounts \leftarrow$ CumulativeSum($tb$) {Compute cumulative counts from the table counts.}
5: $numlevels \leftarrow$ Length($levels$) {Obtain number of levels.}
6: $\mathbf{r} \leftarrow []$ {Create empty vector to store the numeric rank encodings.}
7: **for** $k = 1$ **to** $numlevels$ **do**
8:     $idx \leftarrow$ Which($\mathbf{x}_j == levels[k]$) {Obtain the indexes of the records for which $\mathbf{x}_j$ equals level $k$.}
9:     $lowerbound \leftarrow cumcounts[k] + 1$ {Compute the lower bound for the numerical encoding of level $k$ of $\mathbf{x}_j$.}
10:     $upperbound \leftarrow cumcounts[k + 1]$ {Compute the upper bound for the numerical encoding of level $k$ of $\mathbf{x}_j$.}
11:     $nrseq \leftarrow$ Sequence($lowerbound, upperbound$) {Create a sequence of numeric rank values starting at lowerbound and ending at upper-bound.}
12:     $\mathbf{r}[idx] \leftarrow$ RandomPermutation($nrseq$) {Assign randomly shuffled numeric rank values to the positions of $\mathbf{x}_j$ corresponding to level $k$.}
13: **end for**
14: **Output:** numerical rank encoding $\mathbf{r}$

---

As an example, suppose $\mathbf{x}_j = (A, A, A, A, B, B, B, C, C, C, C, C, D, D, D)^t$. This categorical variable has four levels $A$, $B$, $C$ and $D$ with counts $n_A = 4$, $n_B = 3$, $n_C = 5$ and $n_D = 3$. To generate the numerical rank encoding $\mathbf{r}$ presented in Table 1, the algorithm transforms the categorical values according to the mapping, $map_r$, in Table 1. Note that this mapping corresponds to an arbitrary ranking of the categorical levels according to the arbitrary order $A < B < C < D$. That is, starting with class $A$, the mapping assigns ranks 1, 2, 3, and 4 (in random order) to the four tied elements $A$ in $\mathbf{x}_j$. (Note that by assigning the ranks in random order the mapping is effectively using the "at random" method to break ties among identical values of the categorical variable.) For class $B$, the mapping assigns ranks 5, 6, and 7 (in random order) to the three tied elements $B$ in $\mathbf{x}_j$. For class $C$ it assigns ranks 8 to 12 (in random order) to the five tied elements $C$. Finally, for class $D$, the mapping assigns ranks 13, 14, and 15 (in random order) to the three tied elements $D$ in $\mathbf{x}_j$.

Table 1: Toy example illustrating the application of the numeric rank encoding procedure for categorical variables implemented by Algorithm 10 to $\mathbf{x}_j = (A, A, A, A, B, B, B, C, C, C, C, C, D, D, D)^t$.

| $map_r$ | $A = \{1, 2, 3, 4\},$ | | | | $B = \{5, 6, 7\},$ | | | $C = \{8, 9, 10, 11, 12\}$ | | | | | $D = \{13, 14, 15\}$ | | |
|---|---|---|---|---|---|---|---|---|---|---|---|---|---|---|---|
| $\mathbf{x}_j$ | $A$ | $A$ | $A$ | $A$ | $B$ | $B$ | $B$ | $C$ | $C$ | $C$ | $C$ | $C$ | $D$ | $D$ | $D$ |
| $\mathbf{r}$ | 3 | 1 | 4 | 2 | 7 | 5 | 6 | 9 | 8 | 11 | 12 | 10 | 14 | 15 | 13 |

## G    PROOF OF THEOREM 1

Consider variables $X_j$, $Y$, and $M_j$, where $Y$ is a placeholder notation for any variable other than $X_j$ or $M_j$ ($Y$ could, for example, be any other $X_{j'}$ or $M_{j'}$ for $j' \neq j$). Variables $X_j$ and $Y$ might be continuous or categorical. Variable $M_j$ is always continuous by construction. To prove the result in a non-parametric setting, we treat any continuous variable as a categorical variable with $n$ distinct levels (where $n$ represents the number of samples in the data). We also assume the independent samples setting, where the rows of dataset $\mathbf{X}$ are independent.

### G.1    PROOF OF STATEMENT 1, WHEN $X_j$ AND $Y$ ARE CONTINUOUS

*Proof.* We consider first the case where $X_j$ and $Y$ are originally continuous variables. In this case, all variables are treated as categorical variables with $n$ levels, where $i = 1, \ldots, n$, $l = 1, \ldots, n$, and $k = 1, \ldots, n$ indexes the categorical levels of variables $X_j$, $Y$, and $M_j$, respectively.

The conditional mutual information (CMI) of $X_j$ and $Y$ given $M_j$ is defined as,

$$
\begin{aligned}
I(X_j; Y|M_j) = {} & \\
= {} & \sum_{i=1}^{n} \sum_{l=1}^{n} \sum_{k=1}^{n} P(X_j = i, Y = l, M_j = k) \log \left( \frac{P(X_j = i, Y = l \mid M_j = k)}{P(X_j = i \mid M_j = k) \, P(Y = l \mid M_j = k)} \right) \\
= {} & \sum_{i=1}^{n} \sum_{l=1}^{n} \sum_{k=1}^{n} P(X_j = i, Y = l, M_j = k) \log \left( \frac{P(X_j = i, Y = l, M_j = k) \, P(M_j = k)}{P(X_j = i, M_j = k) \, P(Y = l, M_j = k)} \right) ,
\end{aligned}
\tag{11}
$$

where the joint probabilities are defined from the joint frequency counts in a finite population. For instance, $P(X_j = i, Y = l, M_j = k) = f(X_j = i, Y = l, M_j = k)/n$ where $f(X_j = i, Y = l, M_j = k)$ corresponds to the number of instances in a population of size $n$ for which $X_j = i$ and $Y = l$ and $M_j = k$. Note that because all three variables have $n$ distinct levels, these frequencies will be either 1 or 0 depending on whether the combination of values is observed or not in the dataset.

To avoid division by 0, we consider the smoothed version of the CMI where an arbitrarily small constant $\epsilon$ is added to the frequency counts of each cell in the contingency tables defining these probabilities. Namely,

$$
f^\epsilon(X_j = i, Y = l, M_j = k) = f(X_j = i, Y = l, M_j = k) + \epsilon,
\tag{12}
$$

and $P^\epsilon(X_j = i, Y = l, M_j = k)$ is obtained by normalization as,

$$
\begin{aligned}
P^\epsilon(X_j = i, Y = l, M_j = k) &= \frac{f(X_j = i, Y = l, M_j = k) + \epsilon}{\sum_{i=1}^n \sum_{l=1}^n \sum_{k=1}^n (f(X_j = i, Y = l, M_j = k) + \epsilon)} \\
&= \frac{f(X_j = i, Y = l, M_j = k) + \epsilon}{\sum_{i=1}^n \sum_{l=1}^n \sum_{k=1}^n f(X_j = i, Y = l, M_j = k) + \sum_{i=1}^n \sum_{l=1}^n \sum_{k=1}^n \epsilon} \\
&= \frac{f(X_j = i, Y = l, M_j = k) + \epsilon}{n + n^3\,\epsilon} \; .
\end{aligned}
\tag{13}
$$

Similarly, we have that,

$$
P^\epsilon(X_j = i, M_j = k) = \frac{f(X_j = i, M_j = k) + \epsilon}{n + n^2\,\epsilon} \; ,
\tag{14}
$$

$$
P^\epsilon(Y = l, M_j = k) = \frac{f(Y = l, M_j = k) + \epsilon}{n + n^2\,\epsilon} \; ,
\tag{15}
$$

$$
P^\epsilon(M_j = k) = \frac{f(M_j = k) + \epsilon}{n + n\,\epsilon} = \frac{1 + \epsilon}{n(1 + \epsilon)} = \frac{1}{n} \; .
\tag{16}
$$

Now, because the rank-matching procedure used in the construction of $M_j$ implies a perfect monotonic relation between the values of $X_j$ and $M_j$ we have that the discretized versions of $X_j$ and $M_j$ will be identical (see Table 2 for an illustrative toy example). This correspondence between $X_j$ and $M_j$ implies that these variables cannot concomitantly assume different values at the same time (i.e., the joint frequency $f(X_j = i, M_j = k)$ will always be 0 if $i \neq k$). It also implies that the joint counts $f(X_j = i, M_j = k)$ will always be equal to the marginal counts $f(M_j = k)$ when $i = k$. Hence, we have that,

$$
f(X_j = i, M_j = k) = \begin{cases} f(M_j = k) = 1 \,, & \text{if } i = k \\ 0 \,, & \text{if } i \neq k \end{cases} ,
\tag{17}
$$

and we can re-express equation (14) as,

$$
P^\epsilon(X_j = i, M_j = k) = \begin{cases} \dfrac{f(M_j = k) + \epsilon}{n + n^2\epsilon} = \dfrac{1 + \epsilon}{n + n^2\epsilon} \,, & \text{if } i = k \\[2ex] \dfrac{\epsilon}{n + n^2\,\epsilon} \,, & \text{if } i \neq k \end{cases} .
\tag{18}
$$

Similarly, we have that,

$$
f(X_j = i, Y = l, M_j = k) = \begin{cases} f(Y = l, M_j = k) \,, & \text{if } i = k \\ 0 \,, & \text{if } i \neq k \end{cases} ,
\tag{19}
$$

and we can re-express equation (13) as,

$$
P^\epsilon(X_j = i, Y = l, M_j = k) = \begin{cases} \dfrac{f(Y = l, M_j = k) + \epsilon}{n + n^3\epsilon} \,, & \text{if } i = k \\[2ex] \dfrac{\epsilon}{n + n^3\,\epsilon} \,, & \text{if } i \neq k \end{cases} .
\tag{20}
$$

Re-expressing the conditional mutual information in terms of the smoothed probabilities and separating the summation over the cases where $k = i$ from the cases where $k \neq i$ we have,

$$I^\epsilon(X_j; Y|M_j) = \tag{21}$$

$$= \sum_{i=1}^{n} \sum_{l=1}^{n} P^\epsilon(X_j = i, Y = l, M_j = i) \log \left( \frac{P^\epsilon(X_j = i, Y = l, M_j = i) P^\epsilon(M_j = i)}{P^\epsilon(X_j = i, M_j = i) P^\epsilon(Y = l, M_j = i)} \right) +$$

$$+ \sum_{i=1}^{n} \sum_{l=1}^{n} \sum_{k \neq i} P^\epsilon(X_j = i, Y = l, M_j = k) \log \left( \frac{P^\epsilon(X_j = i, Y = l, M_j = k) P^\epsilon(M_j = k)}{P^\epsilon(X_j = i, M_j = k) P^\epsilon(Y = l, M_j = k)} \right)$$

$$= \sum_{i=1}^{n} \sum_{l=1}^{n} \frac{f(Y = l, M_j = i) + \epsilon}{n + n^3 \epsilon} \log \left( \frac{\left( \frac{f(Y = l, M_j = i) + \epsilon}{n + n^3 \epsilon} \right) \left( \frac{f(M_j = i) + \epsilon}{n + n \epsilon} \right)}{\left( \frac{f(M_j = i) + \epsilon}{n + n^2 \epsilon} \right) \left( \frac{f(Y = l, M_j = i) + \epsilon}{n + n^2 \epsilon} \right)} \right) +$$

$$+ \sum_{i=1}^{n} \sum_{l=1}^{n} \sum_{k \neq i} \frac{\epsilon}{n + n^3 \epsilon} \log \left( \frac{\left( \frac{\epsilon}{n + n^3 \epsilon} \right) \left( \frac{f(M_j = k) + \epsilon}{n + n \epsilon} \right)}{\left( \frac{\epsilon}{n + n^2 \epsilon} \right) \left( \frac{f(Y = l, M_j = k) + \epsilon}{n + n^2 \epsilon} \right)} \right)$$

$$= \sum_{i=1}^{n} \sum_{l=1}^{n} \frac{f(Y = l, M_j = i) + \epsilon}{n + n^3 \epsilon} \log \left( \frac{(n + n^2 \epsilon)(n + n^2 \epsilon)}{(n + n^3 \epsilon)(n + n \epsilon)} \right) +$$

$$+ \sum_{i=1}^{n} \sum_{l=1}^{n} \sum_{k \neq i} \frac{\epsilon}{n + n^3 \epsilon} \log \left( \frac{(n + n^2 \epsilon)(n + n^2 \epsilon)(f(M_j = k) + \epsilon)}{(n + n^3 \epsilon)(n + n \epsilon)(f(Y = l, M_j = k) + \epsilon)} \right)$$

Taking the limit as $\epsilon \to 0$ we have that $I^\epsilon(X_j; Y|M_j)$ converges to,

$$\sum_{i=1}^{n} \sum_{l=1}^{n} \frac{f(Y = l, M_j = i)}{n} \log(1) + \sum_{i=1}^{n} \sum_{l=1}^{n} \sum_{k \neq i} 0 \log \left( \frac{f(M_j = k)}{f(Y = l, M_j = k)} \right) = 0 \,. \tag{22}$$

Observe that the term $0 \log(f(M_j = k)/f(Y = l, M_j = k))$ in the above equation equals 0 because it can be rewritten as $0 \log f(M_j = k) - 0 \log f(Y = l, M_j = k)$ and $0 \log f(M_j = k) = 0 \log(1/n) = 0$ and $0 \log f(Y = l, M_j = k)$ equals $0 \log 1 = 0$ if $l = k$ or $0 \log 0 = 0$ if $l \neq k$. (Note that in this last case, we are adopting the convention that $0 \log 0 = 0$, as customary in information theory.)

$\square$

### G.2 Proof of Statement 1, when $X_j$ and $Y$ are categorical

*Proof.* We now consider the case where $X_j$ and $Y$ are categorical variables. Let $b_x$ and $b_y$ represent the number of levels of $X_j$ and $Y$, and let $i = 1, \ldots, b_x$, $l = 1, \ldots, b_y$, and $k = 1, \ldots, n$ represent the indexes of the levels of variables $X_j$, $Y$, and $M_j$, respectively.

Now, the CMI of $X_j$ and $Y$ conditional of $M_j$ is given by,

$$I(X_j; Y|M_j) =$$

$$= \sum_{i=1}^{b_x} \sum_{l=1}^{b_y} \sum_{k=1}^{n} P(X_j = i, Y = l, M_j = k) \log \left( \frac{P(X_j = i, Y = l \mid M_j = k)}{P(X_j = i \mid M_j = k) P(Y = l \mid M_j = k)} \right)$$

$$= \sum_{i=1}^{b_x} \sum_{l=1}^{b_y} \sum_{k=1}^{n} P(X_j = i, Y = l, M_j = k) \log \left( \frac{P(X_j = i, Y = l, M_j = k) P(Z_j = k)}{P(X_j = i, M_j = k) P(Y = l, M_j = k)} \right) \,.$$

$$\tag{23}$$

To avoid division by 0, we again consider the smoothed version of the CMI with smoothed probabilities given by,

$$P^\epsilon(X_j = i, Y = l, M_j = k) = \frac{f(X_j = i, Y = l, M_j = k) + \epsilon}{n + b_x\, b_y\, n\, \epsilon}\ . \tag{24}$$

$$P^\epsilon(X_j = i, M_j = k) = \frac{f(X_j = i, M_j = k) + \epsilon}{n + b_x\, n\, \epsilon}\ , \tag{25}$$

$$P^\epsilon(Y = l, M_j = k) = \frac{f(Y = l, M_j = k) + \epsilon}{n + b_y\, n\, \epsilon}\ , \tag{26}$$

$$P^\epsilon(M_j = k) = \frac{f(M_j = k) + \epsilon}{n + n\, \epsilon} = \frac{1 + \epsilon}{n(1 + \epsilon)} = \frac{1}{n}\ . \tag{27}$$

By construction, when $X_j$ is categorical, we have that the following relations hold between $X_j$ and the discretized $M_j$,

$$f(X_j = i, M_j = k) = \begin{cases} f(M_j = k)\,, & \text{if } k \in I_i \\ 0\,, & \text{if } k \notin I_i \end{cases}, \tag{28}$$

$$f(X_j = i, Y = l, M_j = k) = \begin{cases} f(Y = l, M_j = k)\,, & \text{if } k \in I_i \\ 0\,, & \text{if } k \notin I_i \end{cases}, \tag{29}$$

where $I_i$ represents the set of indexes of the discretized $M_j$ variable for which $X_j = i$, and $\mathbb{1}\{k \in I_i\}$ represents the indicator function assuming value 1 when $k$ belongs to $I_i$, and 0 otherwise. (Table 3 provides an illustrative example explaining the above relations.)

Using equations 28 and 29 we can re-express equations (25) and (24) as,

$$P^\epsilon(X_j = i, M_j = k) = \begin{cases} \dfrac{f(M_j = k) + \epsilon}{n + b_x\, n\, \epsilon}\,, & \text{if } k \in I_i \\[2mm] \dfrac{\epsilon}{n + b_x\, n\, \epsilon}\,, & \text{if } k \notin I_i \end{cases}, \tag{30}$$

and,

$$P^\epsilon(X_j = i, Y = l, M_j = k) = \begin{cases} \dfrac{f(Y = l, M_j = k) + \epsilon}{n + b_x\, b_y\, n\, \epsilon}\,, & \text{if } k \in I_i \\[2mm] \dfrac{\epsilon}{n + b_x\, b_y\, n\, \epsilon}\,, & \text{if } k \notin I_i \end{cases}. \tag{31}$$

Re-expressing the conditional mutual information in terms of the smoothed probabilities and separating the summation over the cases where $k \in I_i$ from the cases where $k \notin I_i$ we have,

$$I^\epsilon(X_j; Y | M_j) = \tag{32}$$

$$= \sum_{i=1}^{b_x} \sum_{l=1}^{b_y} \sum_{k \in I_i} P^\epsilon(X_j = i, Y = l, M_j = k) \log \left( \frac{P^\epsilon(X_j = i, Y = l, M_j = k) P^\epsilon(M_j = k)}{P^\epsilon(X_j = i, M_j = k) P^\epsilon(Y = l, M_j = k)} \right) +$$

$$+ \sum_{i=1}^{b_x} \sum_{l=1}^{b_y} \sum_{k \notin I_i} P^\epsilon(X_j = i, Y = l, M_j = k) \log \left( \frac{P^\epsilon(X_j = i, Y = l, M_j = k) P^\epsilon(M_j = k)}{P^\epsilon(X_j = i, M_j = k) P^\epsilon(Y = l, M_j = k)} \right)$$

$$= \sum_{i=1}^{b_x} \sum_{l=1}^{b_y} \sum_{k \in I_i} \frac{f(Y = l, M_j = k) + \epsilon}{n + b_x b_y n \epsilon} \log \left( \frac{\left( \frac{f(Y = l, M_j = k) + \epsilon}{n + b_x b_y n \epsilon} \right) \left( \frac{f(M_j = k) + \epsilon}{n + n \epsilon} \right)}{\left( \frac{f(M_j = k) + \epsilon}{n + b_x n \epsilon} \right) \left( \frac{f(Y = l, M_j = k) + \epsilon}{n + b_y n \epsilon} \right)} \right) +$$

$$+ \sum_{i=1}^{b_x} \sum_{l=1}^{b_y} \sum_{k \notin I_i} \frac{\epsilon}{n + b_x b_y n \epsilon} \log \left( \frac{\left( \frac{\epsilon}{n + b_x b_y n \epsilon} \right) \left( \frac{f(M_j = k) + \epsilon}{n + n \epsilon} \right)}{\left( \frac{\epsilon}{n + b_x n \epsilon} \right) \left( \frac{f(Y = l, M_j = k) + \epsilon}{n + b_y n \epsilon} \right)} \right)$$

$$= \sum_{i=1}^{b_x} \sum_{l=1}^{b_y} \sum_{k \in I_i} \frac{f(Y = l, M_j = k) + \epsilon}{n + b_x b_y n \epsilon} \log \left( \frac{(n + b_x n \epsilon)(n + b_y n \epsilon)}{(n + b_x b_y n \epsilon)(n + n \epsilon)} \right) +$$

$$+ \sum_{i=1}^{b_x} \sum_{l=1}^{b_y} \sum_{k \notin I_i} \frac{\epsilon}{n + b_x b_y n \epsilon} \log \left( \frac{(n + b_x n \epsilon)(n + b_y n \epsilon)(f(M_j = k) + \epsilon)}{(n + b_x b_y n \epsilon)(n + n \epsilon)(f(Y = l, M_j = k) + \epsilon)} \right)$$

Taking the limit as $\epsilon \to 0$ we have that $I^\epsilon(X_j; Y | M_j)$ converges to,

$$\sum_{i=1}^{b_x} \sum_{l=1}^{b_y} \sum_{k \in I_i} \frac{f(Y = l, M_j = i)}{n} \log(1) + \sum_{i=1}^{b_x} \sum_{l=1}^{b_y} \sum_{k \notin I_i} 0 \log \left( \frac{f(M_j = k)}{f(Y = l, M_j = k)} \right) = 0 . \tag{33}$$

$$\square$$

### G.3 PROOF OF STATEMENT 1, IN THE REMAINING CASES

*Proof.* In the case where $X_j$ is continuous and $Y$ is categorical, a similar argument to the proof in the case that both variables are continuous shows that,

$$I^\epsilon(X_j; Y | M_j) = \tag{34}$$

$$= \sum_{i=1}^{n} \sum_{l=1}^{b_y} \frac{f(Y = l, M_j = i) + \epsilon}{n + b_y n^2 \epsilon} \log \left( \frac{(n + n^2 \epsilon)(n + b_y n \epsilon)}{(n + b_y n^2 \epsilon)(n + n \epsilon)} \right) +$$

$$+ \sum_{i=1}^{n} \sum_{l=1}^{b_y} \sum_{k \neq i} \frac{\epsilon}{n + b_y n^2 \epsilon} \log \left( \frac{(n + n^2 \epsilon)(n + b_y n \epsilon)(f(M_j = k) + \epsilon)}{(n + b_y n^2 \epsilon)(n + n \epsilon)(f(Y = l, M_j = k) + \epsilon)} \right) ,$$

which again converges to 0 as $\epsilon \to 0$.

In the case that $X_j$ is categorical and $Y$ is continuous, a similar argument to the proof in the case that both variables are categorical shows that,

$$I^\epsilon(X_j; Y | M_j) = \tag{35}$$

$$= \sum_{i=1}^{b_x} \sum_{l=1}^{n} \sum_{k \in I_i} \frac{f(Y = l, M_j = k) + \epsilon}{n + b_x n^2 \epsilon} \log \left( \frac{(n + b_x n \epsilon)(n + n^2 \epsilon)}{(n + b_x n^2 \epsilon)(n + n \epsilon)} \right) +$$

$$+ \sum_{i=1}^{b_x} \sum_{l=1}^{n} \sum_{k \notin I_i} \frac{\epsilon}{n + b_x n^2 \epsilon} \log \left( \frac{(n + b_x n \epsilon)(n + n^2 \epsilon)(f(M_j = k) + \epsilon)}{(n + b_x n^2 \epsilon)(n + n \epsilon)(f(Y = l, M_j = k) + \epsilon)} \right)$$

which again converges to 0 as $\epsilon \to 0$.

$\square$

## G.4 PROOF OF STATEMENT 2

*Proof.* By definition, the conditional entropy of $X_j$ given $M_j$ is given by,

$$
\begin{aligned}
H(X_j \mid M_j) &= \sum_i \sum_k P(X_j = i, M_j = k) \log \left( \frac{P(X_j = i, M_j = k)}{P(M_j = k)} \right) \\
&= \sum_i \sum_k P(X_j = i \mid M_j = k) P(M_j = k) \log P(X_j = i \mid M_j = k)
\end{aligned}
\tag{36}
$$

In the case where $X_j$ is continuous (and discretized into $n$ classes) we have that equation (36) reduces to,

$$
\begin{aligned}
H(X_j \mid M_j) &= \sum_{i=1}^{n} P(X_j = i \mid M_j = i) P(M_j = i) \log P(X_j = i \mid M_j = i) \\
&+ \sum_{i=1}^{n} \sum_{k \neq i} P(X_j = i \mid M_j = k) P(M_j = k) \log P(X_j = i \mid M_j = k),
\end{aligned}
\tag{37}
$$

after we split the summation over $k$ into the cases where $k = i$ and $k \neq i$. Since, by construction, we have that,

$$
P(X_j = i \mid M_j = k) = \begin{cases} 1, & \text{if } k = i \\ 0 & \text{if } k \neq i \end{cases},
\tag{38}
$$

if follows that,

$$
H(X_j \mid M_j) = \sum_{i=1}^{n} P(M_j = i) \log 1 + \sum_{i=1}^{n} \sum_{k \neq i} 0 P(M_j = k) \log 0 = 0
\tag{39}
$$

where we take $0 \log 0$ to be defined as 0 (as its is customary in information theory).

Similarly, in the case where $X_j$ is categorical (with $b_x$ classes) equation (36) reduces to,

$$
\begin{aligned}
H(X_j \mid M_j) &= \sum_{i=1}^{b_x} \sum_{k \in I_i} P(X_j = i \mid M_j = k) P(M_j = k) \log P(X_j = i \mid M_j = k) \\
&+ \sum_{i=1}^{b_x} \sum_{k \notin I_i} P(X_j = i \mid M_j = k) P(M_j = k) \log P(X_j = i \mid M_j = k),
\end{aligned}
\tag{40}
$$

after we split the summation over $k$ into the cases where $k \in I_i$ and $k \notin I_i$. Since, by construction, we also have that in the categorical case,

$$
P(X_j = i \mid M_j = k) = \begin{cases} 1, & \text{if } k \in I_i \\ 0 & \text{if } k \notin I_i \end{cases},
\tag{41}
$$

if follows again that,

$$
H(X_j \mid M_j) = \sum_{i=1}^{b_x} \sum_{k \in I_i} P(M_j = i) \log 1 + \sum_{i=1}^{b_x} \sum_{k \notin I_i} 0 P(M_j = k) \log 0 = 0 .
\tag{42}
$$

$\square$

Table 2: Illustrative toy example showing the generation of $M_j$ using Algorithm 1 when $X_j$ is continuous. The sample $\mathbf{x}_j = (0.92, -1.29, 0.34, -0.93, 0.71, 0.39, 0.65, 0.85, 0.84, 1.38, -0.32)^t$ (with $n = 11$) represents observations from $X_j$. Rows (i) and (ii) display the original values of $X_j$ and their corresponding ranks. Row (iii) shows the generated variable $M_j$, which is uniformly distributed and rank-matched to $X_j$ (i.e., the ordering of $M_j$ is identical to that of $X_j$). Row (iv) confirms this by showing that the ranks of $M_j$ match those of row (ii). Because of this perfect monotonic correspondence, the discretized versions of $X_j$ and $M_j$ are identical for any chosen number of bins. Rows (v) and (vi) illustrate this fact by presenting the discretizations of $X_j$ and $M_j$ based on $n$ bins.

| | | | | | | | | | | | | |
|---|---|---|---|---|---|---|---|---|---|---|---|---|
| (I): | ORIGINAL $X_j$ | 0.92 | -1.29 | 0.34 | -0.93 | 0.71 | 0.39 | 0.65 | 0.85 | 0.84 | 1.38 | -0.32 |
| (II): | RANKS OF $X_j$ | 10 | 1 | 4 | 2 | 7 | 5 | 6 | 9 | 8 | 11 | 3 |
| (III): | ORIGINAL $M_j$ | 0.69 | 0.12 | 0.41 | 0.20 | 0.61 | 0.44 | 0.50 | 0.67 | 0.62 | 0.74 | 0.39 |
| (IV): | RANKS OF $M_j$ | 10 | 1 | 4 | 2 | 7 | 5 | 6 | 9 | 8 | 11 | 3 |
| (V): | $X_j$ (DISCRETIZED) | 10 | 1 | 4 | 2 | 7 | 5 | 6 | 9 | 8 | 11 | 3 |
| (VI): | $M_j$ (DISCRETIZED) | 10 | 1 | 4 | 2 | 7 | 5 | 6 | 9 | 8 | 11 | 3 |

Table 3: Toy illustrative example of the generation of $M_j$ using Algorithms 1 and 10 in the case where $X_j$ is categorical. Let $\mathbf{x}_j = (A, A, A, A, B, B, B, C, C, C, C)^t$ represent a sample of size $n = 11$ from the categorical variable $X_j$, which has 3 distinct levels, $\{A, B, C\}$. Let $i = 1, 2, 3$ represent the indexes of the levels of $X_j$. Row (i) shows the data in its original format, while row (ii) shows the same data in terms of the level indexes representation, where level $A$ is indexed as 1, level $B$ as 2, and level $C$ as 3. Row (iii) shows the output, $\mathbf{r}$, of Algorithm 10 which generates a numeric rank encoding of the categorical variable $X_j$. (It adopts the arbitrary order $A < B < C$ to the categorical levels of $X_j$ and starting with class $A$, it assigns ranks 1, 2, 3, and 4 (in random order) to the four tied elements $A$ in $\mathbf{x}_j$. For class $B$, it assigns ranks 5, 6, and 7 (in random order) to the three tied elements $B$ in $\mathbf{x}_j$. Finally, for class $C$ it assigns ranks 8 to 11 (in random order) to the four tied elements $C$.) Row (iv) shows the output of Algorithm 1. It first sample and sort 11 values from a uniform distribution in the [0, 1] range (line 2) given, in this example, by (0.12, 0.20, 0.39, 0.41, 0.44, 0.50, 0.61, 0.62, 0.67, 0.69, 0.74) and then rank match this sorted random noise vector to the numeric rank encodings $\mathbf{r}$ show in line (iii). The result is a random noise vector with identical ranks as $\mathbf{r}$. Row (v) shows $M_j$ after discretization into $n = 11$ levels (in terms of level indexes representation). Clearly, when we discretize $M_j$ into $n$ levels we recover the numerical rank encoding in row (iii). Now, let $I_i$ represent the indexes of the $M_j$ values for which $X_j = i$. In this example $I_1 = \{1, 2, 3, 4\}$, $I_2 = \{5, 6, 7\}$, and $I_3 = \{8, 9, 10, 11\}$. Note that, by construction, whenever $M_j$ equals 1, 2, 3, or 4, the value of $X_j$ will be 1. This implies that $f(X_j = 1, M_j = k) = f(M_j = k) = 1$ if $k \in \{1, 2, 3, 4\}$, and $f(X_j = 1, M_j = k) = 0$ if $k \notin \{1, 2, 3, 4\}$. Similarly, $f(X_j = 2, M_j = k) = f(M_j = k) = 1$ if $k \in \{5, 6, 7\}$, and $f(X_j = 2, M_j = k) = 0$ if $k \notin \{5, 6, 7\}$, and so on for other values of $i$. Clearly, this result holds in general, as stated in equation 28. A similar argument justifies equation 29.

| | | | | | | | | | | | | |
|---|---|---|---|---|---|---|---|---|---|---|---|---|
| (I): | ORIGINAL $X_j$ | $A$ | $A$ | $A$ | $A$ | $B$ | $B$ | $B$ | $C$ | $C$ | $C$ | $C$ |
| (II): | $X_j$ LEVEL INDEXES | 1 | 1 | 1 | 1 | 2 | 2 | 2 | 3 | 3 | 3 | 3 |
| (III): | NUMERIC RANK ENCODING | 3 | 1 | 4 | 2 | 7 | 5 | 6 | 9 | 8 | 11 | 10 |
| (IV): | $M_j$ (CONTINUOUS) | 0.39 | 0.12 | 0.41 | 0.20 | 0.61 | 0.44 | 0.50 | 0.67 | 0.62 | 0.74 | 0.69 |
| (V): | $M_j$ (DISCRETIZED) | 3 | 1 | 4 | 2 | 7 | 5 | 6 | 9 | 8 | 11 | 10 |
| (VI): | $I_i$ MAPPING | $I_1 = \{1, 2, 3, 4\}$, | | | | $I_2 = \{5, 6, 7\}$, | | | $I_3 = \{8, 9, 10, 11\}$ | | | |

# H  COMPLEXITY ANALYSIS AND RUNTIME EXPERIMENTS

## H.1  COMPLEXITY ANALYSIS

As pointed in Hollmann et al. (2025) the complexity of the TabPFN algorithm scales quadratically with the number of samples ($n$) and the number of features ($p$), i.e., $O(n^2 + p^2)$.

The generation of synthetic data with the MIAV strategy (Algorithm 7) involves calling Algorithm 6 two times (see lines 3 and 4). Algorithm 6, by its turn, involves $p$ calls to TabPFN models trained with a single feature (see the *for loop* in lines 4 to 8 of Algorithm 6). Since the complexity of a TabPNF model trained with a single feature is $O(n^2 + 1) = O(n^2)$, it follows that the complexity of Algorithm 6 is $O(p\,n^2)$.[3] Consequently, the complexity of the MIAV strategy (Algorithm 7) is $O(2\,p\,n^2) = O(p\,n^2)$.

The FC approach (Algorithm 5), on the other hand, has complexity $O(p\,n^2 + p^3)$, since Algorithm 4 involves $p$ calls to TabPFN models with $p - 1$ features, so that,

$$O(p(n^2 + (p-1)^2)) = O(p\,n^2 + p^3) \,. \tag{43}$$

Similarly, the JF approach (Algorithm 3) also has complexity $O(p\,n^2 + p^3)$, since Algorithm 2 involves $p$ calls to TabPFN models with number of features increasing from 1 to $p - 1$ whose total complexity is given by,

$$\sum_{k=1}^{p-1} O(n^2 + k^2) = O\left(\sum_{k=1}^{p-1} n^2\right) + O\left(\sum_{k=1}^{p-1} k^2\right) = O((p-1)n^2) + O\left(\frac{(p-1)p(2p-1)}{6}\right)$$
$$= O(p\,n^2) + O(p^3) = O(p\,n^2 + p^3) \,. \tag{44}$$

These analyses show that, for fixed $n$, the MIAV strategy scales linearly with increasing number of variables, whereas the FC and JF approaches scale cubically with increasing number of variables.

## H.2  RUNTIME EXPERIMENTS

We performed 7 runtime experiments comparing the MIAV, JF, and FC synthetic data generation strategies. The experiments were based on simulated datasets containing 1000, 2000, 3000, 4000, 5000, 10000, or 20000 samples (rows) with number of variables (columns) varying from 10 to 100, in increments of 10. (The data was simulated from correlated beta distributions as described in Appendix C, using $a \sim \text{Uniform}(0, 10)$, $b \sim \text{Uniform}(0, 10)$, and $\rho = 0.5$.) In each experiment, the runtime was measured with the `perf_counter()` function from the `time` Python module. Results are reported in seconds and based on 5 replications of each experiment. All experiments were performed on an AWS EC2 `g5.xlarge` instance with 1 NVIDIA A10G GPU (24 GiB), 4 vCPUs, 16 GiB RAM, 250 GB NVMe SSD, and up to 10 Gbps network bandwidth.

Figure 11 reports the runtime results for number of rows ($n$) increasing from 1000 to 5000, while Figure 12 reports results for $n$ equal to 5000, 10000, and 20000. As expected, both figures show that, as the number of features increases, we observe considerably longer runtimes for FC (blue curves) and JF (orange curves) when compared to the MIAV (red curves). The figures also illustrate that the runtimes of the FC and JF strategies scale cubically with increasing number of features, whereas the runtime of the MIAV approach scales linearly. Panel f in Figure 11 and panel d in Figure 12 report the results for the MIAV approach for the distinct sample sizes side by side. (The red lines represent the median values across the 5 replications.)

---

[3]Note that the complexity of the other computations in Algorithm 6 are dominated by the TabPFN computation. Explicitly, the MIAV generation (line 5 and line 6 of Algorithm 6) has complexity $O(n \log n)$ since, as shown in Algorithm 1, it: (i) involves sorting of a random noise vector (line 2 of Algorithm 1), which has complexity $O(n \log n)$ (since random noise generation of a vector of length $n$ has complexity $O(n)$ and it's sorting has complexity $O(n \log n)$); (ii) involves the ranking of a vector of length $n$ (line 4 of Algorithm 1), which has complexity $O(n \log n)$; and (iii) involves a call to Algorithm 10 (line 7 of Algorithm 1), which has complexity $O(n)$ (as both the CummulativeSum operation (line 4 of Algorithm 10) has complexity $O(n)$ and the repeated applications of the RandomPermutation operation (line 12 of Algorithm 10) inside the *for loop* also have total complexity $O(n)$).

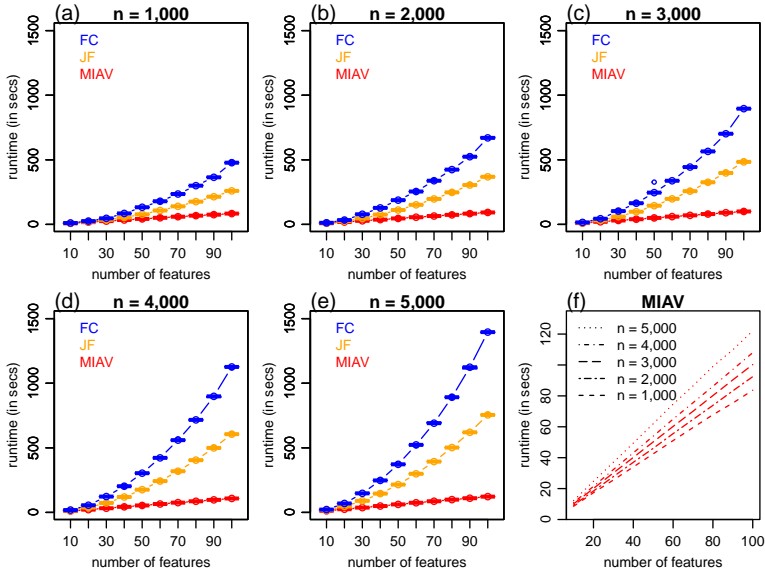

Figure 11: Runtime experiments. Panels a, b, c, d and e report runtime benchmark results comparing the FC, JF, and MIAV synthetic data generation strategies for datasets containing 1,000, 2,000, 3,000, 4,000 and 5,000 samples, respectively. Panel f compares the MIAV results across the different sample sizes.

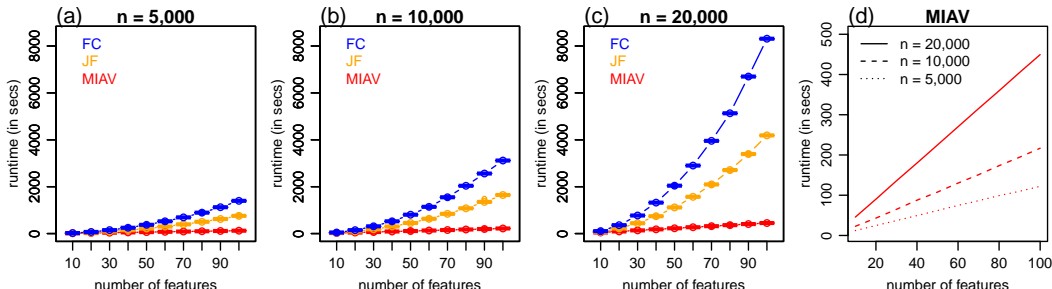

Figure 12: Runtime experiments. Panels a, b, and c report runtime benchmark results comparing the FC, JF, and MIAV synthetic data generation strategies for datasets containing 5,000, 10,000, and 20,000 samples, respectively. Panel d compares the MIAV results across the different sample sizes.

Note that in the computation of the JF approach we use a single variable order for the factorization of the joint posterior predictive distribution in all experiments in this paper. Hollmann et al. (2025) suggest using a permutation sampling approximation of Janossy pooling for combining results across different permutations of the order of the variables. (This is done to account for the fact that the variable order influences the quality of the synthesized data.) Implementation of this strategy would, however, increase the computation time by a factor of $k$, where $k$ represents the number of variable order permutations used by the approximation. (Hollmann et al. adopted $k = 24$ in their experiments.)

## I   EXTENDED EXPERIMENTS SECTION

### I.1   OVERVIEW OF DATA SPLITTING INTO ORIGINAL, HOLDOUT, TRAINING, AND TEST SETS

Our experiments are performed as follows. Each dataset $\mathbf{D}$ is first split into two datasets $\mathbf{X}$ and $\mathbf{X}^h$, denoted as the "original" and the "holdout" datasets, respectively. In our evaluations, the original dataset, $\mathbf{X}$, plays the role of the "real" data, from which we generate a synthetic data copy. (We denote it as the "original" data because $\mathbf{D}$ might be a real-world dataset or a simulated dataset in our experiments.) The holdout dataset, $\mathbf{X}^h$, is used to "estimate" the performance of an ideal data generator capable of generating data from the same distribution as the original data. In our

experiments, in addition to evaluating the synthetic datasets using the metrics described in Section I.5 below, we also compute the metric values for the holdout set in order to get a sense about the range of values we would expect to see for the metric in the ideal case of a generator truly able to draw independent samples from the same distribution as the original data.

For the generation of synthetic data, based on ICL using the TabPFN-based strategies described in the main test, namely, JF, FC, and MIAV, we further split the original data, $\mathbf{X}$, into two subsets, $\mathbf{X}_1$ and $\mathbf{X}_2$. As described in Algorithms 3, 5, and 7 in Section B, the synthetic data generation is performed as follows. First, the algorithms generate a synthetic copy of $\mathbf{X}_1$, using $\mathbf{X}_2$ as the training set and $\mathbf{X}_1$ as the test set. Second, the algorithms generate a synthetic copy of $\mathbf{X}_2$, by using $\mathbf{X}_1$ as the training set and $\mathbf{X}_2$ as the test set. Third, the algorithms concatenate the datasets generated in the previous steps to obtain the full synthetic dataset copy of $\mathbf{X}$.

For the generation of synthetic data using SMOTE (and other baselines), the original data $\mathbf{X}$ is directly fed into the synthesizer with no need for further data splits.

## I.2    SIMULATED DATA EXPERIMENTS

For the simulated data experiments we evaluate the synthetic data generators over 5 distinct settings spanning different correlation structures among the variables. In all settings, we generate datasets with 5 variables from correlated beta distributions as described in Section C. The correlation structure (Toeplitz) is controlled by a single parameter $\rho$, and in our experiments we adopt correlation strengths, $|\rho|$, in the range $|\rho| = \{0, 0.25, 0.5, 0.75, 0.95\}$.

For each experimental setting (i.e., $|\rho|$ value) we simulate 10 distinct datasets, $\mathbf{D}$, of size 800, using Toeplitz correlation parameter randomly set to either $\rho$ or $-\rho$ with equal probability, and adopting different shape parameters, $a$ and $b$, for the beta distributions. For each variable $X_j \sim \text{Beta}(a, b)$, the $a$ and $b$ parameters are randomly sampled from uniform distributions as follows:

- $a \sim U(0.1, 0.9)$, $b \sim U(0.1, 0.9)$, for $X_1$.
- $a \sim U(0.1, 0.9)$, $b \sim U(1, 10)$, for $X_2$.
- $a \sim U(10, 50)$, $b \sim U(1, 10)$, for $X_3$.
- $a \sim U(5, 15)$, $b \sim U(5, 15)$, for $X_4$.
- $a \sim U(1, 10)$, $b \sim U(5, 15)$, for $X_5$.

Each dataset $\mathbf{D}$ is split into original ($\mathbf{X}$) and holdout ($\mathbf{X}^h$) datasets of size 400. Only the $\mathbf{X}$ datasets are used by the synthetic data generators.

## I.3    REAL-WORLD DATA EXPERIMENTS

For the real-world data experiments we selected a subset of the datasets from the OpenML-CC18 benchmark suite (Bischl et al., 2021), which were analyzed in Hollmann et al. (2023, 2025). The OpenML-CC18 suite contains 72 datasets but, similarly to Hollmann et al. (2025), we selected datasets with at most 10,000 rows (examples), at most 500 columns (features), and containing categorical variables with at most 10 classes. Furthermore, because most of the evaluation metrics used in this paper are tailored to numeric data, we applied the additional filter that the datasets needed to contain more numerical variables than categorical ones. After applying these filters to the 72 datasets in OpenML-CC18 we were left with the 36 datasets listed in Table 4.

We also compared the proposed synthetic data generation strategies against popular synthetic data generators using a subset of the datasets evaluated in Hansen et al. (2023) and Chaibub Neto (2025) listed on Table 5. Note that while Table 5 contains datasets with more than 10,000 samples, we are still able to fit TabPFN models because in our evaluations we split the datasets into original and holdout sets, and the synthetic data generation is only applied to the original set (which, by its turn, is further split in 2 subsets during the ICL learning step). We allowed for larger datasets in these baseline comparison experiments because the deep-learning based baselines benefit from larger datasets. (We constrained the maximum number of samples to be less than 20,000, however, because the computation of the evaluation metrics became too slow for larger datasets.)

Similarly to Chaibub Neto (2025), the Abalone data was fetched using the commands:

Table 4: Datasets used in the real-world data evaluations. These include all 36 datasets in the OpenML-CC18 benchmark suite with at most 10,000 samples, 500 variables, 10 classes per categorical variables, and a larger number of numeric variables than categorical ones. In the first column we assign simplified dataset names (D1 to D36) to the original dataset names. The number of categorical variables is abbreviated as #cat, and the number of classes of the categorical variable with most classes is abbreviated as #class.

| NAME | ORIGINAL DATASET NAME | #SAMPLES | #COLUMNS | #CAT | #CLASS | OPENML ID |
|------|----------------------|----------|----------|------|--------|-----------|
| D1 | BALANCE-SCALE | 625 | 5 | 1 | 3 | 11 |
| D2 | MFEAT-FACTORS | 2000 | 217 | 1 | 10 | 12 |
| D3 | MFEAT-FOURIER | 2000 | 77 | 1 | 10 | 14 |
| D4 | BREAST-W | 699 | 10 | 1 | 2 | 15 |
| D5 | MFEAT-KARHUNEN | 2000 | 65 | 1 | 10 | 16 |
| D6 | MFEAT-MORPHOLOGICAL | 2000 | 7 | 1 | 10 | 18 |
| D7 | MFEAT-ZERNIKE | 2000 | 48 | 1 | 10 | 22 |
| D8 | OPTDIGITS | 5620 | 65 | 1 | 10 | 28 |
| D9 | DIABETES | 768 | 9 | 1 | 2 | 37 |
| D10 | SPAMBASE | 4601 | 58 | 1 | 2 | 43 |
| D11 | VEHICLE | 846 | 19 | 1 | 4 | 53 |
| D12 | SATIMAGE | 6430 | 37 | 1 | 6 | 2074 |
| D13 | ANALCATDATA-AUTHORSHIP | 841 | 71 | 1 | 4 | 3549 |
| D14 | PC4 | 1458 | 38 | 1 | 2 | 3902 |
| D15 | PC3 | 1563 | 38 | 1 | 2 | 3903 |
| D16 | KC2 | 522 | 22 | 1 | 2 | 3913 |
| D17 | KC1 | 2109 | 22 | 1 | 2 | 3917 |
| D18 | PC1 | 1109 | 22 | 1 | 2 | 3918 |
| D19 | WDBC | 569 | 31 | 1 | 2 | 9946 |
| D20 | PHONEME | 5404 | 6 | 1 | 2 | 9952 |
| D21 | QSAR-BIODEG | 1055 | 42 | 1 | 2 | 9957 |
| D22 | WALL-ROBOT-NAVIGATION | 5456 | 25 | 1 | 4 | 9960 |
| D23 | SEMEION | 1593 | 257 | 1 | 10 | 9964 |
| D24 | ILPD | 583 | 11 | 2 | 2 | 9971 |
| D25 | OZONE-LEVEL-8HR | 2534 | 73 | 1 | 2 | 9978 |
| D26 | FIRST-ORDER-THEOREM-PROVING | 6118 | 52 | 1 | 6 | 9985 |
| D27 | BANKNOTE-AUTHENTICATION | 1372 | 5 | 1 | 2 | 10093 |
| D28 | BLOOD-TRANSFUSION-SERVICE-CENTER | 748 | 5 | 1 | 2 | 10101 |
| D29 | GESTUREPHASESEGMENTATIONPROCESSED | 9873 | 33 | 1 | 5 | 14969 |
| D30 | MICEPROTEIN | 1080 | 78 | 1 | 8 | 146800 |
| D31 | STEEL-PLATES-FAULT | 1941 | 28 | 1 | 7 | 146817 |
| D32 | CLIMATE-MODEL-SIMULATION-CRASHES | 540 | 19 | 1 | 2 | 146819 |
| D33 | WILT | 4839 | 6 | 1 | 2 | 146820 |
| D34 | SEGMENT | 2310 | 17 | 1 | 7 | 146822 |
| D35 | MFEAT-PIXEL | 2000 | 241 | 1 | 10 | 146824 |
| D36 | CHURN | 5000 | 21 | 5 | 10 | 167141 |

Table 5: Datasets used in the baseline comparison experiments. See Table 4 for the column descriptions.

| NAME | ORIGINAL DATASET NAME | # SAMPLES | #COLUMNS | #CAT | #CLASS | OPENML ID |
|------|----------------------|-----------|----------|------|--------|-----------|
| AB | ABALONE | 4177 | 9 | 1 | 2 | - |
| BM | BANK MARKETING | 10578 | 8 | 1 | 2 | 44126 |
| CR | CREDIT | 16714 | 11 | 1 | 2 | 44089 |
| EM | EYE MOVEMENTS | 7608 | 21 | 1 | 2 | 44130 |
| HO | HOUSE 16H | 13488 | 17 | 1 | 2 | 44123 |
| MT | MAGIC TELESCOPE | 13376 | 11 | 1 | 2 | 44125 |
| PO | POL | 10082 | 27 | 1 | 2 | 44122 |

```
from sklearn.datasets import fetch_openml
```

```
fetch_openml(name="abalone", version=1, as_frame=True)
```

The other datasets were fetched using:

```
openml.datasets.get_dataset(openmlid).
```

## I.4 HYPERPARAMETERS FOR THE BASELINE MODELS

Table 6: Hyperparameters used for the generative models trained on the Abalone (AB) dataset.

| Model | Parameters |
|---|---|
| DDPM | n_iter: 7605
lr: 0.002991978123076162
batch_size: 970
num_timesteps: 407
is_classification: False |
| ARF | num_trees: 80
delta: 0
max_iters: 2
early_stop: False
min_node_size: 2 |
| TVAE | n_iter: 400
lr: 0.001
decoder_n_layers_hidden: 5
weight_decay: 0.0001
batch_size: 128
n_units_embedding: 200
decoder_n_units_hidden: 150
decoder_nonlin: tanh
decoder_dropout: 0.19964446358158816
encoder_n_layers_hidden: 4
encoder_n_units_hidden: 100
encoder_nonlin: relu
encoder_dropout: 0.0820245231222064 |
| CTGAN | n_iter: 700
generator_n_layers_hidden: 1
generator_n_units_hidden: 100
generator_nonlin: elu
generator_dropout: 0.13836424598477665
discriminator_n_layers_hidden: 2
discriminator_n_units_hidden: 100
discriminator_nonlin: tanh
discriminator_n_iter: 5
discriminator_dropout: 0.0238615659365287977
lr: 0.001
weight_decay: 0.0001
batch_size: 200
encoder_max_clusters: 8 |

Table 7: Hyperparameters used for the generative models trained on Bank marketing (BM), Credit (CR), Eye movements (EM), House 16H (HO), Magic telescope (MT), and Pol (PO) datasets.

| Model | Parameters |
|---|---|
| DDPM | n_iter: 1051
lr: 0.0009375080542687667
batch_size: 2929
num_timesteps: 998
is_classification: True |
| ARF | num_trees: 30
delta: 0
max_iters: 10
early_stop: True
min_node_size: 5 |
| TVAE | n_iter: 300
lr: 0.0002
decoder_n_layers_hidden: 4
weight_decay: 0.001
batch_size: 256
n_units_embedding: 200
decoder_n_units_hidden: 300
decoder_nonlin: elu
decoder_dropout: 0.194325119117226
encoder_n_layers_hidden: 1
encoder_n_units_hidden: 450
encoder_nonlin: leaky_relu
encoder_dropout: 0.04288563703094718 |
| CTGAN | n_iter: 1000
generator_n_layers_hidden: 2
generator_n_units_hidden: 50
generator_nonlin: tanh
generator_dropout: 0.0575
discriminator_n_layers_hidden: 4
discriminator_n_units_hidden: 150
discriminator_nonlin: relu
discriminator_n_iter: 1
discriminator_dropout: 0.1
lr: 0.001
weight_decay: 0.001
batch_size: 200
encoder_max_clusters: 10 |

## I.5 EVALUATION METRICS

We evaluated the quality of the synthetic data in terms of data utility, data fidelity, and data privacy.

Data utility was evaluated using the machine learning efficiency (MLE) metric for measuring utility in downstream predictive tasks. MLE was computed using a random forest learner trained in the synthetic data and evaluated in the holdout set (which corresponds to real data). (To measure the ground truth performance, we train the random forest learner on the original data and evaluate its performance on the holdout set.) The predictive performance was measured using AUROC in classification tasks and $R^2$ in regression tasks. Hence, larger values of MLE imply better utility of the synthetic data in downstream ML tasks.

Data fidelity was evaluated with respect to the quality of the marginal distributions, quality of the pairwise statistical associations, overall data quality, and quality of the joint distribution according to the following metrics:

- **Average KS-statistic (KS).** This metric is used to evaluate the quality of the synthetic data marginal distributions. It is based on the Kolmogorov-Smirnov two-sample statistical test (KS-test) for the equality of distributions. For each variable it computes the KS-test statistic between the synthetic and original data, and the metric corresponds to the average KS-statistic across all variables. Lower values of this metric indicate better agreement between the synthetic and original marginal distributions.

- **L2 distance between association matrices (L2D).** This metric is used to evaluate how well the synthetic data recovers the pairwise statistical associations observed in the original data. The L2 distance is computed as the average of the squared difference between the elements of the synthetic and original data association matrices. Lower values of this metric indicate better agreement between the synthetic and original data pairwise statistical associations. Since the datasets might include both numerical and categorical variables, we assess pairwise associations as follows: for numerical pairs we use Pearson correlation; for categorical pairs we use the Cramer's V statistic; for numeric/categorical pairs we regress the numeric variable on the categorical one and use the square root of the $R^2$ statistic as our association measure (this reduces to the absolute correlation coefficient when both variables are numerical).

- **Wasserstein distance (WD).** The WD metric, also known as the Earth Mover's Distance, quantifies the dissimilarity between two probability distributions by measuring the minimum "cost" of transporting probability mass from one distribution to the other. It is used to evaluate how well the joint probability distribution of the synthetic data approximates the joint distribution of the original data. Lower values of this metric indicate better agreement between the synthetic and original joint probability distributions. Because we adopt the squared Euclidean distance cost matrix for the computation of the transport cost, this metric is sensitive to scale (i.e., variables with larger numerical ranges dominate the distance). Hence, we first re-scale all variables before computing the WD.

- **Energy distance (ED).** This metric is also used to evaluate how well the joint probability distribution of the synthetic data approximates the joint distribution of the original data. Again, lower values of this metric indicate better agreement between the synthetic and original joint probability distributions. (Energy distance (Szekely and Rizzo, 2023) represents a special case of the maximum mean discrepancy statistic.) Since ED uses Euclidean norms to compare observations, it is also sensitive to scale (i.e., variables with larger numerical ranges dominate the distance). Hence, we first re-scale all variables before computing the ED.

- **Detection test (DT).** This metric measures the overall quality of the synthetic data by evaluating the performance of a classifier trained to discriminate between synthetic and original data examples. When the synthetic data is indistinguishable from the original data the classifier should achieve a random guess performance level, otherwise the classifier performance should be better than random. In our experiments, we use a random forest classifier and evaluate classification performance using AUROC. Values closer to AUROC = 0.5 indicate better agreement between the synthetic and original data.

Data privacy was evaluated according to the following metrics:

- **Distance to closest record (DCR).** This represents a record-level privacy metric that measures how similar a synthetic record is to the closest record in the original dataset. It is computed by measuring, for each synthetic record, the minimum distance to any record in the original dataset using Euclidean distance; values close to zero indicate higher disclosure risk because synthetic records are nearly indistinguishable from the original data ones, whereas larger values suggest safer privacy protection by ensuring greater separation between synthetic and real data (this might be achieved at the cost of lower data fidelity, though). Since Euclidean distance is sensitive to scale (i.e., variables with larger numerical ranges dominate the distance), we first re-scale all variables before computing the DCR.

- **Sorted distance-based record linkage (SDBRL).** This metric represents a variant of the Distance-Based Record Linkage (DBRL) metric. The DBRL metric (Pagliuca and Seri, 1999; Domingo-Ferrer and Torra, 2001) is a widely used method for evaluating re-identification risk of perturbation methods within the Statistical Disclosure Control field (Drechsler, 2011). It operates by computing the Euclidean distance between each record in the perturbed dataset and all records in the original dataset, designating a perturbed record as 'linked' when its nearest neighbor corresponds to its true original record. The DBRL value is then given by the proportion of perturbed records successfully linked back to their original counterparts. Strictly speaking, the DBRL metric is only intended for evaluating data perturbation methods, since it assumes the existence of a direct mapping between the original and perturbed values (for example, when perturbed data are obtained by adding noise to the original data). In the case of synthetic data, where such a mapping is absent, an approximate correspondence can still be established following the approach of Domingo-Ferrer et al. (2020) and Chaibub Neto (2024, 2025). The idea is to sort the rows of both the original and synthetic datasets by the values of a chosen attribute (column) and then compute the metric on these sorted datasets. (A rationale for this procedure is given in section 3 of Domingo-Ferrer et al. (2020).) This adapted version of the DBRL metric is referred to as the "sorted DBRL" metric, or SDBRL for short.

- **Sorted standard deviation interval distance (SSDID).** This metric represents a variant of the Standard Deviation Interval Distance (SDID) metric. The SDID metric Mateo-Sanz et al. (2004) is a commonly used method for evaluating attribute disclosure risk of perturbation methods in the Statistical Disclosure Control literature. It corresponds to the proportion of original records inside a standard deviation interval whose center is the corresponding perturbed record (where the interval width is computed in terms of a percentage $p$ of the standard deviation of the variable). A record $i$ in the original dataset is considered to be inside the standard deviation interval of the perturbed record $i$ if, for all variables $j$, it is inside the respective standard deviation interval. Similarly to DBRL, the SDID metric also assumes the existence of a mapping between the original and perturbed values, and we adopted the sorted version of this metric (SSDID) proposed by Chaibub Neto (2024, 2025) in our synthetic data evaluations.

In our evaluations, we only compute the KS, WD, ED, DCR, SDBRL, and SSDID metrics for numeric variables. (Note that our simulated datasets contain only numeric variables and, as shown in Table 4, the real-world datasets contained mostly numeric variables as well.)

## I.6    RESULTS FOR ADDITIONAL EVALUATION METRICS

In the main text, we present evaluation results for three fidelity metrics (namely, KS, L2D, and DT) and three privacy metrics (namely, DCR, SDBRL, and SSDID). Here, we present additional evaluation results for the: machine learning efficiency (MLE) metric, which measures data utility in downstream predictive tasks; and the Wasserstein distance and the energy distance metrics, both of which measure data fidelity by evaluating how well the joint probability distribution of the synthetic data approximates the joint distribution of the original data.

Figure 13 report the results (which were computed in the exact same data splits and synthetic datasets as the previous evaluation metrics). Panels a, b, and c show the results from the experiments based on simulated data draw from correlated beta distributions. Panels d, e, and f report the results for the 36 real-world datasets from the OpenML-CC18 benchmark suite described in Table 4. Panels g, h, and i show the results for the 7 additional datasets evaluated in Hansen et al. (2023) and Chaibub Neto (2025) described in Table 5.

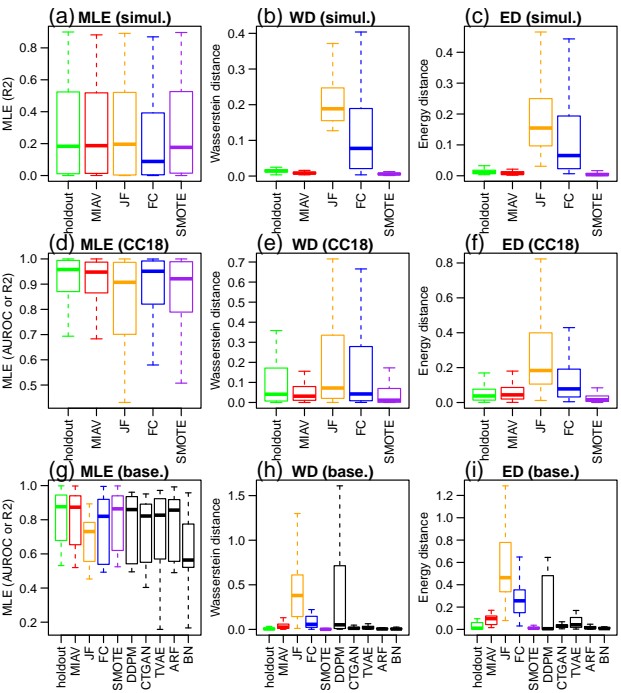

Figure 13: Pooled experimental results. Top panels show results pooled across the 5 simulated dataset settings. The middle panels show results pooled across the 36 real-world datasets selected from the OpenML-CC18 suite. The bottom panels show results pooled across the 7 real-world datasets used for the baseline generator comparisons. For the MLE metric, higher values indicate better data utility (MLE is measured by AUROC in classification tasks and $R^2$ in regression tasks). For the WD and ED metrics, lower values indicate better fidelity.

In terms of MLE (panels a, d, and g), where larger values indicate better utility, the MIAV approach (red boxplots) performs consistently well across all experiments, closely tracking the ground truth performance, reported by the holdout set results (green boxplots). Overall, most baselines tended to show comparable performances, with the exception of the JF and the bayesian-network (BN) generators in panel g (and the FC generator in panel a), which tended to underperform when compared to the other generators.

In terms of the WD and ED fidelity metrics, where lower values indicate better fidelity, the SMOTE and MIAV approaches tended to outperform the JF and FC methods in all experiments. Interestingly, the comparisons of WD and ED across the other baselines (panels h and i) show some surprising results. In particular, the bayesian-network (BN) generator, which achieved the worse data utility performance as measured by MLE (panel g) as well as the worse data fidelity performances in terms of the L2D and DT metrics (see panels n and o in Figure 4 in the main text), tended to produce surprisingly strong results in terms of WD and ED. The DDPM baseline, on the other hand, achieved weak performance in terms of WD and ED despite ranking among the strongest baselines in all other data utility and data fidelity metrics. These observations corroborate similar surprising results reported by Hansen et al. (2023), which also observed better fidelity scores for bayesian-network than for the DDPM model in their experiments. These findings provide yet another example of how distinct performance metrics do not always generate consistent conclusions, and underscore the need for evaluations based on multiple metrics.

## I.7 EXTENDED RESULTS

Due to space limitations, in the main text we only present experimental results pooled across all datasets. Here we provide more detailed results. Figure 14 presents the results from the simulated data experiments separated by simulation setting. Figures 15, 16, 17, and 18 present separated results for each of the OpenML-CC18 real-world datasets. Figure 19 presents separated results for each of the real-world datasets in Table 5.

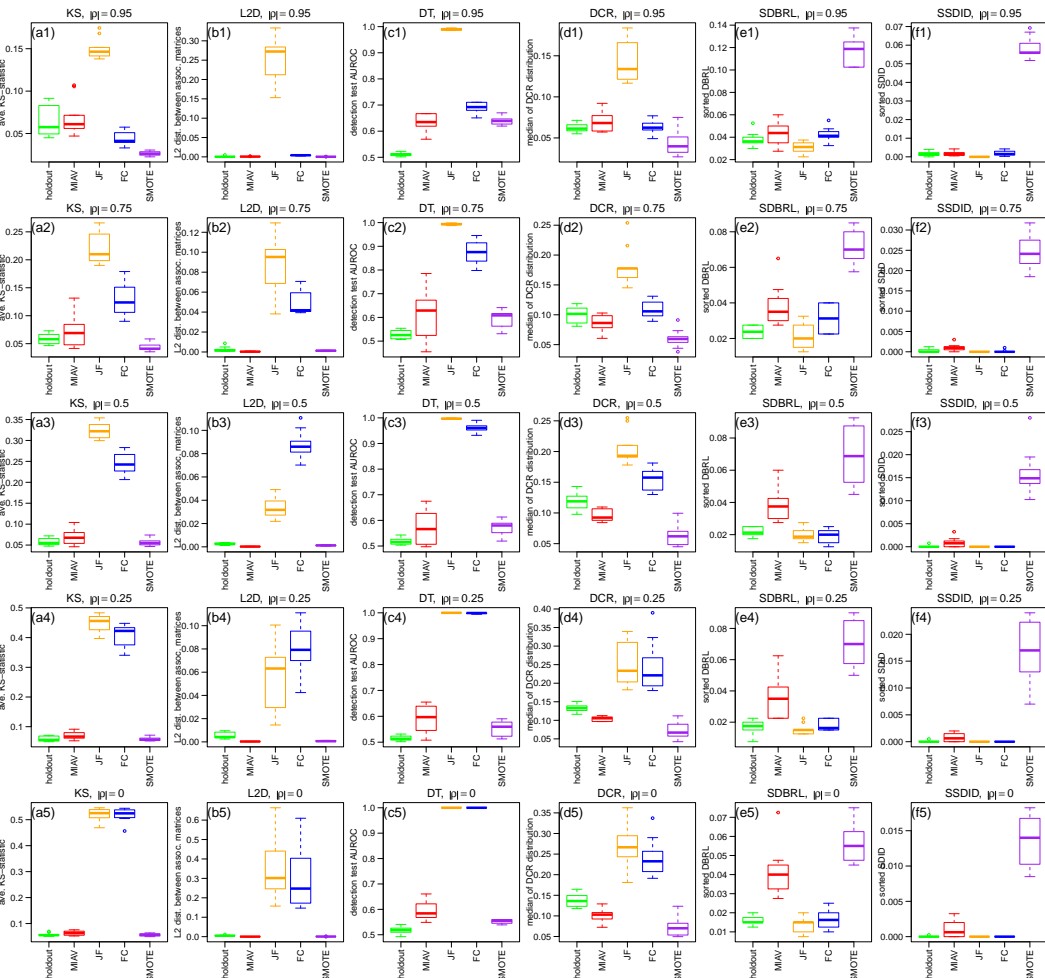

Figure 14: Simulated data experiments separated by simulation setting. Each boxplot displays the results from 10 separate replications based on different simulation parameters.

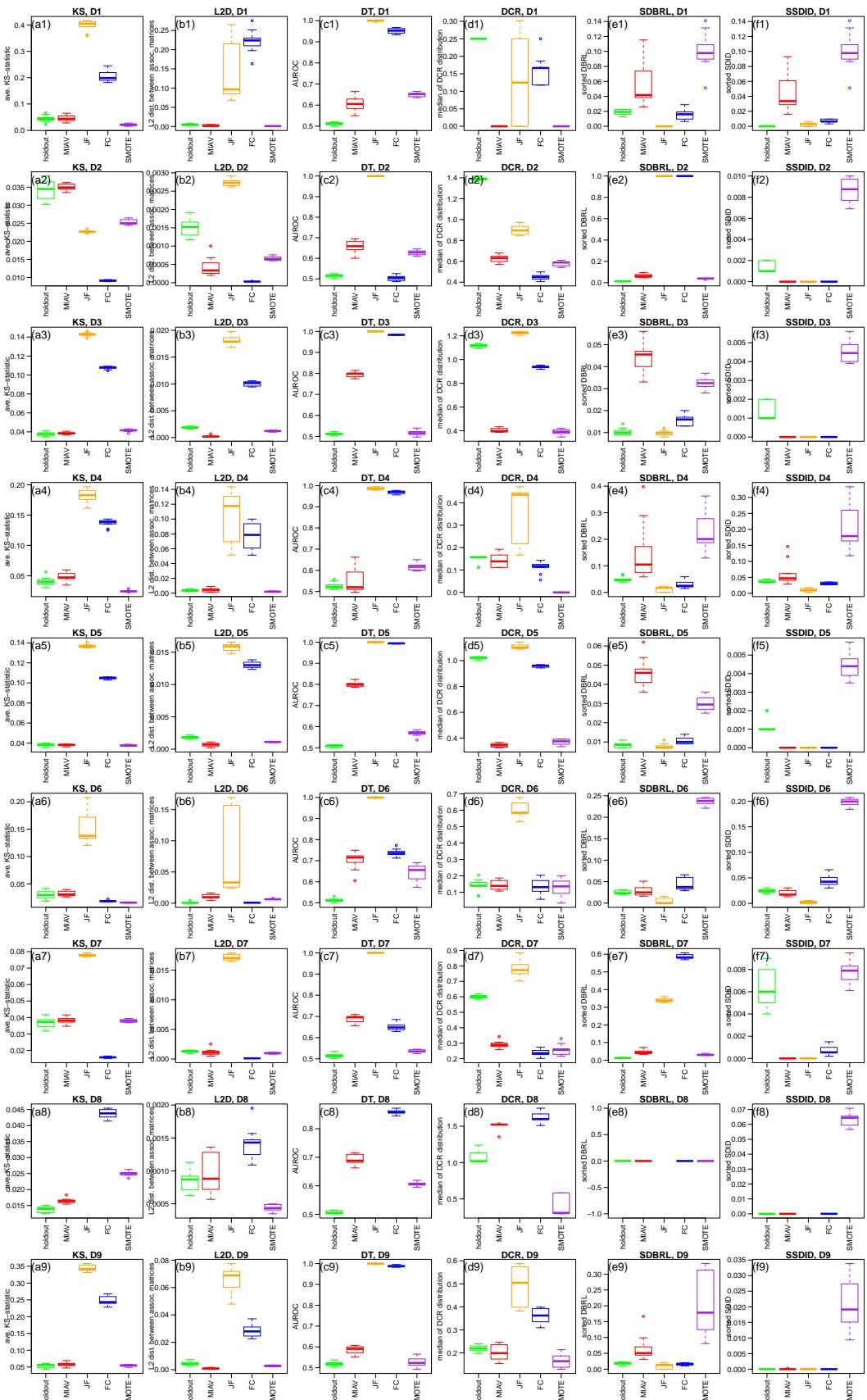

Figure 15: Experiment results for datasets D1 to D9 (see Table 4 for the original dataset names). Each boxplot displays the results from 10 distinct original/holdout data splits.

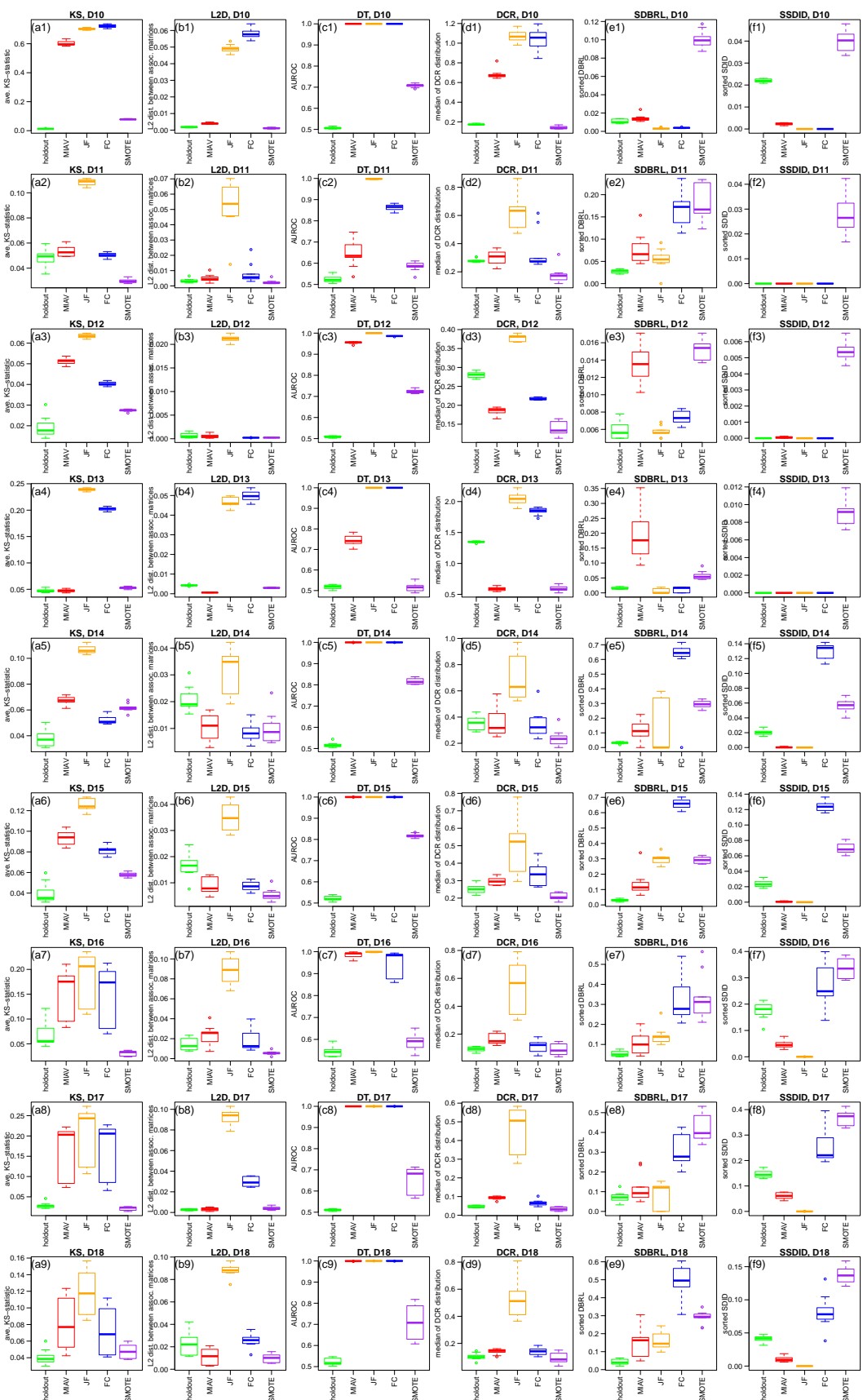

Figure 16: Experiment results for datasets D10 to D18 (see Table 4 for the original dataset names). Each boxplot displays the results from 10 distinct original/holdout data splits.

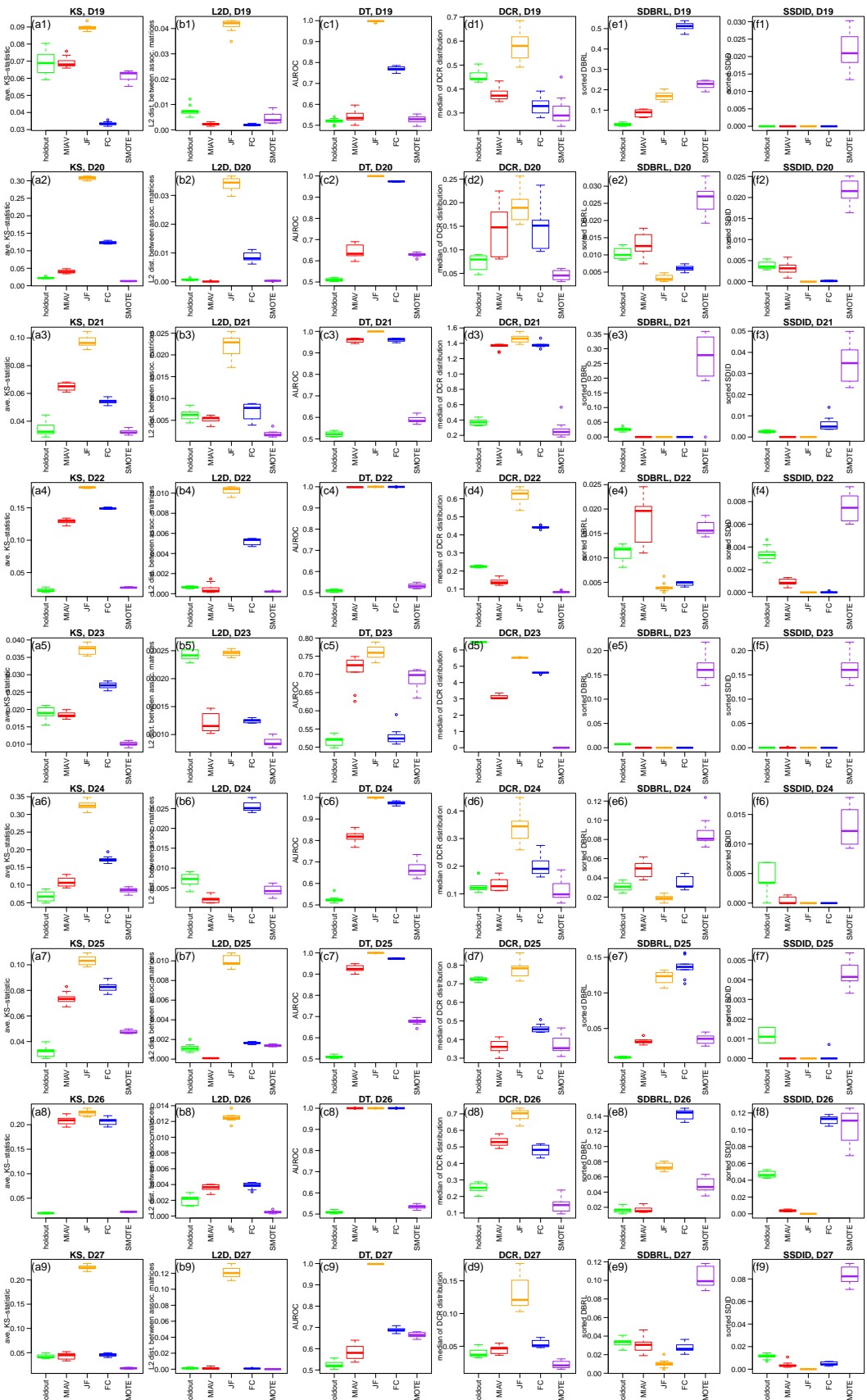

Figure 17: Experiment results for datasets D19 to D27 (see Table 4 for the original dataset names). Each boxplot displays the results from 10 distinct original/holdout data splits.

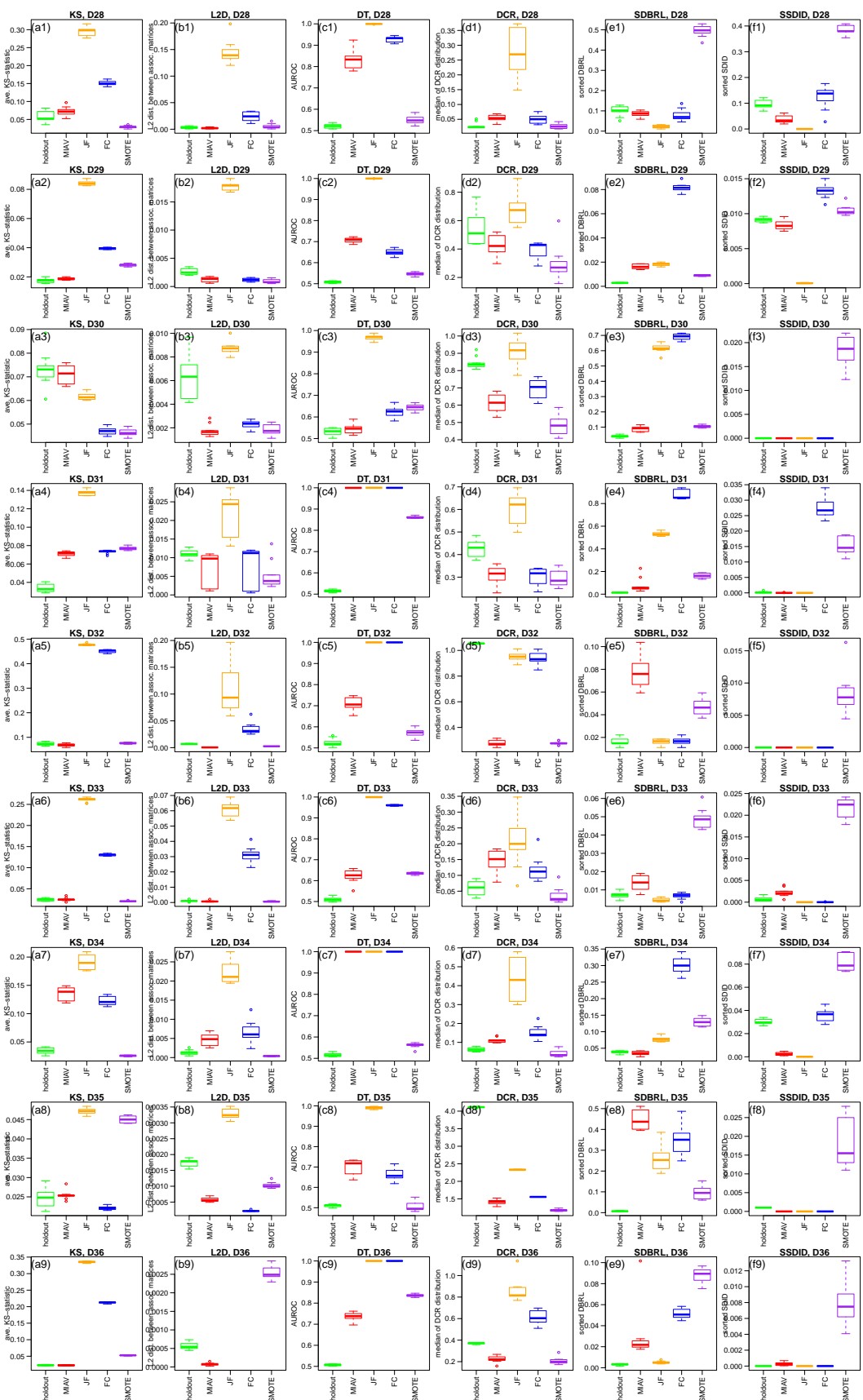

Figure 18: Experiment results for datasets D28 to D36 (see Table 4 for the original dataset names). Each boxplot displays the results from 10 distinct original/holdout data splits.

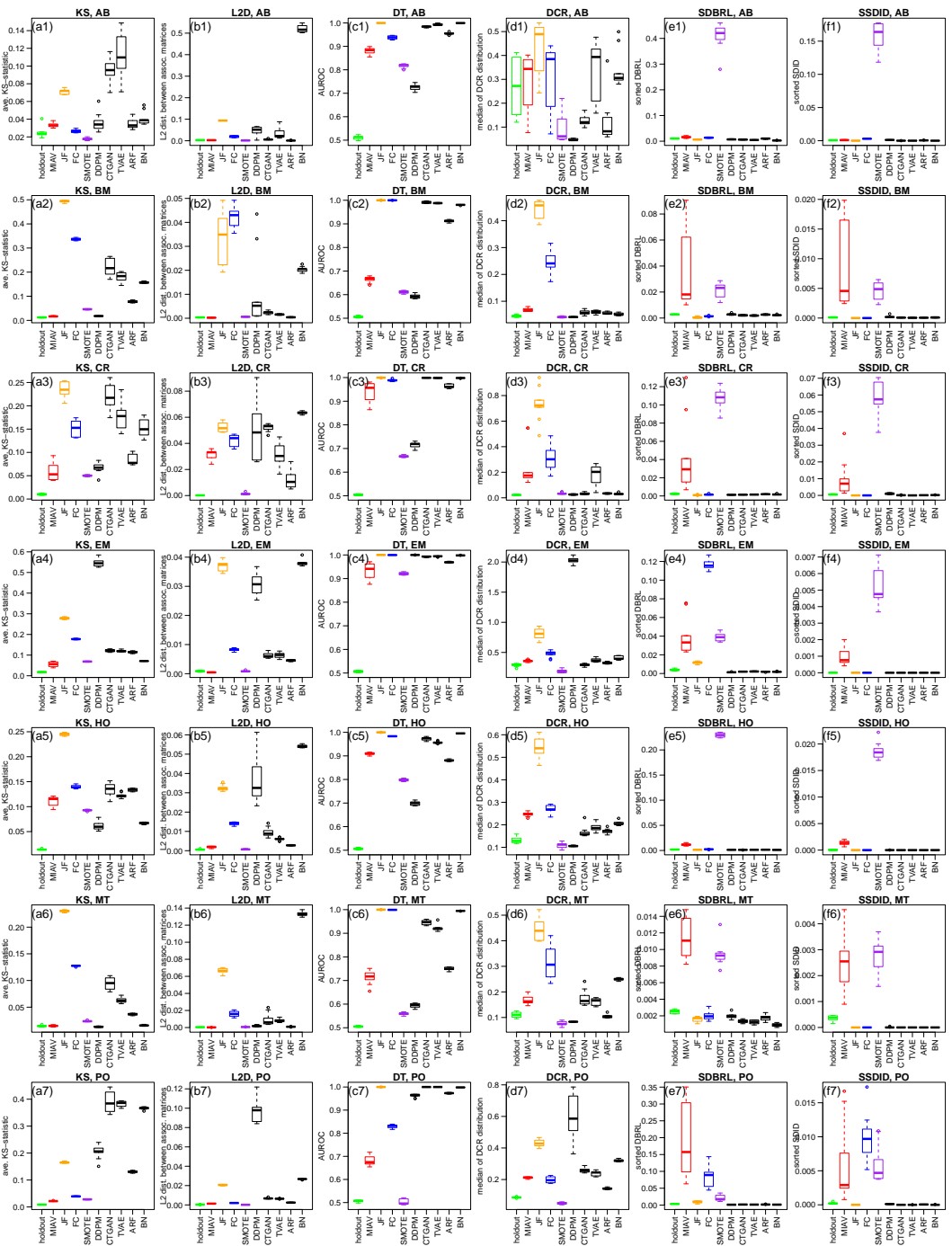

Figure 19: Experiment results for datasets in Table 5. Each boxplot displays the results from 10 distinct original/holdout data splits.

## J  THE NOISY-MIAV STRATEGY

A simple way to improve the privacy of the MIAV synthetic data generation approach is to add controlled amounts of noise to the MIAV before generating the synthetic data with the TabPFN model. The details are presented in Algorithms 11 and 12. As shown in lines 7 and 9 of Algorithm 11, we generate noisy versions of the MIAV by adding Gaussian noise with mean 0 and standard deviance equal to a fixed percentage of the standard deviation of the respective MIAV. By increasing the percent of noise parameter we generate increasingly noisier versions of the MIAV.

---

**Algorithm 11** ICLwithNoisyMIAVTabPFN($\mathbf{X}^{tr}$, $\mathbf{X}^{ts}$, percent)

---
1: **Input:** training data for ICL, $\mathbf{X}^{tr}$; query data for ICL, $\mathbf{X}^{ts}$
2: $n \leftarrow$ NumberOfRows($\mathbf{X}^{ts}$) {Obtain number of rows of $\mathbf{X}^{ts}$.}
3: $p \leftarrow$ NumberOfColumns($\mathbf{X}^{ts}$) {Obtain number of columns of $\mathbf{X}^{ts}$.}
4: $\mathbf{Z}^{ts} \leftarrow [,]$ {Create empty matrix to store the synthetic data.}
5: **for** $j = 1$ **to** $p$ **do**
6:    $\mathbf{m}_j^{tr} \leftarrow$ GenerateMaximalInformationAuxiliaryVariable($\mathbf{x}_j^{tr}$) {Generate the MIAV for $\mathbf{x}_j^{tr}$ using Algorithm 1.}
7:    $\mathbf{m}_j^{tr} \leftarrow \mathbf{m}_j^{tr} +$ GenerateNormalVariable(size $= n$, mean $= 0$, sd $=$ percent $* \operatorname{sd}(\mathbf{m}_j^{tr})$) {Add Gaussian noise to the MIAV according to a specified percent of the MIAV's standard deviation.}
8:    $\mathbf{m}_j^{ts} \leftarrow$ GenerateMaximalInformationAuxiliaryVariable($\mathbf{x}_j^{ts}$) {Generate the MIAV for $\mathbf{x}_j^{ts}$ using Algorithm 1.}
9:    $\mathbf{m}_j^{ts} \leftarrow \mathbf{m}_j^{ts} +$ GenerateNormalVariable(size $= n$, mean $= 0$, sd $=$ percent $* \operatorname{sd}(\mathbf{m}_j^{ts})$) {Add Gaussian noise to the MIAV according to a specified percent of the MIAV's standard deviation.}
10:    $\mathbf{Z}^{ts}[, j] \leftarrow$ GeneratePredictionUsingTabPFN($\mathbf{m}_j^{ts}$, $\mathbf{m}_j^{tr}$, $\mathbf{x}_j^{tr}$) {Predict $\mathbf{x}_j^{ts}$ using $\mathbf{m}_j^{tr}$ and $\mathbf{x}_j^{tr}$ as context, and $\mathbf{m}_j^{ts}$ as query. The prediction can be from a regression or classification TabPFN model, depending on whether $\mathbf{x}_j^{tr}$ is continuous or categorical.}
11: **end for**
12: **Output:** synthetic data $\mathbf{Z}^{ts}$

---

**Algorithm 12** NoisyMIAVTabPFNGenerator($\mathbf{X}$, percent)

---
1: **Input:** the original data, $\mathbf{X}$
2: $\mathbf{X}_1, \mathbf{X}_2 \leftarrow$ DataSplit($\mathbf{X}$) {Split the original data $\mathbf{X}$ into two subsets, $\mathbf{X}_1$ and $\mathbf{X}_2$.}
3: $\mathbf{Z}_1 \leftarrow$ ICLwithNoisyMIAVTabPFN($\mathbf{X}^{tr} = \mathbf{X}_2, \mathbf{X}^{ts} = \mathbf{X}_1$, percent) {Generate a synthetic data copy of $\mathbf{X}_1$ using Alg. 6.}
4: $\mathbf{Z}_2 \leftarrow$ ICLwithNoisyMIAVTabPFN($\mathbf{X}^{tr} = \mathbf{X}_1, \mathbf{X}^{ts} = \mathbf{X}_2$, percent) {Generate a synthetic data copy of $\mathbf{X}_2$ using Alg. 6.}
5: $\mathbf{Z} \leftarrow$ Concatenate($\mathbf{Z}_1, \mathbf{Z}_2$) {Concatenate the synthetic datasets $\mathbf{Z}_1$ and $\mathbf{Z}_2$.}
6: $\mathbf{Z} \leftarrow$ RoundIntegerVariables($\mathbf{X}, \mathbf{Z}$) {This function uses $\mathbf{X}$ to determine which variables have integer type and round the values of the corresponding variables in $\mathbf{Z}$ to the nearest integer.}
7: **Output:** synthetic data $\mathbf{Z}$

---

Figure 20 shows an illustrative example (using simulated dataset with correlated beta distributed data). Panels a1 to e1 show a scatterplots of the MIAVs, $M_j$ versus the original variable, $X_j$. Panel a2 to e2 show the respective scatterplots for the noisy MIAVs obtained by adding Gaussian noise with standard deviation given by $0.1 \times \operatorname{sd}(M_j)$ (where $\operatorname{sd}(M_j)$ represents the standard deviation of the MIAV $M_j$). Similarly, panels a3 to e3 and panels a4 to e4 show the scatterplots for the noisy MIAVs obtained with standard deviations $0.2 \times \operatorname{sd}(M_j)$, and $0.3 \times \operatorname{sd}(M_j)$, respectively.

Figure 21 compares the densities of synthetic data generated with the MIAV approach (panels a1 to e1) against synthetic data generated with the noisy-MIAV approach using increasing amounts of noise (namely, $0.1 \times \operatorname{sd}(M_j)$ for panels a2 to e2, $0.2 \times \operatorname{sd}(M_j)$ for panels a3 to e3, and $0.3 \times \operatorname{sd}(M_j)$ for panels a4 to e4.)

Finally, in Figure 22 we report the results from synthetic and real-world data experiments evaluating the noisy-MIAV approach w.r.t. the same fidelity and privacy metrics using the same 43 datasets evaluated in the main paper. As before, the results were based on 10 distinct random splits of the data into original and holdout datasets and the figure report results pooled across all datasets. As expected, the noisy-MIAV approach trades an increase in data privacy by a decrease in data fidelity.

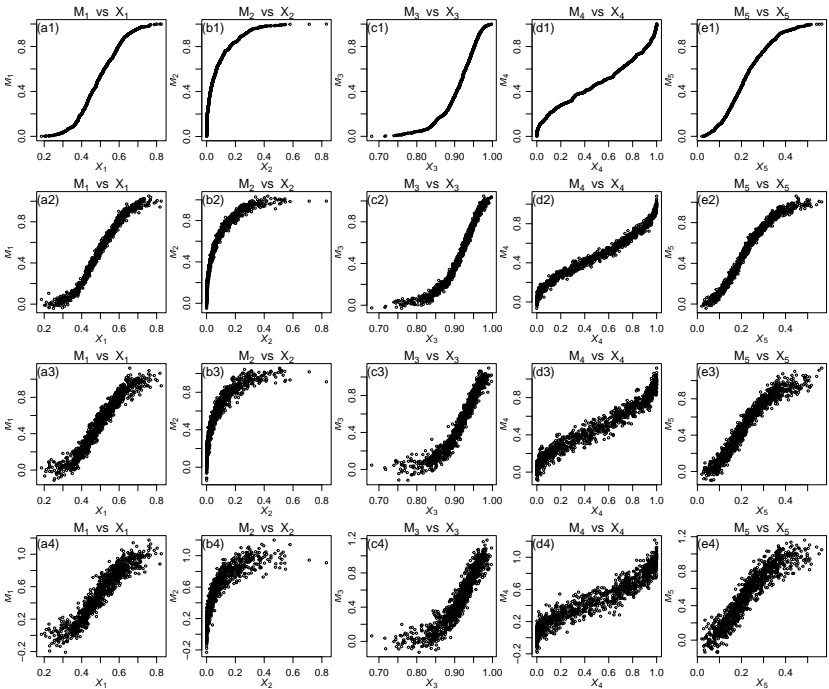

Figure 20: MIAV and noisy-MIAV versus original data scatterplots. Panels a1-e1, a2-e2, a3-e3, and a4-e4 show scatterplots of the original data versus MIAV and original data versus noisy-MIAVs generated with increasing amounts of noise.

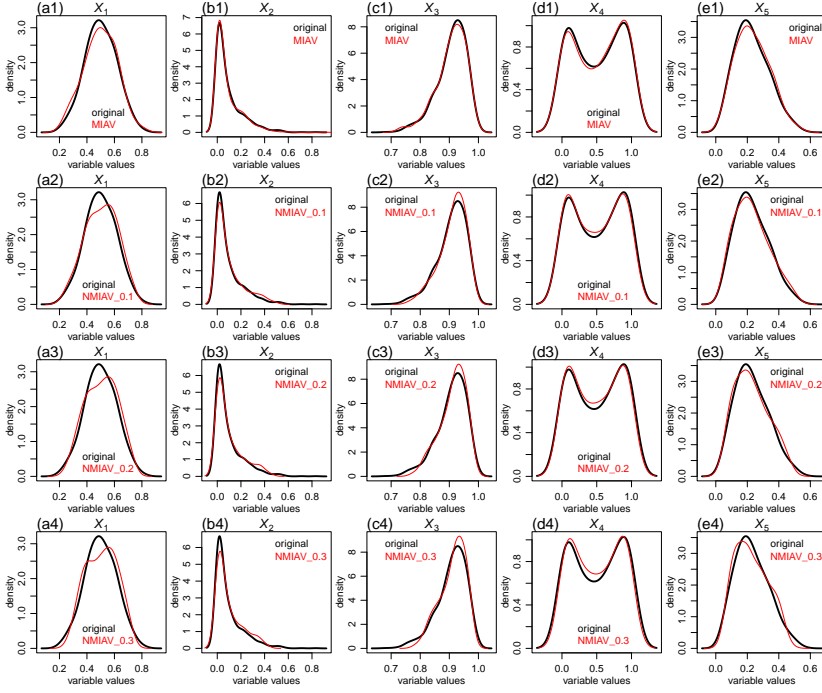

Figure 21: Marginal distributions generated with the noisy-MIAV strategy. Panels a1 to e1 report the densities based on the standard MIAV. Panel a2-e2, a3-e3, and a4-e4 show densities based on the noisy-MIAV (NMIAV) generated with noise percent set to 0.1, 0.2, and 0.3, respectively.

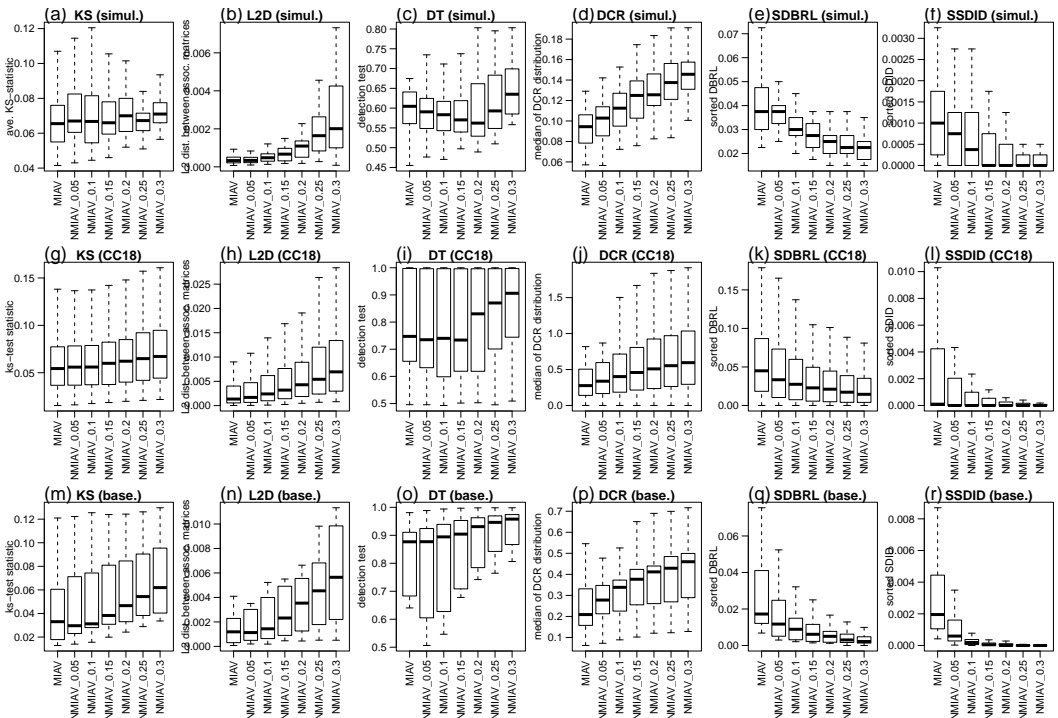

Figure 22: Experiment results for the noisy-MIAV strategy with noise percent increasing from 0.05 to 0.3 (in 0.05 increments). Increasing amounts of noise lead to decreasing data fidelity (larger values for the KS, L2D, and DT metrics), but improved data privacy (larger DCR values and lower SDBRL and SSDID values). Each boxplot displays the results from 10 distinct original/holdout data splits. Top panels show the pooled results for the synthetic data experiments. Middle panels show the pooled results for the OpenML-CC18 datasets in Table 4. Bottom panels show pooled results for the datasets in Table 5. (The boxplots omit outliers to improve visualization.)

# K    Synthetic data generation based on TabICL PFN models

To illustrate that our proposed synthetic data generation strategy is not restricted to TabPFN models, here we illustrate its application in conjunction with the TabICL model (Qu et al., 2025), which corresponds to an alternative PFN-based tabular foundation model. TabICL provides a much more scalable alternative to TabPFN, being able to handle datasets with up to 500,000 examples using affordable compute resources. However, similarly to TabPFN, it is still constrained to datasets with at most 500 features. (Observe, however, that because the MIAV strategy only requires training of TabPFN (or TabICL) models with a single feature per variable, this constraint has no impact on the MIAV approach, which can potentially be used to generate synthetic data versions of datasets containing more than 500 variables.)

Because TabICL currently can only handle classification tasks, our illustrations are restricted to datasets containing only categorical variables. Since we are also comparing the TabICL-based synthetic data generation against the strategies based on TabPFN we restrict our comparisons to the datasets in the OpenML-CC18 benchmark suite with at most 10,000 samples, 500 features, containing categorical variables with at most 10 level classes, and which contain more categorical variables than numeric ones. (We dropped the numerical variables from the few datasets that contained both numeric and categorical variables.) Table 8 shows the selected datasets for these comparisons.

As before, we consider 3 data generation strategies, namely, the MIAV-TabICL, the JF-TabICL, and the FC-TabICL and compare it against the corresponding TabPFN based MIAV, JF, and FC strategies (denoted as MIAV-TabPFN, the JF-TabPFN, and the FC-TabPFN in the plots below).

Table 8: Datasets used for the TabICL illustrations and comparisons with TabPFN. These include all 8 datasets in the OpenML-CC18 benchmark suite with at most 10,000 samples, 500 variables, 10 classes per categorical variables, and a larger number of categorical variables than numeric ones. In the first column we assign simplified dataset names (C1 to C8) to the original dataset names. The number of categorical variables is abbreviated as #cat, and the number of classes of the categorical variable with most classes is abbreviated as #class. In datasets C2 and C3 we remove the numerical variables before running the analyses.

| NAME | ORIGINAL DATASET NAME | #SAMPLES | #COLUMNS | #CAT | #CLASS | OPENML ID |
|---|---|---|---|---|---|---|
| C1 | KR-VS-KP | 3196 | 37 | 37 | 3 | 3 |
| C2 | CMC | 1473 | 10 | 8 | 4 | 23 |
| C3 | CREDIT-G | 1000 | 21 | 14 | 10 | 31 |
| C4 | SPLICE | 3190 | 61 | 61 | 6 | 45 |
| C5 | TIC-TAC-TOE | 958 | 10 | 10 | 3 | 49 |
| C6 | ANALCATDATA-DMFT | 797 | 5 | 5 | 9 | 3560 |
| C7 | CAR | 1728 | 7 | 7 | 4 | 146821 |
| C8 | DNA | 3186 | 181 | 181 | 3 | 167140 |

The implementation of the TabICL-based strategies is analogous to the TabPFN ones (we just need to switch the `GeneratePreditionsUsingTabPFN()` function in Algorithms 2, 4, and 6 by the corresponding `GeneratePreditionsUsingTabICL()` function).

The experiments were run as before, where each dataset was first randomly split into approximately equal sized original and holdout sets, and we report the results from 10 data splits. Since we only consider categorical datasets, we adopt: (i) the average KL-divergence metric to measure the quality of the synthetic data marginal distributions (where we compute separate KL-divergence scores for each column of the dataset and take the average across all columns as the final metric); and (ii) the L2D metric for measuring how well the synthetic data captures the pairwise statistical associations observed in the original data (where the pairwise associations of the categorical variables are measured with the Cramer-V statistic.). For both of these metrics, lower values indicate better data fidelity.

Figures 23 to 24 report the results for the average KL-divergence and L2D metrics, respectively. For comparison, the figures also show the results for the holdout datasets. For all datasets, the results for MIAV-TabICL and MIAV-TabPFN were very close. As before, the MIAV-based strategies tended to outperform the JF and FC ones.

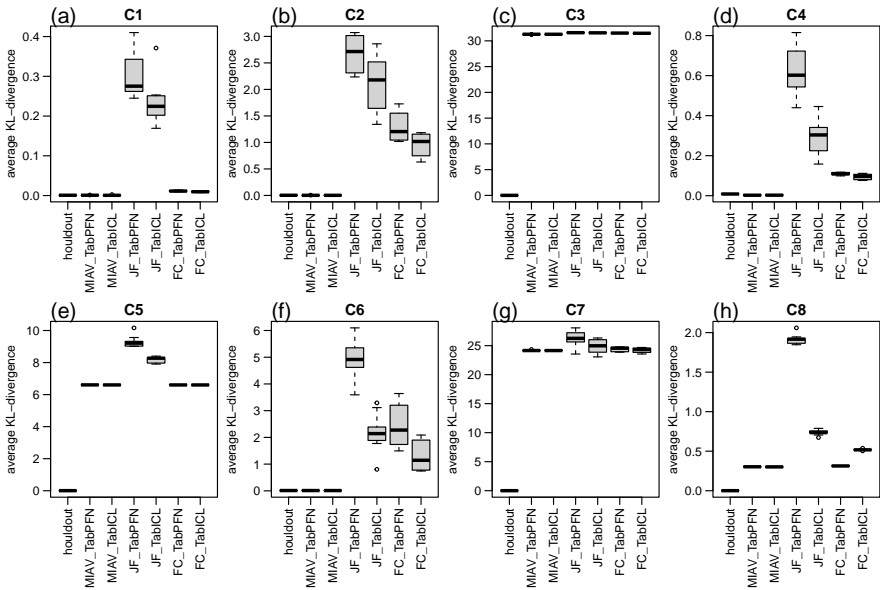

Figure 23: KL-divergence comparison for categorical datasets C1 to C8 (see Table 8 for the original dataset names). Each boxplot displays the results from 10 distinct original/holdout data splits.

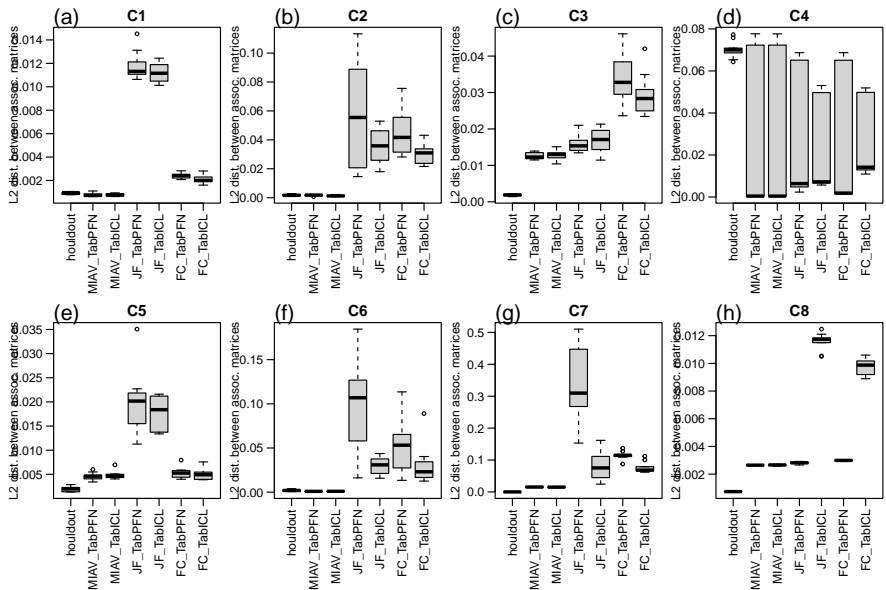

Figure 24: L2D comparison for categorical datasets C1 to C8 (see Table 8 for the original dataset names). Each boxplot displays the results from 10 distinct original/holdout data splits.

## L  INSENSITIVITY TO THE CHOICE OF THE RANDOM NOISE DISTRIBUTION

In all experiments reported in this paper, we construct MIAV variables with random noise sampled from a uniform distribution. A natural question is whether the MIAV approach is sensitive to the choice of the distribution used the generate the MIAV variable. Here, we clarify that the MIAV approach based on the TabPFN model is insensitive to the choice of noise distribution because, internally, TabPFN pre-process all features to approximately standard normal distributions before running them through the transformer. (As described in the Methods section Hollmann et al. (2025), the neural network of TabPFN expects approximately normal features after all pre-processing steps. To this end, for each input, TabPFN employs a Yeo-Johnson power transformation to stabilize variance and make the distributions approximately normal, followed by a z-transformation to center the inputs at 0 and scale their variance to 1.) As a consequence, the choice of noise distribution does not have an impact on the performance of the MIAV approach since the MIAV input variable is internally transformed to approximate a standard normal distribution.

To illustrate this point we compare the performance of the MIAV approach implemented with different random noise distributions including uniform ($U(0, 1)$), gaussian ($N(0, 1)$), and exponential ($Exp(1)$) noise. We again simulate data from correlated beta distributions and evaluate qualitatively the quality of the MIAV-based synthetic data generated with different random noise distributions. (In all these illustrations we use the same original data. Only the synthetic datasets generated with the MIAV approach differ.). Figure 25 reports the marginal distributions of the original data (black densities) and their respective MIAVs (red densities) (panels a to e), alongside scatter-plots of the original and MIAV variables (panels f to j), for synthetic data generated with MIAVs following a uniform distribution. Figures 26 and 27 report analogous comparisons for synthetic data generated with MIAVs following gaussian and exponential distributions.

Figure 28 reports the marginal densities of the original data (black) and the synthetic data (red) generated with uniformly distributed MIAVs. The figure shows the results from 3 separate replications based on different random seeds, where the top, middle, and bottom panels report the results generated with distinct random seeds. Figures 29 and 30 report the analogous results for synthetic data generated with gaussian and exponential random noise distributions, respectively. Comparison of Figures 28, 29, and 30 shows that the quality of the synthetic data remains unchanged with the different choices of random noise distributions.

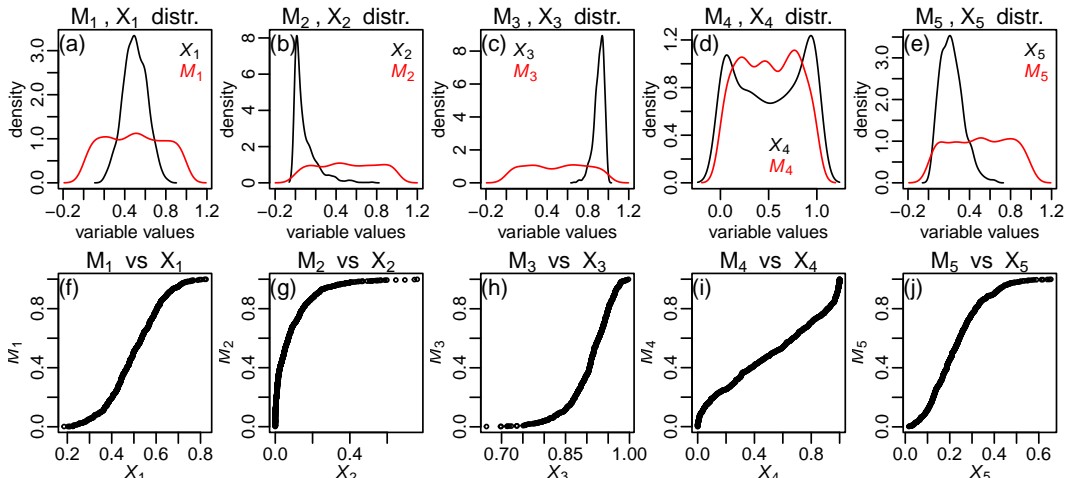

Figure 25: Illustrative example with MIAVs generated with uniform random noise.

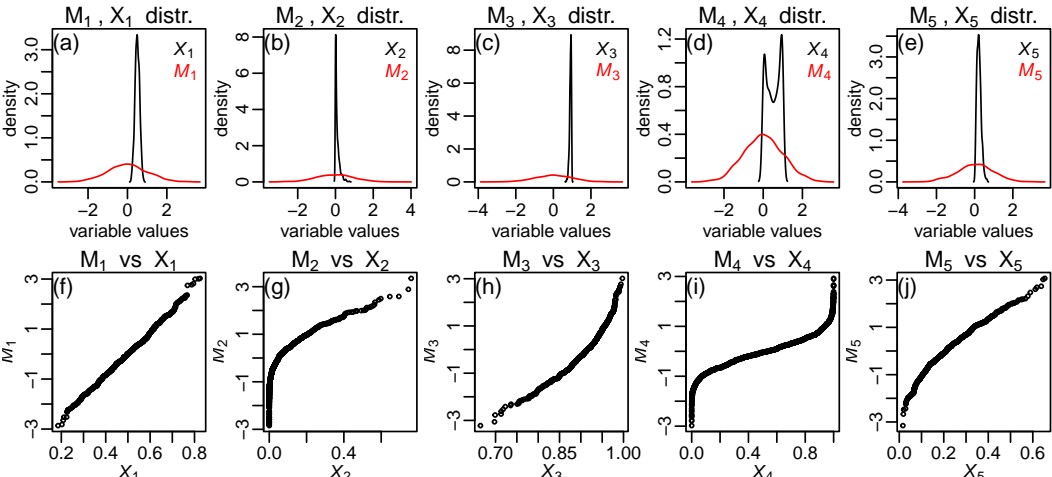

Figure 26: Illustrative example with MIAVs generated with gaussian random noise.

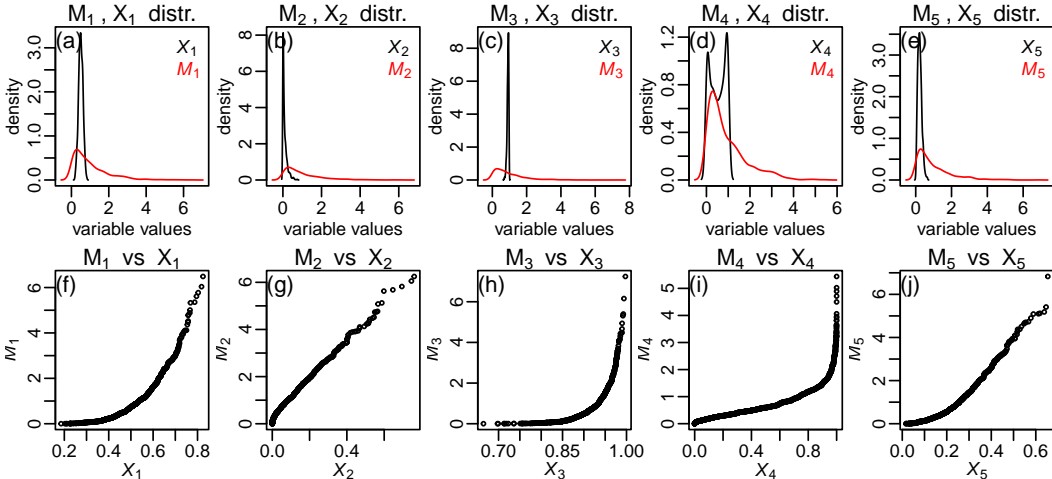

Figure 27: Illustrative example with MIAVs generated with exponential random noise.

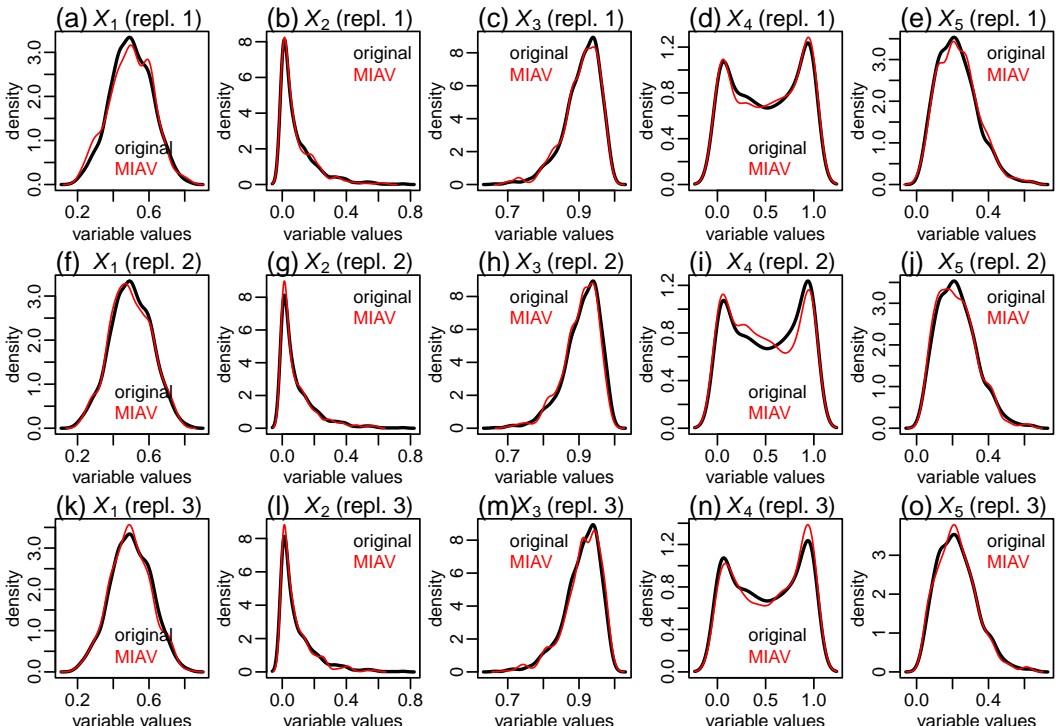

Figure 28: Marginal distributions of the original data (black densities) and the synthetic data (red densities) generated with uniformly distributed MIAVs. The top, middle, and bottom panels report results generated with different random seeds.

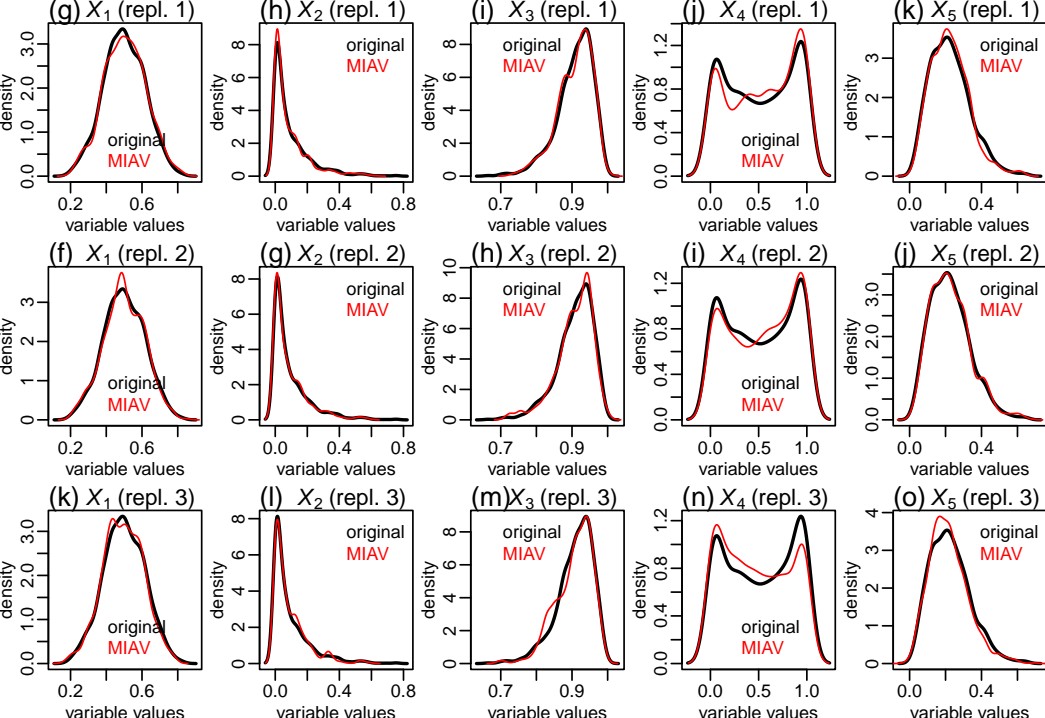

Figure 29: Marginal distributions of the original data (black densities) and the synthetic data (red densities) generated with normally distributed MIAVs. The top, middle, and bottom panels report results generated with different random seeds.

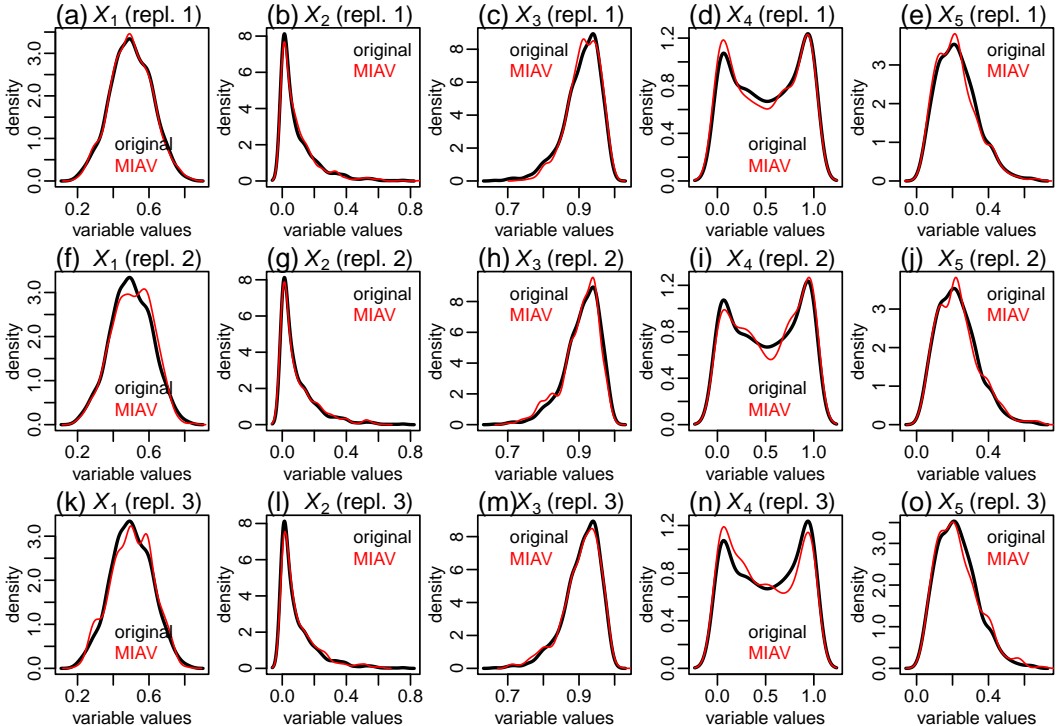

Figure 30: Marginal distributions of the original data (black densities) and the synthetic data (red densities) generated with exponentially distributed MIAVs. The top, middle, and bottom panels report results generated with different random seeds.

## M LLM USAGE

ChatGPT-5 was used to refine the grammar and clarity of some paragraphs of the paper and to translate R code into Python code. We reviewed and verified all AI-generated content for accuracy and take full responsibility for the paper's final content.

