# OpenReview forum: "Using  maximal information auxiliary variables to improve synthetic data generation based on TabPFN foundation models"
_ICLR.cc/2026/Conference — ICLR 2026 Poster_

### Official Review · Reviewer_PSBh · 2025-10-31

**Soundness:** 2
**Presentation:** 1
**Contribution:** 2
**Rating:** 2
**Confidence:** 4

**Summary:**

The paper highlights a limitation of current tabular foundations models (and in particular of TabPFN): namely, it does not perform in features that are poorly correlated with the others.

To overcome this limitation they propose to leverage maximal information auxiliary variables (MIAV). These variables are build from random noise and retain the information about how the datapoints are ordered (from lowest to highest in case of continuous features) with respect to the value of a feature. The authors show that following this construction, each variable $X_j$ conditional on its MIAV is independent of all other variables and the MIAV retains maximal information about $X_j$ in a non-conditional way.

**Strengths:**

The paper identifies a limitation of a well established model.

The authors propose a sensible solution that seems to work.

**Weaknesses:**

1) The paper needs a lot of rewriting to make it clearer and more structured. I give some pointers to some improvements below:
   - The section Notation and Related Work should be two sections: it is not clear why this is a single one
   - Calling vectors of random variables in slanted boldface and matrices as simple boldface makes it difficult to distinguish between the two of them (the space of the matrices of size $mxn$ is isomorphic wrt to the space of vectors of length $mn$, why not treating everything as vectors for example? it might help wrt the notation_
   - In the notation the symbol $X_{-j}$ is used at line 119 but only introduced at line 122
   - In $q_\theta(x_j^{ts} \mid X_{-j}^ts, X^tr)$ it would be clearer if you just define it as $q_\theta(x_j^{ts} \mid X_{-j}, X^tr)$ as you are removing the feature $j$ from both the training at testing
   - $p$ in equation (1) is not defined (I guess it is the number of features)
   - equation (2), the small $x$ might be missing the $\hat$
   - page 4 the authors talk about the permutation sampling approximation of Janossy pooling: explaining what this means takes 2 lines and would make the paper more accessible
   - the experimental analysis is just a big wall of text right now and could use some structuring: the authors could consider adding paragraphs titled "Metrics", "Experimental Setup","Datasets" etc.

2) It is not clear to me why the authors are sorting in line 2 of the algorithm if then the noise needs anyway to be sorted on the ground of the values of the features

3) in both the algorithm and the theorem the a variable $Z$ and vector $z$ appear, which I think are used as $M$ and $m$ but they are not defined anywhere

4) it is not clear how to read the colours in the charts in the last column of Figure 3

5) At page 6 the authors write that $X_j$ is *completely determined* by $M_j$. I am not sure how that is possible if $M_j$ is essentially just modelling the ordering

6) In page 6 Equation (4) I do not understand why the authors suddenly use $X_{<j}$ - I thought they were just considering $X_{-j}$.

7) In the experimental analysis  is missing  common metrics used in tabular data generation like utility and sample generation time. Also, in order to measure the statistical fidelity why didn't the authors use the Wasserstein distance for continuous features? That ia very commonly used.

8) The authors did not consider more recent baselines like GReaT which are transformer based.

Overall, I believe the paper presents an interesting solution but it needs more work before publication.

**Questions:**

See weaknesses

---

> ### Author Response · Authors · 2025-11-22
> **Response to Reviewer PSBh - part 1 (out of 3)**
>
> Thank you for your detailed feedback and constructive comments. We have addressed all of them and made extensive improvements in the manuscript presentation based on it. We uploaded an updated version of the manuscript with changes highlighted in red. See below a point-by-point response to all your comments.
>
> > 1.The paper needs a lot of rewriting …
>
> > The section Notation and Related Work should be two sections: it is not clear why this is a single one
>
> We had a single section before for the sake of space (splitting it in two caused the manuscript to go over the 9-page limit of the initial submission). In the updated manuscript, we split this section into the “Notation” and the “Related work” sections (see page 3 of the updated manuscript).
>
>
>
> > Calling vectors of random variables in slanted boldface and matrices as simple boldface makes it difficult to distinguish between the two of them …
>
> We use standard statistical notation where: (1) vectors and matrices are represented in boldface, while scalars are represented in regular font; and (2) random variables are represented in uppercase italics while instantiations of random variables (i.e., the values that the random variable assume) are represented in lower case italics. (That is the reason we represent vectors of random variables in boldface italics.)
>
> In any case, note that an easy way to distinguish between vectors of random variables and data matrices is to also note that we always use the $P()$ notation in probability statements involving random variables and the $q_\theta()$ notation in statements involving transformer models trained on data matrices and vectors. We have now added a sentence to the Notation section clarifying this point. (See line 116 of the updated manuscript.)
>
>
>
> > In the notation the symbol $X_{-j}$ is used at line 119 but only introduced at line 122
>
> Thanks for pointing this out. In the updated manuscript the symbol is first introduced in line 119 before being used in line 122.
>
>
>
> > In $q_\theta(x_j^{ts} \mid X_{-j}^{ts}, X^{tr})$ it would be clearer if you just define it as as $q_\theta(x_j^{ts} \mid X_{-j}, X^{tr})$ as you are removing the feature $j$ from both the training at testing
>
> We actually think it would be more confusing since we only remove feature $j$ from the test set, but not from the training set.
>
> Note that when we use the TabPFN model to generate a probabilistic prediction of $x_j^{ts}$ we actually engage in two steps:
>
> 1. The ICL training step, where we give context to the model by showing training examples of feature values and the associated target values - that is, where we show the model both the features $X_{-j}^{tr}$ and targets $x_{j}^{tr}$ (or, equivalently, the full training data $X^{tr} = \\{ x_{j}^{tr}, X_{-j}^{tr} \\}$).
>
> 2. The ICL query step, where we show the model the values of the test set features, $X_{-j}^{ts}$, and ask the model to generate a prediction for the test set target vector.
>
> Therefore, the prediction of $x_{j}^{ts}$ is generated conditional on $X_{-j}^{ts}$ and  $X^{tr}$, and the notation $q_\theta(x_j^{ts} \mid X_{-j}^{ts}, X^{tr})$ makes this clear.
>
> We now expanded the explanation of the notation around the TabPFN model in lines 124 to 128 of the updated manuscript.
>
>
>
> > p in equation (1) is not defined (I guess it is the number of features)
>
> Yes, $p$ represents the number of features.  We now added a sentence following equation (1) clarifying this point (see line 162 of the updated manuscript).
>
>
>
> > equation (2), the small x might be missing the \hat
>
> In our notation, instead of having the prediction $\hat{x}$ inside the transformer model $q_{\theta}()$ we refer to predictions generated by the model using the notation $\hat{x} \sim q_{\theta}()$. This is now clarified in line 123 of the updated manuscript.
>
>
>
> > page 4 the authors talk about the permutation sampling approximation of Janossy pooling: explaining what this means takes 2 lines and would make the paper more accessible
>
> We have now added the following footnote, clarifying this point (see lines 214 to 215 of the updated manuscript):
>
> “Namely, Hollmann et al. (2025) generate $N$ distinct synthetic datasets, using different random permutations of the order of the variables during the synthetic data generation process, and average the results across the $N$ synthetic datasets to reduce variability and decrease the dependence of the result on the variable order.”
>
>
>
> > the experimental analysis is just a big wall of text right now and could use some structuring: the authors could consider adding paragraphs titled "Metrics", "Experimental Setup","Datasets" etc.
>
> This is a good point. We have now organized this section in a structured format with paragraphs titled: “Experiments”, “Data splits”, “Evaluation metrics”, “Baselines”, “Experimental details” and a subsection titled “Results”. (See pages 8 and 9 of the updated manuscript.)

---

> > ### Author Response · Authors · 2025-11-22
> > **Response to Reviewer PSBh - part 2 (out of 3)**
> >
> > > 2. It is not clear to me why the authors are sorting in line 2 of the algorithm if then the noise needs anyway to be sorted on the ground of the values of the features
> >
> > This is done for computational convenience. We sort the noise in line 2 of Algorithm 1 because then we can simply use the ranks of the feature data as the indexes of the sorted noise data in order to match the ranks of the noise data to the ranks of the feature data (as implemented in line 9).
> >
> > As a toy example, suppose that the feature data is given by $x_j = (0.32, 3.13, -1.87)^t$, and the sorted random noise data (computed in line 2) is given by $m_j = (0.22, 0.67, 0.79)^t$. The ranks of $x_j$ (computed in line 4) are then given by $r = (2, 3, 1)$, and the reordering of the entries of the sorted random noise vector indexed according to the ranks $r$ (computed in line 9), is given by $m_j = m_j[r] = (0.67, 0.79, 0.22)^t$ (where (0.22, 0.67, 0.79)[2] = 0.67, (0.22, 0.67, 0.79)[3] = 0.79, and (0.22, 0.67, 0.79)[1] = 0.22).  Note that the $x_j = (0.32, 3.13, -1.87)^t$ and $m_j = (0.67, 0.79, 0.22)^t$ vectors are rank-matched because the smallest elements of both vectors are located in the third entries of the vectors, the second smallest elements are located in the first entries, and the largest elements are located in the second entries.
> >
> >
> >
> > > 3. in both the algorithm and the theorem the variable Z and vector z appear, which I think are used as M and m but they are not defined anywhere
> >
> > This is a typo. Thanks for catching it. We now changed $z$ to $m$ and $Z$ to $M$ in lines 279 and 318 of the updated manuscript.
> >
> >
> >
> > > 4.  it is not clear how to read the colours in the charts in the last column of Figure 3
> >
> > Thanks for catching this. We added the following sentence to the caption of Figure 3: “In panels f and l, positive correlations are represented in blue and negative correlations are represented in red.” We also added a similar explanation to the caption of Figure 5 in the Appendix, which also presents correlation plots.
> >
> >
> >
> > > 5.  At page 6 the authors write that $X_j$ is completely determined by $M_j$. I am not sure how that is possible if $M_j$ is essentially just modelling the ordering
> >
> > We meant it in the context of Theorem 1, where $M_j$ completely determines $X_j$ in a non-parametric rank-based sense. We now clarify this point in line 329 of the updated manuscript.
> >
> >
> >
> > > 6.  In page 6 Equation (4) I do not understand why the authors suddenly use $X_{< j}$ - I thought they were just considering $X_{-j}$
> >
> > No, we consider different notations. We apologize the Notation section originally only described the $X_{-j}$ notation. (The $X_{< j}$ notation was only defined in Section 3 immediately after equation 1.) We have now fixed this issue.  The updated Notation section now states that (see lines 119 to 121 of the updated manuscript):
> >
> > “We use the notation $X_{-j}$ to represent the matrix obtained by removing the $j$th column from $X$, and the notation $X_{<j}$ to represent the matrix comprised by the first $j-1$ columns of $X$.”
> >
> > In the case of equation 4, the $X_{< j}$ notation provides a simple way to express the factorization of a multivariate joint probability distribution into a product of conditional probability distributions. For instance, in the case where $p = 3$ (and letting $\boldsymbol{C} = \\{ X^{tr}, M^{ts}, M^{tr} \\}$ to further simply the notation) we have that,
> >
> > $$
> > P(\boldsymbol{X}^{ts} \mid \boldsymbol{C}) = P(X_1^{ts}, X_2^{ts}, X_3^{ts} \mid \boldsymbol{C}) = P(X_1^{ts} \mid \boldsymbol{C}) P(X_2^{ts} | X_1^{ts}, \boldsymbol{C}) P(X_3^{ts} | X_1^{ts}, X_2^{ts}, \boldsymbol{C}) = P(X_1^{ts} | X_{<1}^{ts}, \boldsymbol{C}) P(X_2^{ts} | X_{<2}^{ts}, \boldsymbol{C}) P(X_3^{ts} \mid X_{<3}^{ts}, \boldsymbol{C})
> > $$
> >
> > since for $p=3$ we have that $X_{<1} = \\{ \\}$, $X_{<2} = \\{ X_1 \\}$, and $X_{<3} = \\{ X_1, X_2 \\}$.

---

> > > ### Author Response · Authors · 2025-11-22
> > > **Response to Reviewer PSBh - part 3 (out of 3)**
> > >
> > > > 7. In the experimental analysis is missing common metrics used in tabular data generation like utility and sample generation time. Also, in order to measure the statistical fidelity why didn't the authors use the Wasserstein distance for continuous features? That ia very commonly used.
> > >
> > > As suggested by the Reviewer, we now include the machine learning efficiency (MLE) metric for measuring data utility in downstream prediction tasks and the Wasserstein distance metric for measuring data fidelity with respect to the joint distribution of the data. (We also included the energy distance metric, which is closely related to the maximum mean discrepancy, as another metric for evaluating joint distribution fidelity.)
> > >
> > > We did not compute sample generation time because that would require re-running all our extensive experiments again. (Recall that in addition to running experiments on simulated data across 10 replications over 5 distinct simulation settings, we performed evaluations of 43 real world datasets across 10 distinct random data splits.) On the other hand, the computation of the MLE, Wasserstein distance and energy distance did not require re-runs because we had saved all the synthetic dataset outputs generated in our original experiments.
> > >
> > > The evaluation results for these 3 new metrics are presented in Figure 13 and discussed in detail in a new appendix section (Appendix I.6) located in pages 35 and 36 of the updated manuscript.
> > >
> > > Overall, these new evaluations show that the MIAV approach is again competitive with respect to these new metrics, ranking among the top generators in most experiments. (These evaluations also showed some surprising results for other baselines, such as strong performance of Bayesian networks in terms of Wasserstein and energy distances. Please, refer to Appendix I.6 for further details.)
> > >
> > > Descriptions of the new metrics and their implementations are presented in lines 1785 to 1791 (for the MLE metric) and lines 1812 to 1827 (for the Wasserstein distance and energy distance) on the updated manuscript.
> > >
> > > We refer to these additional metric results in the main text in lines 472 to 476 of the updated manuscript.
> > >
> > >
> > > > 8. The authors did not consider more recent baselines like GReaT which are transformer based.
> > >
> > > Although GReaT is among the most computationally expensive baselines, and requires substantial hyperparameter tuning, it is not especially competitive relative to more lightweight generative models. Multiple benchmarking studies report that GReaT is consistently outperformed by newer diffusion-based approaches, including TabDDPM (which we include as a baseline in our work); see references [1], [2], [3], and [4] below. Consequently, even though we do not include GReaT itself, our evaluation already incorporates a stronger and more recent baseline.
> > >
> > > Finally, please note that the primary goal of our manuscript is not to introduce a new state-of-the-art model for tabular synthetic data generation. Rather, our main contribution is a framework for synthetic data generation that aligns with the emerging paradigm in machine learning which increasingly centers on foundation models and in-context learning. The comparisons with established baselines are really only meant to illustrate that the MIAV approach is already competitive with popular baselines developed under the traditional bespoke-model paradigm.
> > >
> > > References:
> > >
> > > [1] Zhang et al. (2024). Mixed-type tabular data synthesis with score-based diffusion in latent space. ICLR 2024.
> > >
> > > [2] Kindji et al. (2024). Under the hood of tabular data generation models: benchmarks with extensive tuning. arXiv:2406.12945
> > >
> > > [3] Du and Li (2024). Towards principled assessment of tabular data synthesis algorithms. arXiv:2402.06806 (2024).
> > >
> > > [4] Shi et al. (2025). TabDiff: a mixed-type diffusion model for tabular data generation. ICLR 2025.

---

> > > > ### Comment · Reviewer_PSBh · 2025-11-27
> > > >
> > > > The presentation of the paper has greatly improved. My score will hence reflect that.
> > > >
> > > > For presentation sake, would the authors mind moving the figures on top of every page? Right now they are bundled together with the text

---

> ### Author Response · Authors · 2025-11-27
> **Response to additional suggestions by Reviewer PSBh**
>
> Thank you again for the constructive suggestions and positive feedback to our response. We have uploaded a new version of the manuscript where all the figures (and Algorithm 1) in the main text were moved to the top of the respective pages.

---

### Official Review · Reviewer_DDxA · 2025-10-31

**Soundness:** 4
**Presentation:** 4
**Contribution:** 4
**Rating:** 8
**Confidence:** 3

**Summary:**

The paper identifies a problem with using tabular foundation models like TabPFN to generate tabular data, viz., that their performance degrades significantly when variables have weak statistical associations. The paper then proposes a novel method called the Maximal Information Auxiliary Variable (MIAV) strategy to deal with this problem. The MIAV strategy creates an auxiliary variable for every variable through a rank-matching procedure between the real data and a random noise vector. The core insight is that this auxiliary variable is perfect (in the rank-based sense) predictor of the corresponding real variable. A concise theoretical justification for MIAV is provided and its practical benefits, including computational efficiency (reducing complexity from cubic to linear in the number of features) and invariance to column order are demonstrated through extensive empirical evaluations.

**Strengths:**

+ The paper clearly identifies the "weak associations" problem.
+ The MIAV strategy is simple and effective.
+ The strategy reduces computational complexity, which is a practically significant contribution.
+ The Theorem 1 provides strong theoretical grounding for the MIAV strategy. It is also well supported by the extensive evaluations.

**Weaknesses:**

- The MIAV strategy is an "add on" in the sense that its performance of the overall generator  will  always be limited by the underlying model (TabPFN, TabICL etc.). This could be more explictly stated as a limitation
- The method requires generating MIAV for the entire dataset, including the test set. This could cause information leakage if someone were to apply this method for prediction. Although, footnote 1 on page 7 clarifies that this meant for generation not classification task, it should probably be stated even more emphatically in the paper, to prevent misapplication of this method.

**Questions:**

1. A uniform distribution is used as the random noise generator for the rank-matching. What happens if the distribution were changed, say to a Gaussian? How sensitive is the method to this choice of distribution?
2. SMOTE seems to be performing well on  fidelity metrics alone, although MIAV is often better when fidelity-privacy trade-off is considered. Can you elaborate on when a practitioner would prefer MIAV over the much simpler SMOTE?

---

> ### Author Response · Authors · 2025-11-22
> **Response to Reviewer DDxA - part 1 (out of 2)**
>
> Thank you for your positive feedback, thoughtful questions, and constructive comments. We addressed them all and incorporated your feedback in the manuscript. Please, see below our point-by-point response to all your questions/comments.  We have also uploaded an updated version of the manuscript with changes highlighted in red.
>
> > A uniform distribution is used as the random noise generator for the rank-matching. What happens if the distribution were changed, say to a Gaussian? How sensitive is the method to this choice of distribution?
>
> The method is not sensitive to the choice of noise distribution because, internally, TabPFN pre-process all features to approximately standard normal distributions before running them through the transformer. (As described in the Methods section Hollmann et al. (2025), the neural network of TabPFN expects approximately normal features after all pre-processing steps. To this end, for each input, TabPFN employs a Yeo-Johnson power transformation to stabilize variance and make the distributions approximately normal, followed by a z-transformation to center the inputs at 0 and scale their variance at 1.)  Consequently, the choice of noise distribution does not have an impact on the performance of the MIAV approach since the MIAV input variable is internally transformed to approximate a standard normal distribution.
>
> To illustrate this point we compare the performance of the MIAV approach implemented with different random noise distributions including uniform, gaussian, and exponential noise. These comparisons are reported in the new Appendix L section (located in pages 47 to 50 of the updated manuscript). As shown in Figures 28, 29, and 30 (which report the marginal distributions of the original and synthetic data generated with uniform, gaussian, and exponential random noise, respectively) the quality of the synthetic data remains unchanged with the choice of random noise distribution.
>
> We also added one sentence in the main text (lines 268 to 269) clarifying that the choice of noise distribution is not important and pointing to Appendix L for details.
>
>
> > SMOTE seems to be performing well on fidelity metrics alone, although MIAV is often better when fidelity-privacy trade-off is considered. Can you elaborate on when a practitioner would prefer MIAV over the much simpler SMOTE?
>
> Yes, we believe that it is becoming common knowledge among the tabular synthetic data generation community that SMOTE consistently ranks among the top generators in terms of data fidelity at the expense of decreased data privacy. In practice, this means that SMOTE can be a very good generator option in applications that are not privacy sensitive, where high data fidelity is the main goal. However, for privacy sensitive applications (e.g., health data) we believe the MIAV approach provides a better option as it strikes a better fidelity-privacy trade-off.
>
>
> > The method requires generating MIAV for the entire dataset, including the test set. This could cause information leakage if someone were to apply this method for prediction. Although, footnote 1 on page 7 clarifies that this meant for generation not classification task, it should probably be stated even more emphatically in the paper, to prevent misapplication of this method.
>
> Yes, this is very good point. Thanks for bringing this up. We previously only touched this point in passing in footnote 1 and in a short paragraph in the “Final remarks” section. As suggested by the reviewer, we have now expanded on this point in an extended paragraph closing the manuscript (see lines 523 to 533 of the updated manuscript):
>
> “Finally, we point out that the MIAV strategy described here is really only intended for synthetic data generation and should not be used for improving predictive performance of supervised learners. As described in Section 5, generating MIAV variables requires unrestricted access to the full dataset $\mathbf{X}$, which is partitioned into $\mathbf{X}^{tr}$ and $\mathbf{X}^{ts}$ and used to construct the corresponding MIAV matrices $\mathbf{M}^{tr}$ and $\mathbf{M}^{ts}$. Because MIAVs must be computed on the test set, the approach is inherently incompatible with supervised learning scenarios where test-set targets are unavailable. But, more importantly, it should never be used to enhance supervised learning performance in settings where the full dataset is merely split into training and test subsets for evaluation purposes. In such cases, the generation of the MIAV variable associated with the test set target would leak information about the target into the associated MIAV, and inclusion of this MIAV as an input in a supervised model would lead to an artificial boost in predictive performance due to data leakage.”

---

> > ### Author Response · Authors · 2025-11-22
> > **Response to Reviewer DDxA - part 2 (out of 2)**
> >
> > > The MIAV strategy is an "add on" in the sense that its performance of the overall generator will always be limited by the underlying model (TabPFN, TabICL etc.). This could be more explictly stated as a limitation
> >
> > Yes, we agree that this is a limitation of the MIAV approach, and we have already stated this explicitly in lines 471 to 477 of the original manuscript, reproduced below:
> >
> > “Our approach inevitably inherits the limitations of the underlying tabular foundation model used for in-context learning. For TabPFN, these limitations include: …”
> >
> > But, on this topic, we recently learned that a new version of the TabPFN model, TabPFN-2.5, has been very recently released (a couple of weeks from the time of the writing of this response). The technical report [1] describing this new model states that: (i) it is able to handle datasets with up to 50,000 rows and 2,000 columns (a large scaling up compared to the TabPFNv2 model used in our experiments, which is limited to datasets containing up to 10,000 rows and 500 columns); (ii) it achieves 1x to 2.3x speedup in inference time; and (iii) it also achieves better performance in supervised learning tasks.
> >
> > While we have not evaluated this new model (and re-running our extensive experimental evaluations with this new model would certainly be infeasible at this point) we feel that this model could in principle already relax to a considerable degree some of limitations of our work.
> >
> > We have now expanded our limitation paragraph as follows (see lines 502 to 513 of the updated manuscript):
> >
> > “Our approach inevitably inherits the limitations of the underlying tabular foundation model used for in-context learning. For the TabPFN model used in our experiments (namely, TabPFNv2), these limitations include: (i) a maximum data size of 10,000 rows; (ii) memory usage that grows linearly with dataset size, which can become prohibitive for very large data; and (iii) inference speeds that may lag behind alternative baselines. But, as mentioned above, these are early days in the development of PFN-based tabular foundation models and we expect that future releases will likely continue to relax limitations from the previous versions. For instance, a new version of the TabPFN model, denoted TabPFN-2.5, has been recently released which is able to handle datasets with up to 50,000 rows (Grinsztajn et al., 2025). Furthermore, the more scalable TabICL model is already able to handle 500,000 rows but currently supports only classification tasks. Future versions of TabICL that extend to regression could be directly integrated with the MIAV strategy, thereby helping to overcome current model constraints.”
> >
> > Also, we have now added clarification that while PFN-based tabular foundation models also have limitations regarding the number of columns they can handle, this is not a constraint for the MIAV approach, as it relies on a single feature. Lines 514 to 519 of the updated manuscript now state that:
> >
> > “In addition to limitations on the maximum number of rows they can process, PFN-based tabular foundation models are also constrained by the number of columns they can handle. For example, TabPFNv2 and TabICL support datasets with up to 500 features, while TabPFN-2.5 increases this limit to 2,000. Importantly, however, these constraints do not affect the MIAV approach: because MIAV requires training PFN-based models using only a single feature per variable, it can be applied to datasets containing more columns than the column number limit of the underlying PFN model.”
> >
> > References:
> >
> > [1] Grinsztajn et al. (2025). TabPFN-2.5: Advancing the State of the Art in Tabular Foundation Models.

---

> > ### Comment · Reviewer_DDxA · 2025-11-25
> >
> > I am satisfied with the answers to my questions and the response to my review. I have no more questions.

---

### Official Review · Reviewer_zUsD · 2025-10-31

**Soundness:** 3
**Presentation:** 3
**Contribution:** 3
**Rating:** 6
**Confidence:** 4

**Summary:**

The paper proposes an approach to improve tabular data generation models like TabPFN on the synthetic data generation task. Specifically, the authors develop a maximal information auxiliary variable (MIAV) based framework capable of producing accurate synthetic data even in the context of variables with weak associations, overcoming the scalability issues of TabPFN. Experiments performed on numerous synthetic and real-world datasets demonstrate that the proposed MIAV conditioned TabPFN model is comparable to or better than state-of-the-art synthetic data generation approaches. Overall, the paper develops a novel method that is consistent with the claims made in the paper and will contribute to the development of efficient, high-quality synthetic data generation for tabular data especially with weakly associated and small datasets.

**Strengths:**

1. The paper is well written and easy to follow. Further, the problem of synthetic data generation of tabular data comprising continuous and categorical variables, that is being tackled, is of prime importance.

2. The proposed MIAV method is simple yet effective in accounting for weak associations along with improving generation scalability. The method is capable of being applied to both continuous and categorical variables.

3. The paper also develops theory to support its claims of improved scalability and demonstrates state-of-the-art results on numerous synthetic and real-world data compared with multiple baselines.

**Weaknesses:**

1. Out-of-Distribution generalization is alluded to the the introduction but it is unclear if it has been demonstrated in the paper.

2. Run-time experiments have been run on small synthetic datasets (e.g., <= 3000 samples).

**Questions:**

1. How does the proposed model perform in the context of out-of-distribution synthetic-data generation task compared to TabPFN, especially owing to its ability to perform well in the context of variables with weak associations?

2. Scalability is an important aspect of such models and this is somewhat of a concern for me at the moment. Why have synthetic datasets only been generated with <=3000 instances in the scenario evaluating runtime analysis? Is it difficult to scale the model over these to truly large scale data (e.g., 10000 rows or greater)?  If scaling is difficult, what are the bottlenecks?

3. How does the current proposed approach scale with the number of columns in the datasets?

---

> ### Author Response · Authors · 2025-11-22
> **Response to Reviewer zUsD - part 1 (out of 2)**
>
> Thank you for your positive feedback, your thoughtful questions, and constructive comments. We addressed them all and incorporated your feedback in the manuscript. Please, see below our point-by-point response to all your questions/comments.  We also uploaded an updated version of the manuscript with changes highlighted in red.
>
> > Scalability is an important aspect of such models and this is somewhat of a concern for me at the moment. Why have synthetic datasets only been generated with <=3000 instances in the scenario evaluating runtime analysis? Is it difficult to scale the model over these to truly large scale data (e.g., 10000 rows or greater)? If scaling is difficult, what are the bottlenecks?
>
> We only kept the runtime analysis evaluations under 3000 or less examples to save computing time. Scaling up to larger datasets is not an issue, as long as you don’t go over the size limitations of the underlying PFN model (and you have access to GPUs). All the experiments in our manuscript were based on the TabPFNv2 model which can handle datasets with up to 10,000 rows and 500 columns. (Note, however, that only the maximum number of rows that a PFN model can handle represents a limitation to the MIAV approach. The maximum number of columns is immaterial since the MIAV data generation approach only uses a single column (i.e., the MIAV) when performing the in-context learning.)
>
> To illustrate this point we have now expanded our runtime experiments by including 4 additional experiments based on 4000, 5000, 10000, and 20000 instances. Note that 20000 is the highest we can go because the TabPFNv2 model can handle a maximum of 10000 instances, and the implementations of the MIAV, JF, and FC generators are based on two separate calls to the TabPFN model using approximately equal sized splits of the original data (see Algorithms 3, 5, and 7).
>
> The results from the expanded runtime experiments are now presented in Figures 11 and 12 of the updated manuscript (see page 30). Figure 11 report results for number of rows ($n$) varying from 1000 to 5000, while Figure 12 show the results for $n$ equal to 5000, 10000, and 20000.
>
> The bottom line is that the scalability of the MIAV approach will always be limited by the scalability of the underlying PFN model (with respect to number of rows). While this is a limitation, it is important to highlight that these are still early days in the development of PFN models for tabular data, and it is reasonable to expect that new and more scalable models will keep being introduced.  For instance, the TabICL model [1] introduced few months ago is much more scalable than TabPFNv2, being able to handle 500,000 rows (although its current version can only handle classification tasks, so that TabICL-based MIAV generators can only be applied to datasets containing only categorical variables). Furthermore, we have just learned that a new TabPFN model [2], TabPFN-2.5, has been released very recently (on November 6 of 2025) which is now able to handle up to 50,000 rows.
>
> We have included a more detailed discussion of the scalability of our approach in lines 506 to 513 of the updated manuscript.
>
> References:
>
> [1] Qu et al. (2025). TabICL: A Tabular Foundation Model for In-Context Learning on Large Data. ICML 2025.
>
> [2] Grinsztajn et al. (2025). TabPFN-2.5: Advancing the State of the Art in Tabular Foundation Models.
>
>
> > How does the current proposed approach scale with the number of columns in the datasets?
>
> It scales linearly with the number of columns (features) in the datasets. This point is illustrated by the red curves in Figures 11 and 12 (which report the results for the MIAV approach).

---

> ### Author Response · Authors · 2025-11-22
> **Response to Reviewer zUsD - part 2 (out of 2)**
>
> > How does the proposed model perform in the context of out-of-distribution synthetic-data generation task compared to TabPFN, especially owing to its ability to perform well in the context of variables with weak associations?
>
> This is a very interesting (and difficult) question.
>
> To evaluate out-of-distribution generalization (OOD generalization) in the context of synthetic data generation (SDG), we would need to run more elaborate experiments, using a non-representative sample from the true underlying population distribution as the training set, and checking whether the generator would still be able to draw samples consistent with the true population distribution (rather than generating samples consistent with the distribution of the non-representative training data).
>
> In the same way as stable prediction methods rely on causal information to achieve better OOD generalization in the context of predictive modeling, we believe that achieving OOD generalization in the context of SDG would also require causal information describing the data generation process underlying the real data. (One example of a synthetic data generator that leverages causal information is the DECAF [1] generator, which models and intervenes on the causal structural model underlying the real data generation process, in order to generate unbiased/fair synthetic data from biased/unfair training data.)
>
> But our method (as most of the generators in the literature) does not rely on causal information and is not really designed to achieve OOD generalization. (The goal is to generate synthetic copies of the training data which preserve the statistical properties of the training data while mitigating privacy risks – in other words, the goal is to generate independent samples from the same distribution as the training set.) Hence, we would not expect that our method (or any other generator that does not leverage causal information) would be able to achieve strong OOD generalization.
>
> That being said, an interesting research question (which is nonetheless out of the scope of the present manuscript) would be to evaluate the performance of our approach relative to other bespoke ML generators. Even though we would not expect strong performance from any of these methods, it would be interesting to check if our proposed method would be relatively more robust to distribution shifts than bespoke ML generators, given that it leverages tabular foundation models for generating predictions. (This might be a reasonable hypothesis since, at least in the context of supervised learning, foundation models tend to perform better than bespoke ML models in terms of OOD generalization.)
>
> But, while this is a very interesting question, none of our experiments in our manuscript were conducted in a way that could be used to answer this question.
>
> References:
>
> [1] van Breugel et al. (2021). DECAF: generating fair synthetic data using causality-aware generative networks. NeurIPS 2021.
>
>
>
> > Out-of-Distribution generalization is alluded to in the introduction but it is unclear if it has been demonstrated in the paper.
>
> Yes, the reviewer is correct. As explained above, our manuscript does not demonstrate OOD generalization.
>
> Although it was never our intent to suggest that our method achieves OOD generalization, we recognize that our brief mention of it in the introduction may have given readers that impression, and we understand why the reviewer raised this concern.
>
> To eliminate the risk of further confusion, we no longer mention OOD generalization in the introduction. We now revised the second paragraph (the only place where OOD generalization is mentioned) as follows (see lines 38 to 49 of the updated manuscript):
>
> “While synthetic data has long been explored through bespoke statistical models and machine learning algorithms, the field is now undergoing a paradigm shift driven by advances in large-scale, general-purpose models. Traditional approaches, such as those by Borisov et al. (2023), Cresswell and Kim (2024), Jolicoeur-Martineau et al. (2024), Kotelnikov et al. (2023), Nowok et al. (2016), Reiter (2005), Shi et al. (2025), Watson et al. (2023), Xu et al. (2019), Young et al. (2009), Zhang et al. (2024), Xu et al. (2025), among many others, typically rely on dataset-specific training, demand substantial domain expertise, and often struggle with knowledge transfer across datasets. Tabular foundation models (Hollmann et al., 2023; den Breejen et al., 2024; Koshil et al., 2024; Feuer et al., 2024; Ma et al., 2024a; Ma et al., 2024b; Xu et al., 2024; Zeng et al., 2024; Muller et al., 2025; Hollmann et al., 2025; Qu et al., 2025; Garg et al., 2025) offer a promising alternative. By learning broad, transferable representations of tabular data, they enable strong performance in supervised learning tasks with minimal additional training.”

---

> > ### Comment · Reviewer_zUsD · 2025-11-25
> > **Response to Authors**
> >
> > Thanks to the authors for their responses. The OOD question has been addressed, and my concerns regarding scalability have been addressed by 12(a) - (d) as data containing 20K rows and up to 100 features (somewhat more representative of real-world scale) have been successfully evaluated. I have no further questions and will raise my score to recommend acceptance.

---

### Official Review · Reviewer_4vKk · 2025-10-31

**Soundness:** 3
**Presentation:** 4
**Contribution:** 3
**Rating:** 8
**Confidence:** 4

**Summary:**

This paper focuses on tabular foundation models (like TabPFN) and explores the use of these models for synthetic data generation (as opposed to tabular classification). The authors claim that when such models are used in an ICL (in context learning) mode weak associations between columns can cause poor performance. As a result they propose a so-called maximal information auxiliary variable (MIAV) strategy, which enriches the model’s context using auxiliary variables that they derive through a rank-matching process.

**Strengths:**

- Focuses on synthetic data generation which appears to be understudied in the tabular foundation model space
- Generalizes across multiple tabular foundation models
- The proposed MIAV appears to be theoretically grounded

**Weaknesses:**

-  The proposed framework seems highly reminiscent of Bayesian structure discovery algorithms which also need to learn an ordering of variables to support better sampling (e.g., for MCMC methods). Can you comment on the relationships (or lack thereof) to that literature?
- The idea of creating auxiliary variables is good but it seems orthogonal to seeing whether a different ordering of variables will work. For instance, see this work: https://arxiv.org/abs/2406.14541. Can you comment on when we will need a different ordering vs when your approach is needed?

**Questions:**

- Please address the above questions from the Weaknesses section.

---

> ### Author Response · Authors · 2025-11-22
> **Response to Reviewer 4vKk**
>
> Thank you for your positive feedback and thoughtful questions. We address both of them below. We have also uploaded an updated version of the manuscript with changes highlighted in red.
>
> > The proposed framework seems highly reminiscent of Bayesian structure discovery algorithms which also need to learn an ordering of variables to support better sampling (e.g., for MCMC methods). Can you comment on the relationships (or lack thereof) to that literature?
>
> We do not see a direct connection between the proposed framework and Bayesian structure discovery algorithms (such as the MCMC samplers over variable orderings, rather than network structures, as proposed by [1]).
>
> Please, note that the proposed MIAV approach is invariant with respect to the ordering of the variables and, therefore, does not need to learn any particular ordering of the variables to support better sampling of synthetic data. (This is explicitly shown in equation 9 where, conditional on the MIAVs, the posterior predictive distribution of the tests set features factorizes into a product of independent terms.)
>
> Does this answer your question? (Please, let us know if we misunderstood it.)
>
> References:
>
> [1] Friedman and Koller (2003). Being Bayesian About Network Structure. A Bayesian Approach to Structure Discovery in Bayesian Networks. Machine Learning, 50, 95-125.
>
>
> > The idea of creating auxiliary variables is good but it seems orthogonal to seeing whether a different ordering of variables will work. For instance, see this work: https://arxiv.org/abs/2406.14541. Can you comment on when we will need a different ordering vs when your approach is needed?
>
> Thanks for bringing this work to our attention. We included it as a new reference - Xu et al. (2025) - in our manuscript (see line 43 of the updated manuscript).
>
> Back to your question, we see our approach more as an alternative to having to find an optimal ordering of the variables than as an orthogonal approach (as our MIAV approach and the work pointed by the reviewer – namely, the PAFT strategy proposed by Xu et al – could be both concomitantly applied across a wide range of settings).
>
> One setting where the PAFT approach can still be used while the MIAV approach cannot, is with datasets that surpass the data size limits that current TabPFN models can handle. But outside this setting, we believe that our approach might have a few practical advantages over PAFT.
>
> PAFT introduces interesting ideas on how to overcome some of the key limitations of tabular synthetic data generation based on LLMs. We agree with its main argument that, due to the autoregressive nature of LLMs, it is very important to set up the order of the variables in a way that respects the functional dependencies (FD) between the attributes in the dataset. And while Xu et al introduced a grounded approach for learning an optimal feature order from the data, the approach encompasses several steps including the learning of FDs using FD discovery algorithms, and the distillation of the FDs generated by the discovery algorithm into a coherent dependency graph. As described in Sections 4.2 and 4.3 of Xu et al, each of these steps are non-trivial to implement (and we would guess that the imprecise estimation of the dependency graph might have an impact on the performance of the generator).
>
> Our approach, on the other hand, is invariant with respect to the order of the features during the synthetic data generation process and, therefore, bypasses the need to infer an optimal variable order all together.
>
> Additionally, because our approach builds on a TabPFN foundation model rather than relying on fine-tuned LLMs (as in PAFT), it might also benefit from a few other practical advantages. For instance, generating synthetic data with TabPFN requires only a single forward pass over the TabPFN model neural network and no hyperparameter tuning, making it generally more lightweight than fine-tuning an LLM on text-encoded tables. TabPFN is particularly effective in low-data regimes, where LLM fine-tuning might become less reliable. Finally, since TabPFN is pretrained entirely on synthetically generated tabular datasets, it is not exposed to the kinds of data-contamination risks that can occasionally arise in LLM-based approaches trained on large corpora of real data.

---

### Author Response · Authors · 2025-11-22
**Overview of the updated manuscript**

We thank all the reviewers for the very thoughtful comments. Below we summarize our responses and provide a summary of the updates we made to the manuscript.

### **SUMMARY OF THE RESPONSES**

We replied to **every single one** of the questions, comments, and concerns raised by the reviewers, including:

1. Clarifications regarding notation, concepts, and algorithms.

2. Expanded discussions regarding limitations and scalability.

3. Improvements in the manuscript presentation.

4. The inclusion of additional experiments and analyses including: 4 additional runtime experiments based on larger scale datasets; a sensitivity analysis to the noise distribution used to construct the MIAVs; and the inclusion of 3 additional evaluation metrics in our experimental results.

All together we replied to 25 questions/comments/concerns.

### **SUMMARY OF UPDATES MADE TO THE MANUSCRIPT**

We have uploaded a revised manuscript that incorporates the reviewer’s feedback. All changes relative to the original version are highlighted in red. Below we provide a list of these revisions, organized by manuscript section, along with brief descriptions (and indicating in parenthesis which reviewer asked for the clarification):

**Introduction:**

1. Removed the brief mention to out-of-distribution generalization from second paragraph. (zUsD)

2. Added the Xu et al. (2025) reference. (4vKk)

**Notation:**

1. Created a separate notation section. (PSBh)

2. Clarified the notation around the use of random variable vectors versus data matrices. (PSBh)

3. Changed order in which the $X_{-j}$ notation is introduced, and added a description about the $X_{<j}$ notation. (PSBh)

4. Provided a more detailed explanation regarding the notation describing the TabPFN model. (PSBh)

**Related work:**

1. Created a separate related work section. (PSBh)

**Direct strategies for SDG based on TabPFN:**

1. Clarified the $p$ notation. (PSBh)

2. Added description about the permutation sampling approximate of Janossy pooling. (PSBh)

**Constructing maximal information auxiliary variables:**

1. Added a sentence clarifying that the MIAV approach is not sensitive to the choice of noise distribution used in its construction. (DDxA)

2. Added clarification about the colors in the correlation plot panels. (PSBh)

3. Added clarification that $M_j$ determines $X_j$ in a non-parametric rank-based sense. (PSBh)

**Experiments based on TabPFN models:**

1. Improved the structure of the section by including paragraphs titled: “Experiments”, “Data splits”, “Evaluation metrics”, “Baselines”, “Experimental details” and a subsection titled “Results”.  (PSBh)

2. Added descriptions of the additional evaluation metrics to the metrics paragraph. (PSBh)

3. Expanded the Results subsection to describe results based on the additional metrics. (PSBh)

**Experiments based on TabICL models:**

1. Created a separate section to describe the experiments based the TabICL model. (This information was described previously in the last paragraph of the Experiments section.)

**Final remarks:**

1. Expanded our discussion on the limitations and scalability of the proposed approach. (zUsD, DDxA)

2. Expanded our discussion that the MIAV approach is only meant for synthetic data generation and that its misapplication to supervised learning tasks is problematic. (DDxA)

**Appendix D:**

1. Added clarification about the colors in the correlation plot panels. (PSBh)

**Appendix H.2:**
1. Scaled up the runtime experiments with 4 new experiments based on 4000, 5000, 10000, and 20000 instances. (zUsD)

**Appendix I.5 and Appendix I.6:**

1. Added descriptions for the 3 additional evaluation metrics included in the manuscript. (PSBh)

2. Included a new appendix section describing the results based on the 3 additional evaluation metrics. (PSBh)

**Appendix L:**

1. Included a new appendix section describing the results of the sensitivity analysis to the noise distribution used to construct the MIAVs. (DDxA)

---

### Meta-Review · Area_Chair_koVc · 2025-12-08

**Summary:**

The paper introduces MIAV, an auxiliary-variable strategy that mitigates the degeneration of TabPFN-based synthetic data generators when features have weak associations. Reviewers broadly agree that the problem is important, the method is simple and theoretically grounded, and the empirical evaluation is extensive. Reviewers `4vKk`, `zUsD`, and `DDxA` judged the contribution solid and raised focused technical questions, all of which were addressed in the rebuttal through clarifications, added theory, expanded experiments (including runtime scaling to 20k rows), additional metrics (Wasserstein, energy distance, MLE) and improved discussions of limitations and proper use. Reviewer `PSBh` initially had substantial concerns about clarity and notation. The authors improved the presentation, corrected notation, expanded explanations and addressed additional technical questions. After rebuttal, all reviewers except `PSBh` clearly recommended acceptance, but `PSBh` acknowledged major improvement and was willing to raised their score.

Overall, the submission now presents a clear story, addresses the core methodological limitation of TabPFN for SDG, and provides compelling empirical evidence and theoretical justification.

**Reviewer Concerns:**

Below are the main concerns from each reviewer that the rebuttal addresses:

- `R4vKk`’s questions on the relationship to Bayesian structure-learning and ordering-based methods were addressed through theoretical clarification and explicit comparison to PAFT.

- `zUsD`’s concerns regarding OOD claims, runtime scalability, and scaling in number of features were fully resolved via manuscript edits, new experiments up to 20k rows, and clarified limitations.

- `DDxA`’s questions on noise distribution sensitivity, MIAV applicability, and privacy-leakage risks were addressed through new experiments (Appendix L), strengthened warnings, and expanded limitations.

- `PSBh`’s concerns on presentation, notation, undefined symbols, figure readability, missing metric families, and clarity around Theorem 1 were comprehensively addressed through manuscript restructuring and additional analyses.

After the rebuttal, there are no relevant outstanding concerns.

**Reviewer Scores:**

`4vKk`: Score likely unchanged, as it was already an accept (8).

`zUsD`: Explicitly stated that they would increase their score, presumably from weak accept (6) to accept (8).

`DDxA`: Score likely unchanged, as it was already an accept (8).

`PSBh`: Indicated that they would raise their score after the substantial presentation improvements; likely moving from reject (2) to weak accept (6) (or possibly only to weak reject (4)).

---

### Decision · Program_Chairs · 2026-01-26

Accept (Poster)